# Structure-guided screening identifies Tucatinib as dual inhibitor for MCT1/2

Binghong Xu [ID][1,5][✉], Xiaoyu Zhou[1,5], Yuanyue Shan [ID][2,5], Sai Shi[1,3,5], Jiachen Li[1,5], Qinqin Liang[1], Ziyu Wang[1], Mingfeng Zhang [ID][4][✉], Yaxin Wang [ID][1][✉], Duanqing Pei [ID][4][✉] & Sheng Ye [ID][1][✉]

## Abstract

Cell surface glycoproteins Basigin or embigin form heterodimers with monocarboxylate transporters (MCTs), enhancing their membrane trafficking and modulating their transport functions. Cancer cells often reprogram their metabolism and depend on proton-coupled lactate transport mediated by MCTs to sustain their glycolytic state and to maintain intracellular pH. A deeper understanding of MCTs regulation may open avenues for the development of novel inhibitors, potentially applicable in clinical settings. Here, we determine the cryo-EM structures of the human MCT2-embigin complex in both apo and AR-C155858-bound states and observe that embigin engages in extensive interactions with MCT2, facilitating its localization to the plasma membrane and substrate transport. Given the high structural conservation among MCTs, we conduct virtual screening based on MCT1/2 structures and identify Tucatinib as an effective inhibitor of pyruvate transport mediated by both MCT1 and MCT2. We show that Tucatinib potently inhibits the proliferation and migration of cervical tumor cells in vitro and tumor growth in a mouse xenograft model, while exhibiting excellent biological safety. These findings offer molecular insights into the structural and functional mechanism of MCT2 and identify Tucatinib as novel dual inhibitor of both transporters.

Keywords Monocarboxylate Ttransporters; Cryo-EM; Cervical Cancer; Drug Targets; Tucatinib
Subject Categories Cancer; Membranes & Trafficking; Structural Biology

## Introduction

Monocarboxylates, including lactate, pyruvate, ketone bodies and so on, are crucial for maintaining cellular monocarboxylate homeostasis and pH balance (Adijanto and Philp, 2012; Halestrap and Price, 1999). In humans, the rapid exchange and circulation of these metabolically significant intracellular monocarboxylates are primarily mediated by monocarboxylate transporters (MCTs) encoded by the solute carrier family 16 (SLC16) (Felmlee et al, 2020a; Halestrap, 2012, 2013a, b). Among the 14 MCT members, MCT1–MCT4 are known to catalyze proton-coupled transport of monocarboxylates (Halestrap and Meredith, 2004; Poole and Halestrap, 1993), with the direction of transport determined by both the proton motive force and the concentration gradient of the monocarboxylate substrates.

Functionally active MCT1-4 typically require ancillary proteins for proper membrane localization and transporter activity. Two glycoproteins, Basigin (CD147) and embigin (gp70), have emerged as molecular chaperones that facilitate membrane trafficking of MCT1-4 (Poole and Halestrap, 1997; Wilson et al, 2005). Specifically, MCT1, MCT3, and MCT4 preferentially interact with Basigin (Halestrap, 2013b; Philp et al, 2003), whereas embigin is recognized as the auxiliary protein for MCT2 (Ovens et al, 2010b; Wilson et al, 2005). Notably, embigin could also act as a chaperone for the intracellular trafficking of MCT1 to the plasma membrane (Wilson et al, 2005; Xu et al, 2022).

In the absence of ancillary proteins, MCT2 alone forms a dimer, contributing to the cooperative transport, which is crucial for maintaining an intracellular monocarboxylate homeostasis (Zhang et al, 2020). The Halestrap group demonstrated that the associated ancillary protein modulates MCTs' function (Ovens et al, 2010b; Wilson et al, 2005), though the molecular basis of the modulation is unclear. The cryo-electron microscopy (cryo-EM) structures of the human MCT1-basigin-2 complex were determined, revealing the substrate recognition, proton-coupling, and mode of action of three anti-cancer drug candidates (Wang et al, 2021) but leaving other questions unanswered. Subsequently, our group determined the cryo-EM structure of the MCT1-embigin heterodimer. Comparative analysis with the homodimeric structure of MCT2 revealed several key structural differences: Embigin effectively creates steric hindrance to block the homodimerization of MCTs; TM1 undergoes a conformational shift, impairing the hydrogen-bond network between TM1 and TM4–5 that underlies cooperativity in MCT2; TM5 is straight in MCT1-embigin but bent ~30° toward the cavity in MCT2 homodimer, altering substrate affinity, which was confirmed by SPR assay (Xu et al, 2022). We also observed that

[1]State Key Laboratory of Synthetic Biology, Tianjin Key Laboratory of Function and Application of Biological Macromolecular Structures, School of Life Sciences, Tianjin University, Tianjin 300072, China. [2]GIBH-HKU Guangdong-Hong Kong Stem Cell and Regenerative Medicine Research Centre, Hong Kong Institute of Science & Innovation, Guangzhou, Guangdong, Guangdong 510530, China. [3]Department of Medical and Pharmaceutical Informatics, Hebei Medical University, Shijiazhuang 050017, China. [4]Laboratory of Cell Fate Control, School of Life Sciences, Westlake University, Hangzhou 310000, China. [5]These authors contributed equally: Binghong Xu, Xiaoyu Zhou, Yuanyue Shan, Sai Shi, Jiachen Li. ✉E-mail: binghong_xu@tju.edu.cn; zhangmingfeng@westlake.edu.cn; wangyaxin@tju.edu.cn; peiduanqing@westlake.edu.cn; sye@tju.edu.cn

the substrate-binding pocket is largely conserved between MCT1 and MCT2. Key residues forming the monocarboxylate-binding site include Tyr34 and Lys38 on TM1, Asp309 (Asp293 in MCT2) and Arg313 (Arg297 in MCT2) on TM8, as well as Phe367 (Phe351 in MCT2) and Ser371 (Ser355 in MCT2) on TM10. Through structural comparison, the structure of the MCT1-embigin complex is highly similar to that of the MCT1-basigin-2 complex (Wang et al, 2021), which is not surprising given that embigin is homologous with basigin, especially in the transmembrane helix. Moreover, the binding affinities of MCT1-basigin-2 complex for pyruvate and L-lactate are similar to that of MCT1-embigin complex (Xu et al, 2022). These findings demonstrate that the presence or absence of ancillary proteins significantly influences both the structure and function of MCTs. Consequently, the first question arises: what is the molecular basis for embigin's modulation of MCT2 function?

MCT1/2/4 serve as major effectors underlying the aberrant metabolic rewiring observed in cancer (Payen et al, 2020). In glycolytic tumors, MCTs facilitate lactic acid efflux, crucial for maintaining intracellular pH, thereby preventing apoptosis and fostering a conducive microenvironment for tumor invasion (Feron, 2009; Sonveaux et al, 2008). Meanwhile, cancer cells undergoing oxidative stress induce adjacent stromal fibroblasts to adopt aerobic glycolysis, thereby acquiring additional lactate through MCT1/4-facilitated lactate transport (Pavlides et al, 2009; Whitaker-Menezes et al, 2011). The lactate shuttling fosters "metabolic symbiosis" in the tumor microenvironment, thereby contributing to cancer development. Inhibiting MCTs to interfere with lactate metabolism offers a promising chemotherapy strategy (Payen et al, 2020). Early MCTs transport inhibitors such as phloretin, quercetin, CHC, and DIDS exhibited low affinity and poor specificity (Carpenter and Halestrap, 1994; Park et al, 2018; Perez-Escuredo et al, 2016). Recently, several more potent MCT inhibitors, including AR-C155858, AZD3965, BAY-8002, and 7ACC2, have been identified and developed (Draoui et al, 2013; Ovens et al, 2010a; Perez-Escuredo et al, 2016; Quanz et al, 2018). AR-C155858, a pyrrole pyrimidine derivative, demonstrated potent inhibition of MCT1 and MCT2 with $K_i$ values of 2.3 nM and <10 nM, respectively (Ovens et al, 2010a; Ovens et al, 2010b). As a derivative of AR-C155858, AZD3965 is a second-generation dual MCT1/2 inhibitor developed by AstraZeneca, with $K_d$ values of 1.6 nM and 9.6 nM for MCT1 and MCT2, respectively (Critchlow et al, 2012). Currently, AZD3965 is in Phase I clinical trials for adult solid tumors, including Burkitt lymphoma and diffuse large B-cell lymphoma (DBCL; NCT01791595) (Halford et al, 2017; Halford et al, 2021). Despite promising antitumor efficacy, AZD3965 exhibits multiple adverse effects, such as retinopathy, fatigue, anorexia and constipation (Halford et al, 2023), thus leading to slow pharmacological progress. Additionally, only a limited number of MCT-targeted inhibitors have reached clinical investigation stages. Hence, the second issue emerges: how to develop novel MCT-targeted inhibitors for future cancer therapy?

Cervical cancer is a major contributor to cancer-related morbidity and mortality globally, affecting over 660,000 women annually and resulting in over 348,000 deaths, particularly in developing countries (Sahasrabuddhe, 2024). Chronic persistent oncogenic human papillomavirus (HPV) infection has been identified as a causative factor for cervical cancer. Although HPV vaccines are currently available, they primarily protect individuals who have not yet been exposed to the virus, thereby reducing their effectiveness for those who are already infected. In addition, there are still a large number of people in the world who are unable to access HPV vaccines. Particularly, even in the early stages of cervical cancer, lymph node metastases can occur, which directly reduces the effectiveness of treatment (Cho et al, 2019). Therefore, exploring new molecular targets in cervical cancer is essential for understanding the potential mechanisms driving its carcinogenesis. Lactic acid can participate in angiogenesis and tissue remodeling, regulate the metabolism of innate and adaptive immune cells, and promote tumor growth and invasion (Certo et al, 2021a; Certo et al, 2021b), which is also applicable in cervical cancer (Yang et al, 2023). Studies have shown that monocarboxylic acid transporters MCT1 and MCT4 are significantly overexpressed in cervical cancer (Yang et al, 2023). This leads to the third question: Can inhibitors targeting MCT1/2 effectively inhibit cervical cancer?

To address the above questions, we performed cryo-EM structural analysis on MCTs and conducted structure-based drug screening. We solved cryo-EM structures of the human MCT2-embigin complex in apo and AR-C155858-bound states, both in outward-facing form. Cell colocalization and substrate transport assay show that embigin could facilitate MCT2 proper trafficking and function at the plasma membrane. The detailed structure-function analyses further verify the conservation of MCT1 and MCT2 in terms of their structure and function. Based on the structures of MCT1-Basigin-2 and MCT2-embigin, we performed virtual screening and identified Tucatinib, an investigational, oral tyrosine kinase inhibitor, as a compound with the potential to inhibit the transport activity of both MCT1 and MCT2. The computational and mutation analyses show that Tucatinib occupies the central substrate-binding cavity and blocks the transport activity of MCTs by preventing alternating access. Additionally, cellular and animal-based experiments have demonstrated that Tucatinib exhibits potent inhibitory effects on the proliferation and migration of HeLa cells, ultimately resulting in the attenuation of cervical carcinoma progression in murine models. Taken together, our results reveal the structural and mechanistic features of the MCT2-embigin complex and provide a structural platform for anti-cancer drug development targeting MCT1/2.

## Results

### Structural determination of MCT2-embigin complex

MCTs rely on a variety of ancillary proteins for their proper trafficking and function at the plasma membrane. Embigin, one such protein, has been shown to closely associate with MCT2 in kidney plasma membranes (Wilson et al, 2005). Similar to the structural study of MCT1-embigin complex (Xu et al, 2022), we identified the fusion protein as a promising candidate for structural studies of MCT2-embigin complex. The fusion protein was expressed and purified from HEK-293F cells. For cryo-EM sample preparation, the purified proteins were supplemented with AR-C155858 at a final concentration of 100 μM. We determined the cryo-EM structure of MCT2-embigin at a resolution of 3.67 Å in the apo state (named MCT2-emb) and 3.2 Å in the AR-C155858 bound state (named MCT2R-emb) (Appendix Figs. S1 and S2; Appendix Table S1). The MCT2-emb density map shows a clear,

well-resolved transmembrane domain (TMD) with clearly visible α-helical features, enabling the building of a molecular model that includes all side chains for all the transmembrane helices (TMs) of MCT2, together with most of the loops between TMs, and the single transmembrane helix (TM) of embigin (Appendix Figs. S1 and S3). Similar to that of MCT1-embigin (Xu et al, 2022) and MCT1-Basigin-2 (Wang et al, 2021), the large cytosolic loop between TM6 and TM7 of MCT2 was not modeled due to poor density. Additionally, the extracellular domain (ECD) in embigin has a lower resolution, reflecting its flexible nature, allowing only for the tracing of the TM-connected Ig2 domain and rigid-body docking of the distal Ig1 domain (Fig. 1A,B).

MCT2 exhibits a characteristic major facilitator superfamily (MFS) fold that includes 12 TMs organized into two bundles, each comprising six helices: the N-terminal domain (NTD) and the C-terminal domain (CTD) (Fig. 1B–D). The MCT2 structure assumes an outward-open conformation, thus creating a large cavity that is exclusively connected to the extracellular side (Fig. 1B,C). We used a pull-down assay to characterize the expressions and interactions of MCT2 and embigin (Appendix Table S2). The single TM of embigin primarily interacts with TM6 of MCT2 through hydrophobic interactions (Fig. 1E–H). Besides, two hydrogen bonds and a salt bridge between embigin and MCT2 play pivotal roles in the heterodimeric complex: embigin (D174)-MCT2 (K175), embigin (E270)-MCT2 (N187), and embigin (E282)-MCT2 (R86) (Fig. 1E–H).

A domain-wise comparison of MCT2-emb (PDB ID: 9L0B) and MCT1-embigin (MCT1-emb, PDB ID: 7YR5, inward-open) reveals that the alternating access of MCT2 is achieved through rigid-body rotation of NTD and CTD (Fig. EV1A). The two domains of MCT2-embigin can be respectively superimposed with those in MCT1-embigin with root-mean-standard deviation (RMSD) values of 0.92 Å over 184 Cα atoms and 0.98 Å over 183 Cα atoms (Fig. EV1A). To keep the TM region within the membrane, the NTD and CTD both undergo rotations relative to an axis that is parallel to the membrane plane by ~19° and 17°, respectively (Fig. EV1B). To reveal domain movements in state transitions, we aligned the two structures with the MCT2 N-domain as a reference, which demonstrates that the C-domain of the cytosol-facing MCT1 swings by ~30 degrees when transitioning to the MCT2 lumen-facing state (Fig. EV1C). These global movements are in line with the "rocker switch" like alternating-access mechanism of MFS transporters.

MCT2 forms a homodimer, and the dimer interface is extensive, involving 4TMs from each subunit (Zhang et al, 2020) (Fig. EV1D). However, similar to its effect on MCT1, embigin prevents MCT2 from forming a homodimer, likely by creating steric hindrance that blocks homodimerization (Xu et al, 2022). To determine whether embigin is a suitable molecular chaperone, we expressed a fusion of MCT2-embigin tagged with GFP in HEK-293T and HeLa cells and observed their localizations by confocal imaging (Fig. 1I). When embigin was fused to the N-terminus of MCT2 with a four-amino acid linker (GGSG), single transfection led to substantial expression of the fusion protein at the cell surface. In addition, co-expression of MCT2 and embigin also led to substantial expression of both proteins at the cell surface, implying that embigin facilitates MCT2 trafficking to the plasma membrane.

A comparison of the structure between the MCT2-embigin complex and the MCT2 dimer reveals similarities to the differences observed between MCT1-embigin and MCT2 (Xu et al, 2022). The NTD and CTD of MCT2 can be respectively superimposed with those in MCT2-emb with RMSD values of 1.85 Å over 180 Cα atoms and 1.09 Å over 201 Cα atoms (Fig. EV1E). During the transition from the homodimeric to the heterodimeric state, TM5 undergoes a significant conformational change, bending approximately 30° away from the central cavity (Fig. EV1F,G). In addition to TM5, TM1, and TM4 also exhibit slight deflections (Fig. EV1F). These structural discrepancies further complement our previous research findings that during the conformational change from homodimeric coupling to heterodimeric decoupling in MCTs, the monocarboxylate-binding site exhibits two distinct stereochemical binding modes (Xu et al, 2022). Actually, we attempted to solve the cryo-EM structure of MCT2-AR-C155858 complex but were unsuccessful. Despite the addition of excess AR-C155858 to the cryo-EM sample of MCT2, no extra density was observed in the central cavity. This suggests that the observed difference in substrate-binding pocket between the two stereochemical binding modes may be one possible explanation for the lack of detectable bound inhibitor.

## Inhibition of MCT2 by AR-C155858

The secondary structure features of MCT2R-emb are well-defined, allowing us to visualize the binding mode of AR-C155858 (Appendix Figs. S2 and S4). The overall structure of the outward-open MCT2R-emb closely resembles that of MCT2-emb, with an RMSD of 0.67 Å over 387 aligned Cα atoms (Fig. EV2A). The Y-shaped AR-C155858 comprises three ring structures, the thienopyrimidine-2,4-dione ring, the pyrazol ring, and the oxazolidine ring, all of which are well-resolved (Appendix Fig. S4A). The largest thienopyrimidine-2,4-dione ring is positioned in a big cleft between the NTD and CTD, while the other two rings extend into two smaller pockets slightly above (Fig. 2A–C; Appendix Fig. S4B). The thienopyrimidine-2,4-dione core is anchored by Y34, Y70, F258, R297, and F351, engaging in a combination of hydrophobic and polar interactions. Notably, Y34 and F351 occupy diametrically opposed positions on the pyrimidine plane, while R297 establishes hydrogen bonds with the 4-carbonyl moiety. The pyrazol ring is snugly accommodated within a hydrophobic pocket formed by P37, K38, L66, F262, and I265. Furthermore, the oxazolidine ring interacts with S154 and P155, and its 4-hydroxyl group forms a hydrogen bond with D293, emphasizing its intricate bonding network (Figs. 2B,C and EV2B). In comparison to the architecture of MCT2-emb, the side chains of K38 and F262 in MCT2R-emb undergo repositioning to avoid clashes with AR-C155858 (Fig. EV2C–E).

Moreover, we compared MCT2R-emb (PDB ID: 9L0C) and MCT1A (MCT1-Basigin-2-AZD3965, PDB ID: 6LYY (Wang et al, 2021)), both in an outward-open conformation, and found a high degree of structural similarity, with an RMSD of 0.85 Å over 367 aligned Cα atoms (Fig. EV2F). Despite binding to different protein targets, the residues surrounding AR-C155858 in MCT2 and those around AZD3965 in MCT1 exhibit highly similar conformations (Fig. EV2G–J). The structural conservation, particularly within the substrate-binding cavity, explains why both AR-C155858 and AZD3965 can act as dual inhibitors of MCT1 and MCT2. This remarkable structural conservation between MCTs is a testament to the importance of preserving their structure to catalyze selective

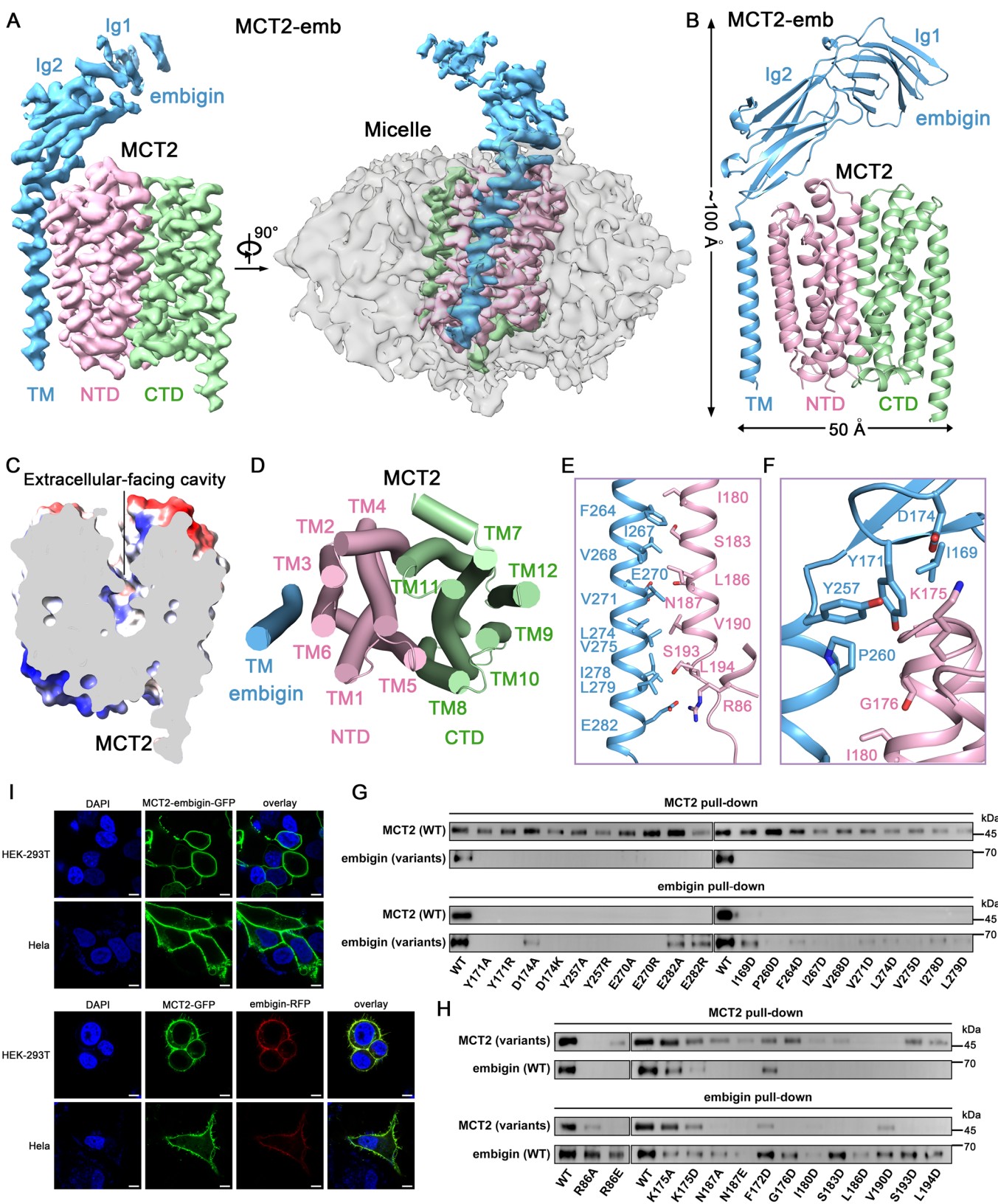

◀ **Figure 1.  Structure of human MCT2-embigin complex in the apo state.**

(A) 3D EM reconstruction of MCT2-emb. Two perpendicular side views are shown. The GDN micelle is presented as a semi-transparent surface on the right. (B) Overall structure of MCT2-emb at outward-open conformation. NTD (pink): amino terminal domain; CTD (green): carboxy terminal domain; Ig (blue): immunoglobulin; TM (blue): transmembrane segment. (C) A slab of cut-open side view of the electrostatic surface potential of MCT2 is shown to facilitate visualization of the outward-facing cavity. (D) Cylinder representation of MCT2-emb structure viewed from the intracellular side. TM regions of MCT2 and embigin are displayed as cylinders and colored by pink (NTD of MCT2), green (CTD of MCT2) and blue (TM of embigin), respectively. (E) The interactions between the transmembrane segment of embigin and TM3, TM6 of MCT2. The residues that constitute the transmembrane interface are shown as sticks. (F) The extracellular interface of embigin and MCT2. The residues that constitute the extracellular interface are shown as sticks. (G) Mutations of embigin-Tyr171, Asp174, Tyr257, Glu270, Glu282, Ile169, Pro260, Phe264, Ile267, Val268, Val271, Leu274, Val275, Ile278, Leu279, influenced complex formation with MCT2. (H) Mutations of MCT2-Arg86, Lys175, Asn187, Phe172, Ile180, Ser183, Leu186, Val190, Ser193, Leu194, partially affected complex formation with embigin. WT means wildtype of MCT2 and embigin. Shown here are the pull-down results. (I) HEK-293T or HeLa cells were transiently transfected with MCT2-embigin fusion tagged with GFP (Top 2 lines). HEK-293T or HeLa cells were transiently transfected with GFP-tagged MCT2, and RFP tagged embigin (Bottom 2 lines). Excitation confocal microscopy was used to reveal the expression and location of MCT1-embigin fusion, MCT2 and embigin. The bars represent 5 μm. Source data are available online for this figure.

monocarboxylate transport across membranes (Appendix Figs. S5 and S6).

To further validate the binding position of AR-C155858 within the MCT2 structure, we performed molecular docking and molecular dynamics (MD) simulations to corroborate the accuracy of its placement based on the cryo-EM density. The structural alignment results show that the docking-based AR-C155858 binding conformation closely matches the molecular pose observed in the structure (Fig. 2D). In addition, we performed molecular dynamics simulations using MCT2R-emb (PDB ID: 9L0C) as the starting model. The RMSD of the protein backbone atoms in three 500-ns simulation systems was less than 3 Å, and the RMSD of AR-C155858 was less than 1.5 Å without significant fluctuations, indicating the stability of the MCT2R-emb complex (Fig. 2E,F; Appendix Fig. S7A,B). The binding free energy decomposition spectra based on MMGBSA calculations revealed that Y34, F262, D293, R297, and F351 are key residues stabilizing the binding of AR-C155858 (Fig. 2G,H).

To validate the structural observations, we sought to examine the inhibitory effect of AR-C155858 on MCT2-embigin variants. HEK-293T cells co-transfected with pyronicSF, a highly responsive pyruvate sensor (Arce-Molina et al, 2020), and either empty vector control or MCT2-embigin variants, were exposed to pyruvate, leading to an increase in the GFP-based fluorescence signal indicative of pyruvate uptake. This signal was significantly decreased upon pre−incubation with AR-C155858 (Fig. 2I). Because of the overlapping binding site, most of the tested mutations affected the transport activity of MCT2-embigin (Fig. 2I). We then used Bio-Layer Interferometry (BLI) to measure the binding affinity of AR-C155858 and MCT2-embigin variants or MCT1-Basigin-2. The affinity between AR-C155858 and wild-type MCT2-embigin is 0.6 μM (Fig. 2J,L), while its affinity for MCT1-Basigin-2 is slightly stronger at 0.1 μM (Fig. 2K,L). Consistent with the structural analysis, the affinity between AR-C155858 and MCT2-embigin mutants decreased to varying degrees (Figs. 2L and EV3A–E). The most significant effect was observed with the R297A mutation, where affinity decreased approximately 100-fold (Fig. EV3D), while the impact of F351 was relatively smaller, showing an approximately 11-fold reduction (Fig. EV3E).

## Tucatinib represents a novel dual inhibitor of MCT1 and MCT2

We performed a virtual screen of more than 2300 drugs based on their binding affinity to the substrate pockets of MCT1 and MCT2,

aiming to discover novel inhibitors with higher activity (Fig. 3A,B). The selection of candidates for pyruvate transport activity testing was based on the predicted affinity between the drugs and MCT1/2, as determined by the Vina function (Fig. 3C). The pyruvate transport assay revealed that Tucatinib, a potent, orally active, and selective HER2 (human epidermal growth factor receptor 2) inhibitor, significantly inhibited pyruvate uptake by both MCT1 and MCT2 in HEK-293T cells (Fig. 3D−F), with IC$_{50}$ values of 2.596 μM for MCT1 and 2.712 μM for MCT2, respectively (Fig. 3G,I). As positive controls, the IC$_{50}$ values for AZD3965 on MCT1 and AR-C155858 on MCT2 were measured as 1.238 μM and 0.981 μM, respectively (Fig. 3H,J). Although the inhibitory effect of Tucatinib is slightly less than that of these commercial inhibitors, its performance is still notably impressive. Additionally, we examined the inhibitory effect of Tucatinib on MCT2 transport activity under different pH conditions. The results demonstrated that Tucatinib effectively inhibits substrate transport by MCT2 in neutral and acidic environments (Fig. EV4A–D). However, this inhibitory effect was significantly attenuated under alkaline conditions (Fig. EV4E,F).

To visually assess the inhibitory effect of Tucatinib, we used confocal microscopy to observe the inhibition of pyruvate transport by MCT1/2 under varying concentrations of Tucatinib. The results indicate that a significant inhibitory effect on pyruvate transport is noticeable when the concentration of Tucatinib exceeds 6 μM. Moreover, at a concentration of 25 μM, pyruvate transport is nearly completely suppressed (Fig. 3K,L), demonstrating the potent inhibitory capability of Tucatinib against MCT1/2-mediated pyruvate transport.

To elucidate the interaction mode between Tucatinib and MCT1/2, we performed docking and MD simulations based on the cryo-EM structures of the inhibitor binding states (MCT1A, PDB ID: 6LYY; MCT2R-emb, PDB ID: 9L0C). The docked binding mode of MCT2-Tucatinib was similar to that of MCT1-Tucatinib, with an RMSD of 1.00 Å over 377 aligned Cα atoms (Fig. 4A–C). Appendix Fig. S8A,B depict detailed views of the Tucatinib-protein interactions. To further determine the molecular mechanism of MCT1/2 inhibition by Tucatinib, we constructed MCT1-Tucatinib and MCT2-Tucatinib mimetic systems and performed all-atom simulations for multiple 500 ns (Appendix Fig. S7C,D for MCT1, Appendix Fig. S7E,F for MCT2). We tested the binding affinity of Tucatinib to MCT1 and MCT2, with a binding free energy of −35.9, −36.7 kcal/mol, respectively. MMGBSA decomposition mapping revealed that several key residues in the substrate active pocket are essential for stabilizing Tucatinib binding, including

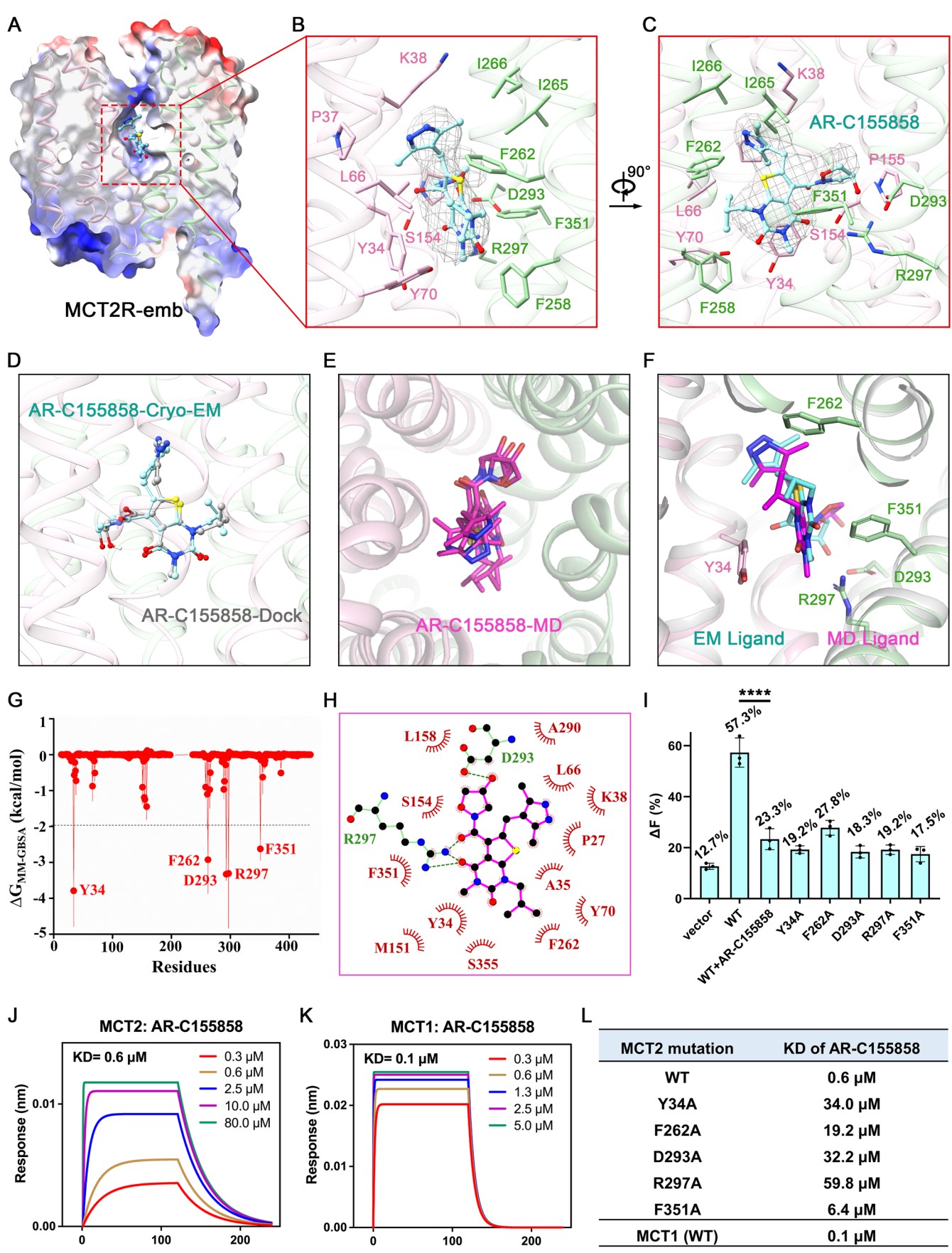

**Figure 2. Molecular basis for the inhibition of MCT2 by AR-C155858.**

(A) Cut-open side views of the electrostatic surface potential of MCT2R-emb. AR-C155858 is buried in a large pocket formed by the NTD and CTD of MCT2. (B, C) Coordination of AR-C155858 by MCT2. Both hydrophobic residues and hydrophilic residues are involved in inhibitor recognition. (D) Molecular docking and cryo-EM structural overlap. The pink-green-cyan structure represents the resolved cryo-EM structure of the MCT2-embigin-AR-C155858 complex, while the gray structure shows the computationally docked pose of AR-C155858 into the MCT2-embigin binding site. (E) Multiple molecular simulations of steady-state structures overlap. (F) Molecularly simulated steady-state structures overlap with the cryo-EM structure. (G) Binding free energy decomposition mapping. (H) Interactions between AR-C155858 and MCT2 in the MD state. Schematics is generated by LIGPLOT + 1.4. Each eyelash motif indicates a hydrophobic contact. Green dashed lines indicate hydrogen bonds between inhibitor and residues. (I) Biochemical validation of the potential AR-C155858 binding residues using the pyruvate transport assay. Vector refers to the uptake by HEK-293T cells transfected with an empty vector. The graphical presentation and data analysis were conducted using GraphPad Prism 9. The data are displayed as mean ± standard deviation (SD). $n = 3$ biological replicates. Bio-layer interferometry (BLI) measurement of the binding affinities between AR-C155858 and MCT2-embigin (J) or MCT1-Basigin-2 (K). (L) Bio-layer interferometry (BLI) measurement of the binding affinities between AR-C155858 and MCT2-embigin variants or MCT1-Basigin-2 .Source data are available online for this figure.

Y34, L66, F278, L281, and L302 in MCT1; and Y34, L66 F258, F262, I265, L286, and F351 in MCT2 (Fig. 4D,G). Tucatinib binds in a similar pattern in MCT1 and MCT2, with the quinazoline ring binding deep in the pocket, the triazole oriented toward the outside of the pocket, and the hydrophobic residues wrapping around the entire molecule (Fig. 4E,F,H,I), demonstrating the substrate competitive inhibition mechanism of Tucatinib.

To validate the structural observations, we sought to examine the inhibitory effect of Tucatinib on MCT1 or MCT2 variants, each containing a single point substitution of some of the key interaction residues. We used biolayer interferometry (BLI) to measure the binding affinity of Tucatinib and MCT1 or MCT2 variants (Fig. 4J–P). Consistent with the structural analysis, F278A resulted in undetectable binding between Tucatinib and MCT1 (data not detectable), and Y34A and R313A weakened the interaction by approximately 11- and 15-folds, respectively (Fig. 4J–L). Among the three MCT2 mutants, F351A exhibited the most pronounced effect, with a 350-fold reduction in binding affinity. The other two mutants, Y34A and F262A, showed reductions of 23-fold and 55-fold, respectively (Fig. 4M–P).

## Tucatinib inhibits the proliferation and migration of HeLa cells by targeting MCT1/2

MCT1, MCT2, and MCT4 have been established as key players in cancer progression via diverse mechanisms (Payen et al, 2020). Tucatinib, a potent, orally bioavailable, and selective HER2 inhibitor, has shown efficacy in the treatment of HER2-positive breast cancer. To investigate the targeted inhibition of MCT1/2 by Tucatinib in cancer cell development while excluding the influence of the HER2 gene, we chose HER2-deficient HeLa cells to conduct research on the anti-cancer activity of Tucatinib (Jia et al, 2022). Moreover, we measured the mRNA expression levels of HER2 relative to MCTs in cells. The results indicated that the expression level of HER2 was lower than that of MCT1 or MCT2, accounting for less than 7% of their expression levels (Appendix Fig. S9).

First, we assessed MCT1/2 expression in HeLa cells and the mouse islet β-cell line Min6, using Min6 as a negative control due to its lack of endogenous MCT1 or MCT2 expression (Zhao et al, 2001). Figure 5A,B show high expression of both MCT1 and MCT2 in HeLa cells, in contrast to Min6 cells. We then examined the inhibitory activity of Tucatinib on the proliferation and colony formation of HeLa cells. The CCK-8 assay demonstrated a concentration-dependent inhibition of HeLa cells' viability by Tucatinib (Fig. 5C), with no significant effect on the viability of

Min6 cells (Fig. 5D). Additionally, CCK-8 assays with AR-C155858 and AZD3965 revealed that both inhibitors could inhibit the proliferation of HeLa cells, albeit to a lesser extent than Tucatinib (Fig. 5E; Appendix Fig. S10A). These two inhibitors also had minimal impact on Min6 cells (Fig. 5F; Appendix Fig. S10B). Colony formation assays were utilized to further assess the suppressive influence of Tucatinib on cell proliferation. The inhibition rates of HeLa cell colony formation by 12.5 μM Tucatinib reached 73.7%, while the suppression rates of 25 μM AR-C155858 and AZD3965 were 48.6 and 59.8%, respectively (Fig. 5G).

To affirm MCT1/2 as the target of Tucatinib, we employed shRNA to knock down MCT1 or MCT2 expression in HeLa cells (Fig. 5H,I). Notably, the viability of HeLa cells significantly diminished subsequent to shRNA transfection, while the influence of sh-MCT1 is slightly greater than that of sh-MCT2. The addition of 12.5 μM Tucatinib did not replicate this decrease but did have some effect (Fig. 5J). Since we knocked down MCT1 or MCT2 individually, this result aligns with our expectations, confirming both MCT1 and MCT2 as the targets of Tucatinib. Furthermore, the results of the gene rescue experiment demonstrated that when MCT1 or MCT2 was reintroduced into the corresponding knock-down cell lines, the phenomenon of Tucatinib-induced inhibition of cell proliferation reappeared. However, the mutant of MCT1 (F278A) or MCT2 (F351A), which is a key residue for Tucatinib binding to MCT1 or MCT2, failed to achieve a similar effect. These findings once again provide evidence for the targeting of Tucatinib towards MCT1/2 (Appendix Fig. S11A,B).

When MCT1 or MCT2 expression was silenced through shRNA, a significant decrease in colony count was observed. Notably, supplementing with 12.5 μM Tucatinib did not enhance the inhibitory effect in either sh-MCT1 or sh-MCT2 groups, compared to corresponding sh-NC groups (Fig. 5K,L). The CCK-8 and colony formation assays highlight the specificity and differential efficacies of these compounds in targeting cell proliferation, with Tucatinib demonstrating a more pronounced inhibitory effect on HeLa cells than AR-C155858 or AZD3965 by targeting MCT1 and MCT2.

Furthermore, the wound-healing assay was employed to assess the inhibitory effect of Tucatinib on HeLa cell migration. As depicted in Fig. 6A, the findings revealed a significant suppression of HeLa cell migration by Tucatinib. Statistical analysis further confirmed that this inhibition was both concentration- and time-dependent, as varying concentrations of Tucatinib produced corresponding reductions in cell migration (Fig. 6B). For

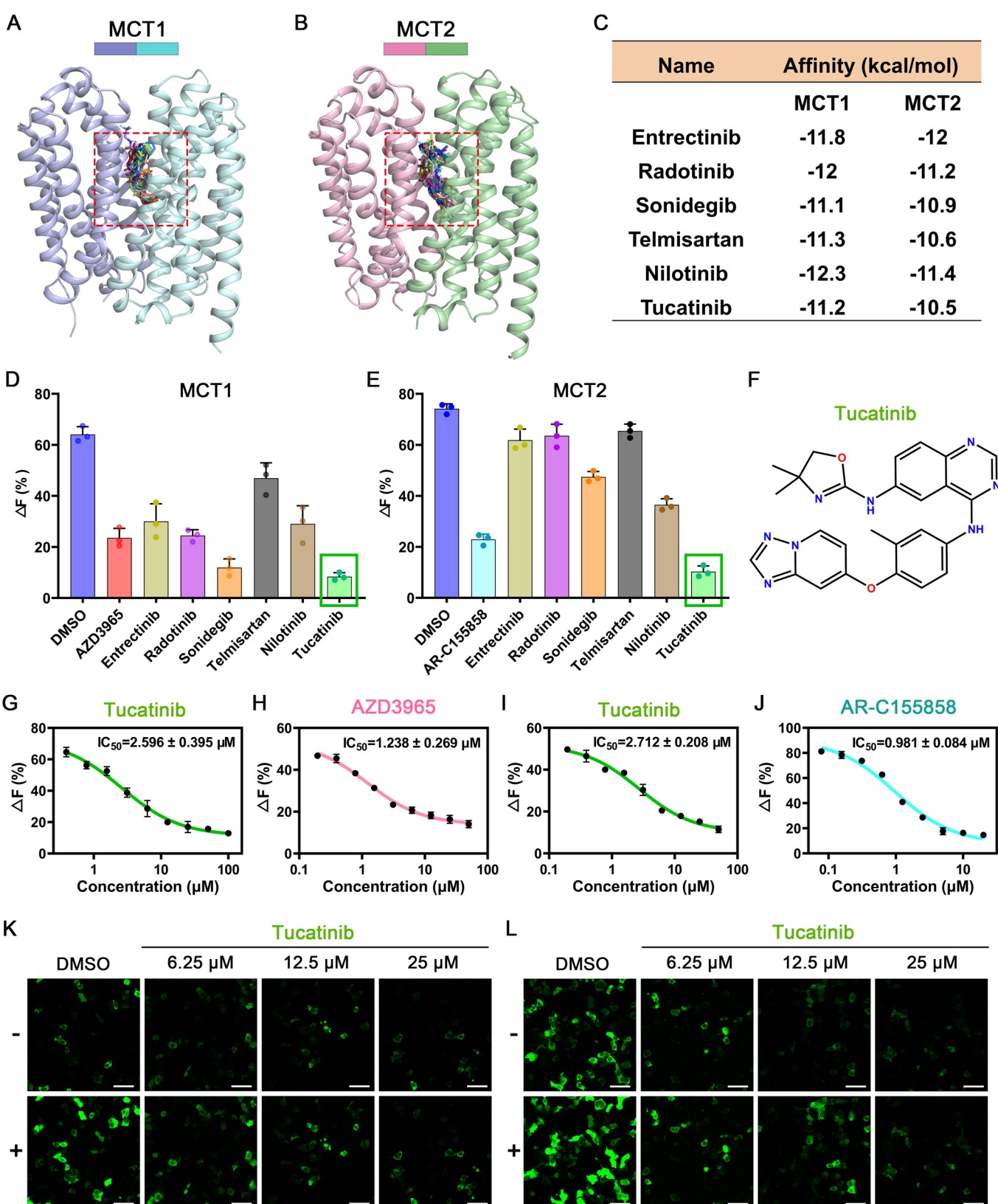

**Figure 3.   Tucatinib is a dual inhibitor of MCT1 and MCT2.**

Drug screening based on centralized pockets of MCT1 (A), MCT2 (B). The binding pocket, outlined by a dashed red box, represents the site used for small-molecule screening based on the MCT1A (PDB ID: 6LYY) and MCT2R-emb (PDB ID: 9LOC) structure, respectively. (C) Virtual screening based on a marketed drug library. Inhibitory effect on pyruvate transport of candidates (5 µM) on HEK-293T cells of MCT1 (D), MCT2 (E). $n = 3$ biological replicates. (F) Chemical structural formula of Tucatinib. $IC_{50}$ of Tucatinib (G), AZD3965 (H) on MCT1-mediated pyruvate transport; $IC_{50}$ of Tucatinib (I), AR-C155858 (J) on MCT2-mediated pyruvate transport. $n = 3$ biological replicates. Concentration-dependent inhibition of pyruvate transport treated with Tucatinib. HEK-293T cells were transfected with pyronicSF and MCT2-embigin (K) or with pyronicSF and MCT1-Basigin-2 (L), with or without treatment by various concentrations of Tucatinib (6.25−25 µM) for 24 h. The GFP fluorescence signals were used to monitor the amount of pyruvate transported. -: buffer treated; +: 5 mM pyruvate treated. Scale bar, 50 µm. The graphical presentation and data analysis were conducted using GraphPad Prism 9. The data were displayed as mean ± standard deviation (SD). Source data are available online for this figure.

comparison, migration inhibition experiments with AR-C155858 (Fig. 6C,D) and AZD3965 (Appendix Fig. S10C,D) showed that both inhibitors could inhibit the migration of HeLa cells, but not as significantly as Tucatinib, which almost completely restrained HeLa cell migration at a high concentration-25 µM. Moreover, shRNA-mediated knockdown of MCT1 or MCT2 significantly hindered HeLa cell migration, and subsequent addition of 12.5 µM Tucatinib elicited a certain degree of inhibition, but to a lesser extent compared to the wild-type condition, thereby validating MCT1 and MCT2 both as the primary targets of Tucatinib's migration-inhibitory effect in HeLa cells (Fig. 6E–H).

In summary, these findings illustrate the mechanism by which Tucatinib regulates HeLa cell proliferation and migration through targeting inhibition of both MCT1 and MCT2, and the effects are superior to those of AR-C155858 or AZD3965, indicating that Tucatinib exhibits high potential in the treatment of cervical cancer.

### Tucatinib inhibits cervical carcinoma growth in vivo

To assess the vivo efficacy of Tucatinib in suppressing cervical carcinoma growth, we employed a tumor xenograft model in female BALB/c nude mice (5−6 weeks of age). HeLa cells ($5 \times 10^6$) suspended in serum-free DMEM medium were injected into the right flank region of each mouse. Following tumor establishment, mice were treated with cisplatin or Tucatinib for 21 days, as outlined in Fig. 7A. The curve of tumor volume over time demonstrated a clear trend: the control group experienced a continual increase in tumor size, whereas the growth of tumors in treatment groups was suppressed to varying extents. At a low dose of 50 mg/kg, Tucatinib had a minimal inhibitory effect on tumor growth, reducing tumor volume by only 13%. However, at a higher dose of 100 mg/kg, Tucatinib's tumor-suppressing effect (57%) was comparable to that of cisplatin (54%) (Fig. 7B). Neither Tucatinib nor cisplatin significantly affected the body weights of the mice (Fig. 7C). After completion of the treatments, mice were euthanized, and tumors were excised (Fig. 7D). The results of tumor weight were consistent with those of tumor volume, with the 100 mg/kg dose of Tucatinib achieving a similar tumor inhibition rate to cisplatin (Fig. 7E). To further investigate the pharmacological mechanism of Tucatinib in vivo, frozen sections and H&E staining were performed on tumor tissues and organs. Pathological analysis indicated that Tucatinib caused minimal harm to the heart, liver, spleen, lung, and kidney of the treated mice. Notably, tumor tissues of all drug groups exhibited varying degrees of cellular shrinkage and deformation, along with nuclear lysis, which may

account for the observed tumor inhibition (Fig. 7F). In conclusion, Tucatinib, a novel dual MCT1/2 inhibitor with minimal cytotoxicity, is a potent anti-cervical cancer drug candidate suitable for clinical studies.

## Discussion

In our previous work, we observed that human MCT2 cooperatively transports pyruvate and demonstrated that such a cooperative transport arises from the strong inter-subunit cooperativity within the MCT2 dimer (Zhang et al, 2020). This was further confirmed in our subsequent studies on the structure and function of MCT1-embigin complex, where we identified distinct monocarboxylate-binding pockets and substrate-binding affinities between MCT homodimer and heterodimer (Xu et al, 2022).

This article extends the aforementioned studies by presenting cryo-EM structures of the human MCT2-embigin complex in apo and AR-C155858-bound states. Cell colocalization and substrate transport assays indicate that embigin facilitates proper trafficking and function of MCT2 at the plasma membrane. Given the structural similarity between MCT2-embigin heterodimer and the MCT1-embigin structures, the differences observed between MCT1 heterodimer and MCT2 homodimer (Xu et al, 2022) are also present between MCT2-embigin and MCT2 alone. Therefore, the coupling between two signature motifs (TM1, TM4-TM5 loop), one from each subunit, observed in MCT2 dimer (Zhang et al, 2020) and the conformational changes of the two signature motifs after the decoupling observed in MCT2-embigin complex further convoy the hypothesis that substrate-induced motion originating in one subunit could be transmitted from one subunit to the adjacent subunit to alter its ligand-binding affinity (Xu et al, 2022).

Regarding activity, both homodimeric and heterodimeric states are believed to be transport-competent. Both the homodimeric MCT2 (Zhang et al, 2020) and the heterodimeric MCT2-embigin complex (Fig. 2I, this study) can be inhibited by AR-C155858, effectively blocking their transport activity. However, the efficacy of inhibition may differ between these states, which represents one of the key directions for our future research. Its mechanism likely involves stabilizing a specific conformational state rather than exclusively "locking" the transporter in a heterodimeric form. Beyond the possibility of differential binding mechanisms, we also acknowledge that the inability to resolve the homodimeric MCT2-AR-C155858 complex could be attributed to several technical factors, such as sample heterogeneity, suboptimal vitrification conditions, or limitations in current reconstruction algorithms.

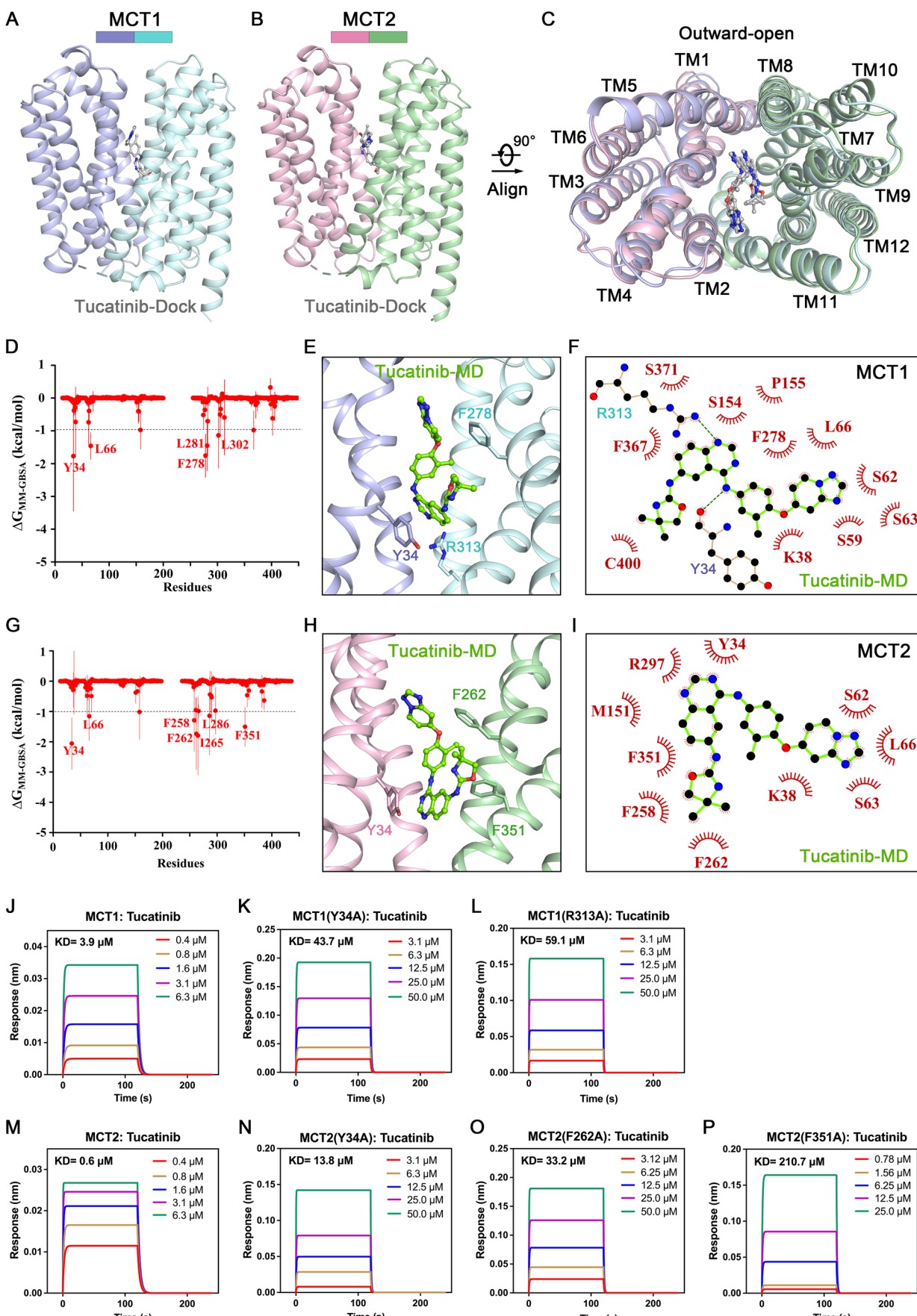

**Figure 4. Binding mode of Tucatinib with MCT1 or MCT2.**

(A–C) Molecular docking results of Tucatinib with MCT1 and MCT2. Free energy decomposition maps of Tucatinib with MCT1 (D), MCT2 (G). Binding modes of Tucatinib and MCT1 (E) and MCT2 (H). 2D interactions mode of Fig. (E, F) and (H, I). (J–P) Bio-layer interferometry (BLI) measurement of the binding affinities between MCT1 or MCT2 variants and Tucatinib. The binding kinetics of Tucatinib bound to MCT1-basigin-2 (J), MCT1(Y34A)-basigin-2 (K), MCT1(R313A)-basigin-2 (L); The binding kinetics of Tucatinib bound to MCT2-embigin (M), MCT2(Y34A)-embigin (N), MCT2(F262A)-embigin (O), MCT2(F351A)-embigin (P). Source data are available online for this figure.

Within the SLC16 gene family, MCT1 and MCT4 are the most extensively studied in cancers due to their overexpression in various tumor types, ranging from solid tumors to hematological malignancies (Singh et al, 2023). MCT2, less studied than MCT1, also plays a significant role in cancer development. A recent lung cancer study found that higher MCT2 expression correlates with elevated cell senescence, implying a potential beneficial effect of increased MCT2 activity on disease progression (Giatromanolaki et al, 2020). Valenca et al demonstrated that elevated MCT2 expression promotes proliferation in prostate cancer cells (Valenca et al, 2020). Moreover, MCT2 is strongly expressed in the cytoplasm of colorectal cancer cells (Koukourakis et al, 2006), and is the primary isoform in human glioblastoma multiforme and glioma-derived cell lines (Mathupala et al, 2004). Therefore, MCT2 could also serve as a promising target and biomarker for various cancers (Felmlee et al, 2020b; Halestrap and Wilson, 2012; Pinheiro et al, 2012). Given the remarkable structural conservation among MCTs, we conducted virtual screening based on the cryo-EM structures of MCT1-Basigin-2 (PDB ID: 6LYY) and MCT2-embigin (PDB ID: 9L0B) to identify novel MCT1/2 inhibitors. The candidate compound, Tucatinib, demonstrates the most pronounced inhibitory effect targeting both MCT1 and MCT2.

Tucatinib is an oral, small-molecule, selective HER2 kinase inhibitor initially developed by Array BioPharma (a Pfizer subsidiary) and subsequently developed by Seattle Genetics for the treatment of HER2-positive breast cancer and colorectal cancer (Lee, 2020). On April 17, 2020, Tucatinib was approved by the US FDA in combination with trastuzumab and capecitabine for the treatment of patients with advanced unresectable or metastatic HER2-positive breast cancer (Shah et al, 2021). In addition, Tucatinib was approved in Switzerland in May 2020, and is under regulatory review in Australia, Canada and Singapore under Project Orbis (Lee, 2020). On January 19, 2023, the FDA granted accelerated approval to Tucatinib in combination with trastuzumab for the treatment of adult patients with RAS wild-type HER2+ unresectable or metastatic colorectal cancer (Casak et al, 2023). However, there is no evidence to reveal the inhibitory activity of Tucatinib on cervical cancer, nor any clinical trials. As a transmembrane tyrosine kinase receptor, HER2 is overexpressed in ~20% of breast cancers and in subsets of gastric, colorectal, and esophageal cancers (Kulukian et al, 2020), but not in cervical cancer (Jia et al, 2023). In our study, both qPCR and Western Blot results indicate high expression of MCT1 and MCT2 in HeLa cells. Therefore, HER2-deficient HeLa tumor grafts serve as an ideal model to investigate Tucatinib's targeting of MCTs. Furthermore, our experimental results indicate that Tucatinib effectively inhibits the substrate transport function of MCT in an acidic environment (Fig. EV4), demonstrating its potential to exert pharmacological activity within the acidic tumor microenvironment. All assays employed on cells and in vivo consistently demonstrate that Tucatinib exhibits potent inhibitory effects on the proliferation and metastasis of cervical cancer cells, indicating that Tucatinib could exhibit therapeutic potential not only in HER2+ cancers but also in HER2−/MCT+ cancers by targeting MCT1/2.

While our data demonstrate that Tucatinib is a dual inhibitor of MCT1/2 transport activity and cell proliferation, we acknowledge the complexities inherent in genetically validating its mechanism of action in HeLa cells. The profound reliance of these cells on MCT function under standard culture conditions was initially unexpected but demonstrates the critical role of lactate flux homeostasis in maintaining their glycolytic phenotype, evidenced by the strong viability defect upon single isoform knockdown (Halestrap, 2012; Payen et al, 2020). This baseline dependency introduced a challenge for classical genetic rescue experiments, as the metabolic rewiring caused by sustained knockdown may not be fully reversible upon transient re-expression. Consequently, while re-expression of wild-type MCTs successfully restored sensitivity to Tucatinib, it did not fully rescue the growth defect (Appendix Fig. S11A,B). We therefore emphasize that the most critical evidence for on-target activity comes not from the rescue of proliferation, but from the specific restoration of drug sensitivity. The fact that Tucatinib regained efficacy only in cells reconstituted with functional, drug-binding-competent MCTs and not in those expressing binding-deficient mutants provides a rigorous genetic-pharmacological nexus that strongly argues for target specificity (Appendix Fig. S11A,B). This approach, while complex, is designed to test causality directly: the biological effect of Tucatinib is absolutely dependent on its physical interaction with specific residues on MCT1 or MCT2. Finally, the potent antitumor efficacy observed in HeLa xenografts is consistent with the proposed in vitro mechanism. Although in vivo models cannot recapitulate all genetic validation controls, the concordance between effects in both settings supports the conclusion that Tucatinib exerts its therapeutic effect through inhibition of MCT function in vivo. Collectively, our integrated pharmacological, genetic, and in vivo data provide a compelling case that Tucatinib is a promising dual MCT1/2 inhibitor whose biological effects are primarily mediated through on-target engagement.

Taken together, the structural studies of MCT2-embigin in apo and AR-C155858-bound states further deepened our understanding of the substrate recognition and transport mechanisms within the MCT family. The favorable therapeutic outcomes of Tucatinib in cervical cancer highlight the efficacy of structure-based screening for MCT-targeted inhibitors as a sophisticated approach to identifying potential anti-cancer active molecules.

					

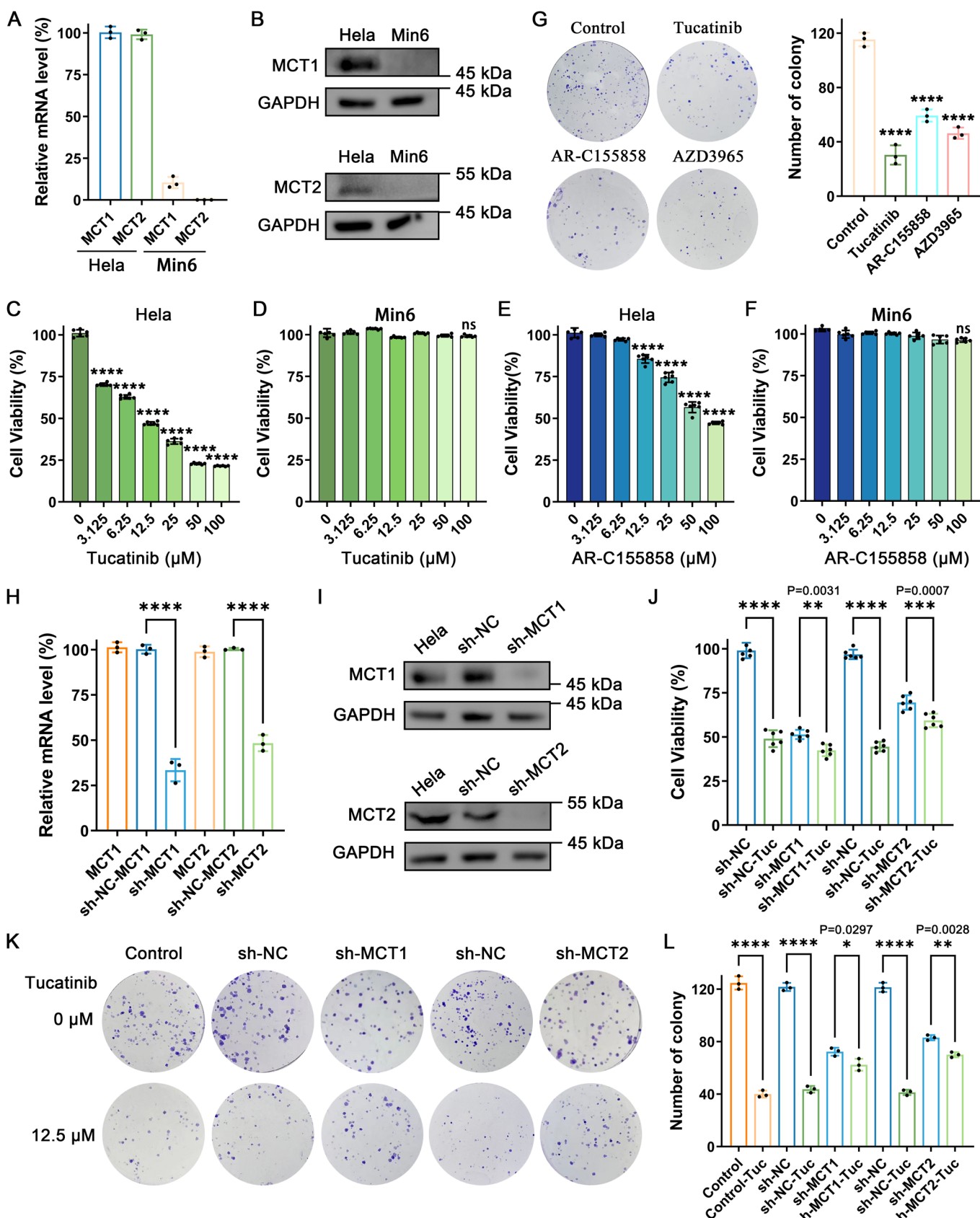

◀ **Figure 5. Tucatinib inhibits the proliferation of HeLa cells by targeting MCT1/2.**

(A) Relative mRNA level of MCT1 and MCT2 in HeLa and Min6 cells. $n = 3$ biological replicates. (B) Western blot images of MCT1 and MCT2 expression in HeLa and Min6 cells. Inhibitory effect of Tucatinib on the proliferation of HeLa (C), Min6 (D). $n = 6$ biological replicates. Inhibitory effect of AR-C155858 on the proliferation of HeLa (E), Min6 (F). $n = 6$ biological replicates. (G) Long-term colony formation assay was performed for 2 weeks on HeLa cells (purple), cultured with Tucatinib (12.5 μM), AR-C155858 (25 μM), and AZD3965 (25 μM), respectively. The bar chart on the right shows the statistical results. $n = 3$ biological replicates. (H) Relative mRNA level of the endogenous MCT1 or MCT2 knocked down by shRNA in HeLa cells using qPCR. $n = 3$ biological replicates. (I) The expression of the endogenous MCT1 or MCT2 was knocked down by shRNA in HeLa cells using Western blot. (J) The cell viability of HeLa cells after knocking down the endogenous MCT1 or MCT2 was treated with Tucatinib. $n = 6$ biological replicates. (K) The colony formation abilities of HeLla cells after knocking down the endogenous MCT1 or MCT2 were treated with Tucatinib. (L) Statistical results of (K). $n = 3$ biological replicates. sh-NC Negative control vector knock down HeLa cells, sh-MCT1 MCT1 knock down HeLa cells, sh-MCT2 MCT2 knock down HeLa cells, Tuc Tucatinib. The graphical presentation and data analysis were conducted using GraphPad Prism 9. The data are displayed as mean ± standard deviation (SD). Statistical significance of the differences between group means was evaluated by one-way analysis of variance (ANOVA) using the Tukey honestly significant difference test as a post hoc test; $p$ values ≤0.05 were considered statistically significant (*$p < 0.05$; **$p < 0.01$; ***$p < 0.001$; ****$p < 0.0001$, ns not significant). Source data are available online for this figure.

# Methods

### Reagents and tools table

| Reagent/resource | Reference or source | Identifier or catalog number |
|---|---|---|
| **Experimental models** | | |
| Sf9 insect cells | ATCC | CRL-1711 |
| HEK-293F | Thermo Fisher Scientific | R79007 |
| HEK-293T | Invitrogen | CRL-3216 |
| HeLa cells | Invitrogen | R714-07 |
| Min6 cells | MeisenCTCC | SCC623 |
| **Recombinant DNA** | | |
| pEG BacMam empty vector | Addgene | 160451 |
| MCT2-embigin cloned into pEG BacMam vector with twin-strep tag (pEG BacMam-MCT2-embigin-Strep) | This paper | N/A |
| MCT2 cloned into pEG BacMam vector with 10×His tag (pEG BacMam-MCT2-His) | This paper | N/A |
| pEG BacMam-MCT2 (R86A)-His | This paper | N/A |
| pEG BacMam-MCT2 (R86E)-His | This paper | N/A |
| pEG BacMam-MCT2 (K175A)-His | This paper | N/A |
| pEG BacMam-MCT2 (K175D)-His | This paper | N/A |
| pEG BacMam-MCT2 (N187A)-His | This paper | N/A |
| pEG BacMam-MCT2 (N187E)-His | This paper | N/A |
| pEG BacMam-MCT2 (F172D)-His | This paper | N/A |
| pEG BacMam-MCT2 (G176D)-His | This paper | N/A |
| pEG BacMam-MCT2 (I180D)-His | This paper | N/A |

| Reagent/resource | Reference or source | Identifier or catalog number |
|---|---|---|
| pEG BacMam-MCT2 (S183D)-His | This paper | N/A |
| pEG BacMam-MCT2 (L186D)-His | This paper | N/A |
| pEG BacMam-MCT2 (V190D)-His | This paper | N/A |
| pEG BacMam-MCT2 (S183D)-His | This paper | N/A |
| pEG BacMam-MCT2 (L194D)-His | This paper | N/A |
| Embigin cloned into pEG BacMam vector with twin-strep tag (pEG BacMam-embigin-Strep) | | |
| pEG BacMam-embigin (Y171A)-Strep | This paper | N/A |
| pEG BacMam-embigin (Y171R)-Strep | This paper | N/A |
| pEG BacMam-embigin (D174A)-Strep | This paper | N/A |
| pEG BacMam-embigin (D174K)-Strep | This paper | N/A |
| pEG BacMam-embigin (Y257A)-Strep | This paper | N/A |
| pEG BacMam-embigin (Y257R)-Strep | This paper | N/A |
| pEG BacMam-embigin (E270A)-Strep | This paper | N/A |
| pEG BacMam-embigin (E270R)-Strep | This paper | N/A |
| pEG BacMam-embigin (E282A)-Strep | This paper | N/A |
| pEG BacMam-embigin (E282R)-Strep | This paper | N/A |
| pEG BacMam-embigin (I169D)-Strep | This paper | N/A |
| pEG BacMam-embigin (P260D)-Strep | This paper | N/A |
| pEG BacMam-embigin (F264D)-Strep | This paper | N/A |

| Reagent/resource | Reference or source | Identifier or catalog number |
|---|---|---|
| pEG BacMam-embigin (I267D)-Strep | This paper | N/A |
| pEG BacMam-embigin (V268D)-Strep | This paper | N/A |
| pEG BacMam-embigin (V271D)-Strep | This paper | N/A |
| pEG BacMam-embigin (L274D)-Strep | This paper | N/A |
| pEG BacMam-embigin (V275D)-Strep | This paper | N/A |
| pEG BacMam-embigin (I278D)-Strep | This paper | N/A |
| pEG BacMam-embigin (L279D)-Strep | This paper | N/A |
| pcDNA3.1 empty vector | addgene | 128034 |
| pcDNA3.1-MCT2-GFP | This paper | N/A |
| pcDNA3.1-embigin-RFP | This paper | N/A |
| pcDNA3.1-MCT2-embigin-GFP | This paper | N/A |
| **Antibodies** | | |
| Anti-His | TransGen Biotech | HT501-01 |
| Anti-Strep | Abcam | ab180957 |
| Anti MCT1 | Abcam | Ab314172 |
| Anti MCT2 | Proteintech | 85145-3-RR |
| Anti GAPDH | Abcam | ab8245 |
| **Bacterial and virus strains** | | |
| E. coli DH5a | TIANGEN | CB101-02 |
| E. coli DH10Bac | Invitrogen | 10361012 |
| **Oligonucleotides and other sequence-based reagents** | | |
| PCR primers | This paper | Appendix Table S2 |
| **Chemicals, enzymes and other reagents** | | |
| SIM SF medium | Sino Biological | MSF1 |
| Fetal bovine serum (FBS) | GIBCO | 10091148 |
| Penicillin-Streptomycin (10,000 U/mL) | GIBCO | 15140163 |
| RPMI Medium 1640 | GIBCO | C11875500BT |
| Opti-MEM | GIBCO | 51985-034 |
| Lipofectamine 3000 transfection reagent | Life Technologies | L3000015 |
| Cellfectin™ II Reagent | Thermo Fisher Scientific | 10362100 |
| leupeptin | Solarbio | L8110 |
| pepstatin A | Solarbio | P8100 |
| benzamidine | Solarbio | B8070 |
| aprotinin | BBI Life Sciences | A600153-0250 |
| Phenylmethanesulfonyl fluoride (PMSF) | BBI Life Sciences | 329-98-6 |
| n-Dodecyl-β-D-maltopyranoside (DDM) | Anatrace | D310 |

| Reagent/resource | Reference or source | Identifier or catalog number |
|---|---|---|
| Synthetic drop-in substitute for Digitonin • glyco-diosgenin (GDN) | | GDN101 |
| Imidazole | Sangon Biotech | A500529-0001 |
| d-Desthiobiotin | Sigma | D1411 |
| $Ni^{2+}$-nitrilotriacetate affinity resin (Ni-NTA) | GenScript | L00250 |
| Strep-Tactin Sepharose resin | Smart-Lifesciences | SA053025 |
| Polyvinylidene difluoride (PVDF) membrane | Millipore | ISEQ00010 |
| **Software** | | |
| PyMOL | The PyMOL Molecular Graphics System, Schrödinger, Inc. | https://pymol.org/2/ |
| UCSF Chimera | Pettersen et al, 2004 | http://www.cgl.ucsf.edu/chimera |
| COOT | Emsley et al, 2010 | https://www2.mrc-lmb.cam.ac.uk/personal/pemsley/coot/ |
| PHENIX | Adams et al, 2010 | http://www.phenix-online.org/ |
| CryoSPARC | Punjani et al, 2017 | https://www.cryosparc.com/ |
| MotionCor2 | Zheng et al, 2017 | https://emcore.ucsf.edu/ucsf-motioncor2 |
| Gctf | | https://www2.mrc-lmb.cam.ac.uk/research/locally-developed-software/zhang-software/#gctf |
| GraphPad Prism 7.5.1 | | https://www.graphpad.com |
| ImageJ | | https://imagej.nih.gov/ij/index.ht |
| **Other** | | |
| Grids: R1.2/1.3 Au 300 mesh | Quantifoil | Q17080 |
| Superose 6 Increase 10/300 GL column | GE Healthcare | 29-0915-96 |

## Cell culture

All *E. coli* cells were cultured in LB medium at 37 °C. The *E. coli* strain DH5α was used to generate and amplify plasmids for MCT2-embigin, MCT1-Basigin-2, MCT1/2 variants, embigin variants, and strain DH10Bac cells were used to generate bacmids.

Sf9 cells were cultured in SIM SF medium (Sino Biological) at 28 °C. Bacmids transfection was carried out using Cellfectin II Reagent (Thermo Fisher Scientific) according to the manufacturer's instructions. The baculoviruses were generated and amplified in Sf9 insect cells in SIM SF medium. HEK-293F cells were cultured in

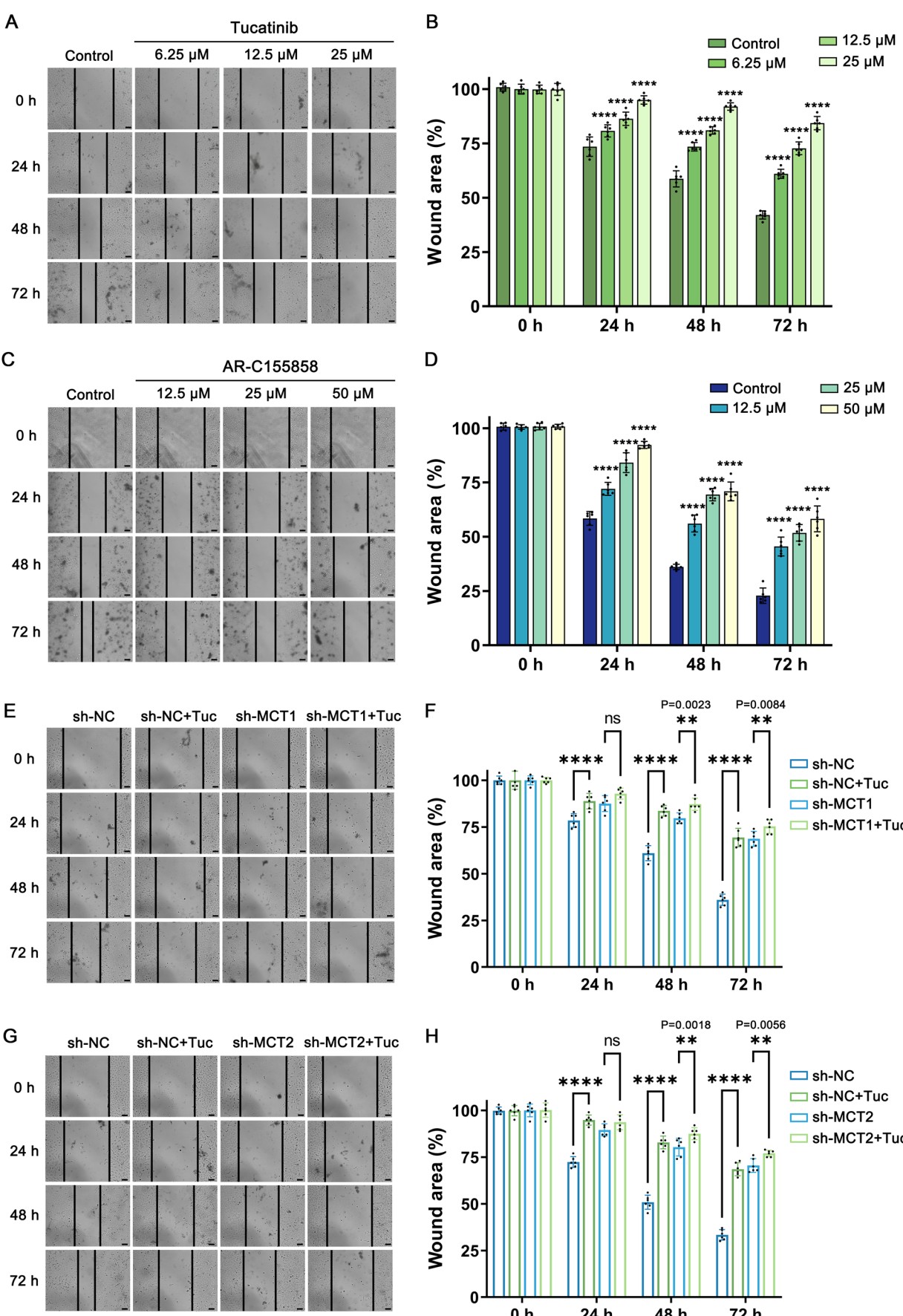

Figure 6. Tucatinib inhibits the migration of HeLa cells by targeting MCT1/2.

(A) Inhibitory effect of the migration of HeLa cells with different concentrations of Tucatinib. (B) Statistical results of (A). $n = 6$ biological replicates. (C) Inhibitory effect of the migration of HeLa cells with different concentrations of AR-C155858. (D) Statistical results of (C). $n = 6$ biological replicates. (E) The migration inhibition effects of HeLa cells after knocking down the endogenous MCT1 treated with Tucatinib. (F) Statistical results of (E). $n = 6$ biological replicates. (G) The migration inhibition effects of HeLa cells after knocking down the endogenous MCT2 treated with Tucatinib. (H) Statistical results of (G). $n = 6$ biological replicates. Scale bar, 100 μm. sh-NC negative control vector knock down HeLa cells, sh-MCT1 MCT1 knock down HeLa cells, sh-MCT2 MCT2 knock down HeLa cells, Tuc Tucatinib. The graphical presentation and data analysis were conducted using GraphPad Prism 9. The data were displayed as mean ± standard deviation (SD). Statistical significance of the differences between group means was evaluated by one-way analysis of variance (ANOVA) using the Tukey honestly significant difference test as a post hoc test; $p$ values ≤0.05 were considered statistically significant (*$p < 0.05$; **$p < 0.01$; ***$p < 0.001$; ****$p < 0.0001$, ns not significant). Source data are available online for this figure.

SMM 293-TI medium (Sino Biological) at 37 °C, 5% $CO_2$ and were used for protein sample expression.

HeLa and HEK-293T cells were cultured in RPMI Medium 1640 (GIBCO) at 37 °C, 5% $CO_2$. The plasmids were generated and transfected into HeLa cells in Opti-MEM (GIBCO). Plasmid transfection were carried out using Lipofectamine 3000 transfection reagent (Life Technologies) according to the manufacturer's instructions.

Min6 cells (MeisenCTCC) were cultured in RPMI Medium 1640 (GIBCO) at 37 °C, 5% $CO_2$.

All cell lines used in this study were routinely tested for mycoplasma contamination using a PCR-based method and were confirmed to be negative.

## Constructs, protein expression, and purification

Fusion construct human MCT2-embigin were generated by linking the C-terminus of embigin (NCBI accession NM_198449.3) to the N-terminus of MCT2 (NCBI accession NM_001270622) using a flexible GGSG linker. The fusion construct was cloned into a pEG-BacMam vector with a C-terminal twin-Strep tag for protein overexpression and purification.

For protein expression and purification, MCT2-embigin was heterologously expressed in HEK-293F cells (Thermo Fisher Scientific; R79007) using the BacMam system (Thermo Fisher Scientific). The baculovirus was generated in Sf9 cells (ATCC; CRL-1711) following standard protocol and was used to infect HEK-293F cells at MOI = 20, supplemented with 10 mM sodium butyrate to boost protein expression. Cells were cultured in suspension at 37 °C, 5% $CO_2$ for 48 h and harvested by centrifugation at 4000 rpm. The cell pellet was lysed by sonication on ice in buffer A (25 mM Tris-HCl, pH 8.0, 150 mM NaCl) supplemented with a protease inhibitor cocktail (1 μg/mL leupeptin, 1 μg/mL pepstatin A, 1 μg/mL benzamidine, 1 μg/mL aprotinin, and 1 mM PMSF). MCT2-embigin was extracted with 2% (w:v) $n$-Dodecyl-β-D-maltopyranoside (DDM, Anatrace) by gentle agitation for 2 h at 4 °C. After extraction, the supernatant was collected following a 40-min centrifugation at 38,000 rpm and incubated with Streptactin Beads 4FF (Smart-Lifesciences) with gentle agitation at 4 °C. After 1 h, the resin was collected on a disposable gravity column (Bio-Rad), washed in buffer B (buffer A + 0.02% GDN) for 20 column volumes and was then eluted with 10 mM D-Desthiobiotin. The protein sample was further purified by size exclusion chromatography on a Superose 6 10/300 GL column (GE Healthcare) pre-equilibrated with buffer B. The protein peak fraction was collected and concentrated to 10.0 mg/ml for cryo-electron microscopy analysis.

## Cryo-EM data collection

For cryo-EM sample preparation, the purified MCT2-embigin protein with or without AR-C155858 at 10 mg/ml was applied to a glow-discharged holey carbon grid (Quantifoil Au R1.2/1.3, 300 mesh), blotted under 100% humidity at 8 °C and plunged into liquid ethane using a Mark IV Vitrobot (FEI). Micrographs were acquired on a Titan Krios electron microscope (FEI) operating at 300 kV, equipped with the GIF-Quantum energy filter and a K3 Summit direct electron detector (Gatan). SerialEM software (FEI) was used for automated data collection following the standard FEI procedure. Images were recorded at a normal magnification of 105,000×, corresponding to a pixel size of 0.839 Å for MCT2-emb and 0.849 Å for MCT2R-emb per pixel and with a set defocus range of 1.5 to 2.0 μm. Each micrograph was dose-fractionated to 32 frames recorded every 0.08 s under a dose rate of 6.66 e-/pixel/s, resulting in an accumulated dose of ~60 e-/Å².

## Image processing and model building

The flowcharts for the MCT2-emb, MCT2R-emb data processing using the cryoSPARC suite are presented in Appendix Figs. S1 and S3, respectively. Patch CTF estimation was carried out after alignment and summary of all 32 frames in each stack using the patch motion correction (Zheng et al, 2017). Initial particles were picked from a few micrographs using blob picker in cryoSPARC, and 2D averages were generated (Punjani et al, 2017). Final particle picking was done by template picker using templates from those 2D results. After three rounds of 2D classification, ab-initio reconstruction, nonuniform refinement and local refinement for reconstructing the density map. All maps were low-pass filtered to the map-model FSC value. The reported resolutions were based on the FSC = 0.143 criterion. An initial model was generated by MCT1-embigin (PDB: 7YR5). Then, we manually completed and refined the model using Coot (Emsley et al, 2010). Subsequently, the models were refined against the corresponding maps by PHENIX (Adams et al, 2010). The statistics for the models' geometries was generated using MolProbity (Chen et al, 2010) (Appendix Table S1). All the figures were prepared in PyMol and Chimera (Pettersen et al, 2004).

## Pull-down assay

MCT2 (wildtype or variants) with a 10×His tag and embigin (wildtype or variants) with a twin-Strep tag were co-expressed in HEK-293F cells and extracted with DDM according to previously described protocols. The supernatant of high-speed centrifugation

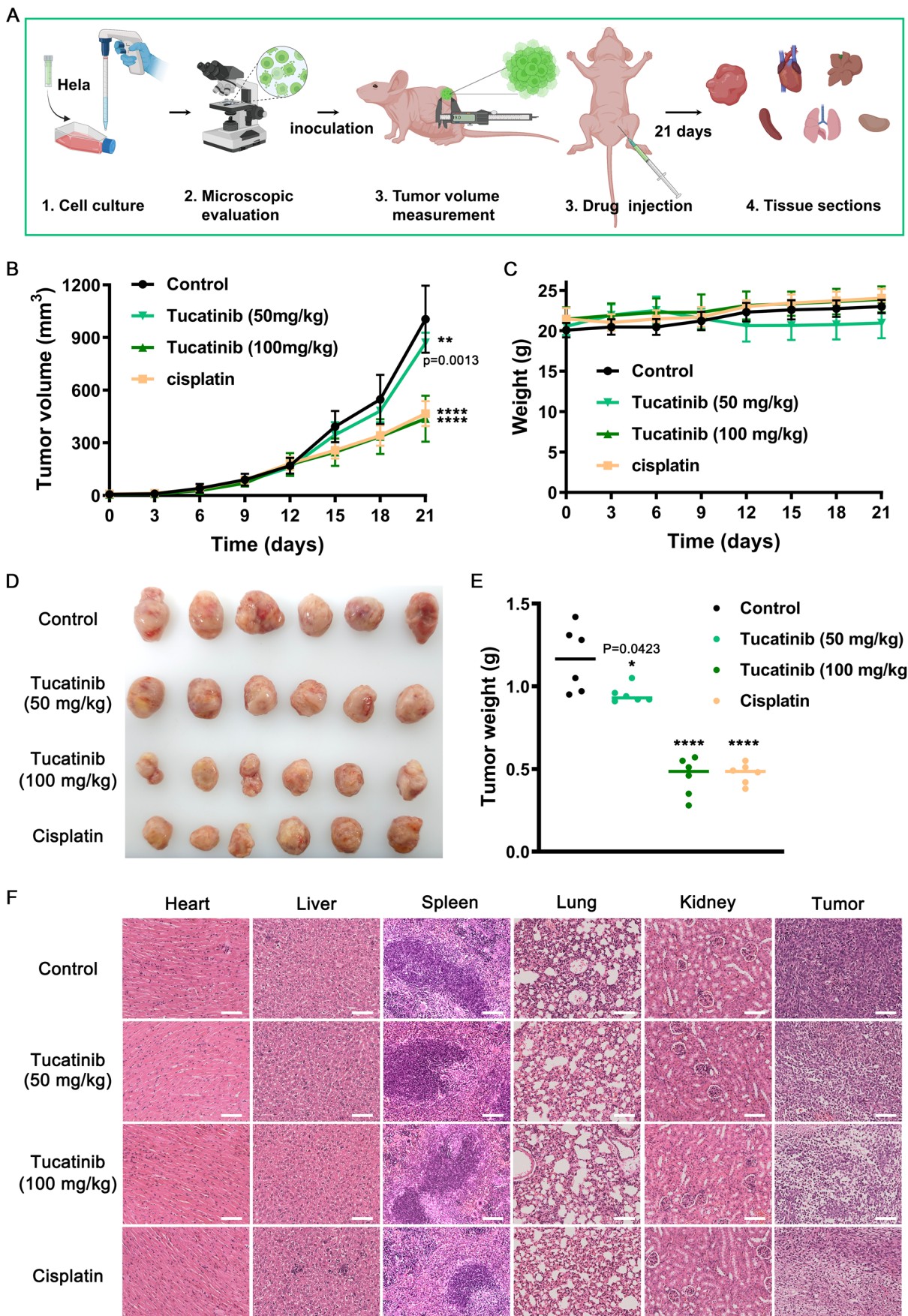

**Figure 7. Tucatinib inhibits the growth of cervical carcinoma in the xenografted model mice.**

(A) Schematic diagram of the experimental protocol. (B) Tumor volume growth in different groups. $n = 6$ biological replicates. (C) Body weight change curve in different groups. $n = 6$ biological replicates. (D) The stripped image of the tumor entity after 21 days of treatment with Tucatinib. (E) Statistical results of tumor tissue weight in (D). $n = 6$ biological replicates. (F) Tissue sections of major organs (heart, liver, spleen, lung, and kidney) and tumor tissues in different administration groups. Scale bar, 100 μm. The graphical presentation and data analysis were conducted using GraphPad Prism 9. The data were displayed as mean ± standard deviation (SD). Statistical significance of the differences between group means was evaluated by one-way analysis of variance (ANOVA) using the Tukey honestly significant difference test as a post hoc test; $p$ values ≤0.05 were considered statistically significant (*$p < 0.05$; **$p < 0.01$; ***$p < 0.001$; ****$p < 0.0001$, ns not significant). Source data are available online for this figure.

was prepared for two copies and applied to either the NI-NTA resin (His pull-down) or StrepTactin resin (Strep pull-down), respectively. Elution was applied to sodium dodecyl sulfate-polyacrylamide gel electrophoresis (SDS-PAGE), followed by blotting to polyvinylidene difluoride membrane (Millipore). MCT2 was labeled with mouse anti-His antibodies (TransGen Biotech). Embigin was labeled by rabbit anti-Strep antibodies (Abcam). Quantification of protein bands was visualized by the Chemiluminescence mode of Alliance Q9 (UVITEC Cambridge) using the NiceAlliance Q9 software.

## Confocal microscopy imaging analysis

The constructs of MCT2-GFP, embigin-RFP and fusion MCT2-embigin-GFP were cloned into pcDNA3.1 for cell surface localization analysis. HEK-293T or HeLa cells were transferred to a Confocal Dish (NEST) for 24 h prior to the experiment. These cells were transiently transfected with 2 μg plasmid DNA of MCT1, embigin, and fusion MCT2-embigin using Lipofectamine 3000 transfection reagent (Life Technologies) following the manufacturer's protocol, respectively. Twenty-four hours post transfection, cells were monitored using a laser scanning confocal microscope (Leica Stellaris 5, Japan).

## Virtual screening

The cryo-electron microscopy structure of MCT1A (PDB ID: 6LYY) and MCT2R-emb (PDB ID: 9L0C) was used for virtual screening. Marketed drugs (~2300 compounds) were screened using the molecular docking program Vina (Trott and Olson, 2010). A semi-flexible docking protocol was used to perform the calculations, and the size of the binding pocket search space was $20 \times 20 \times 20$ Å³. The global search exhaustiveness value was set to 50. The maximum energy difference between the optimal binding mode and the worst case was set to 5 kcal/mol.

## Pyruvate transport inhibition analysis

HEK-293T cells (ATCC; CRL-3216) were transferred to a 35 mm tissue culture dish (Corning) for 24 h prior to the experiment. These cells were then transiently transfected by Lipofectamine 3000 (Life Technologies) with 4 μg plasmid DNA of pyronicSF and fusion MCT2-embigin variants or fusion MCT1-Basigin-2 variants following the manufacturer's protocol, respectively. After 24 h, HEK-293T cells were collected and washed three times with PBS. They are then seeded into 96-well plates at a density of 50,000 cells per well. Following this, the cells are incubated with different concentrations of inhibitors for 30 min. After the incubation

period, 5 mM of pyruvate was added. The fluorescence intensities were read at λex = 488 nm and λem = 525 nm on a Microplate Reader (Tecan, Austria). The pyruvate transport activity was calculated as the ratio of the experimental group wells minus the blank group wells, divided by the blank group wells (ΔF% = (F-F0)/F0*100%).

For fluorescence detection, HEK-293T cells were transferred to a confocal dish (NEST) for 24 h prior to the experiment. These cells were then transiently transfected by Lipofectamine 3000 (Life Technologies) with 2 μg plasmid DNA of pyronicSF and fusion MCT2-embigin or fusion MCT1-Basigin-2 following the manufacturer's protocol, respectively. Twenty-four hours post transfection, the cells are incubated with different concentrations of Tucatinib for 30 min. After the incubation period, 5 mM of pyruvate was added. And cells were monitored using a laser scanning confocal microscope (Leica Stellaris 5, Japan).

## Affinity determination by biolayer interferometry (BLI)

Affinity assays were performed on an Octet® R8 biolayer interferometry instrument (Sartorius, Germany) at 25 °C with shaking at 1000 rpm. To measure the affinity of MCT1-Basigin-2 variants with Tucatinib and MCT2-embigin variants with Tucatinib or AR-C155858, Super Streptavidin (SSA) biosensors (Sartorius, Germany) were hydrated in water for 30 min prior to 60 s (sec) incubation in a kinetic buffer (0.1 M Hepes,1.5 M NaCl, 0.03 M EDTA, 0.5% (v/v) Surfactant P20, and pH 7.4). The purified protein was loaded in a kinetic buffer for 120 s prior to baseline equilibration for 120 s in a kinetic buffer. The data were baseline-subtracted before fitting was performed using a 1:1 binding model and the Octet® R8 data analysis software. KD, $K$a, and $K$d values were evaluated with a global fit applied to all data.

## MD simulations

The simulation system containing MCT, POPC, and ligands (AR-C155858, Tucatinib) was generated by CHARMM-GUI (box size 86.5 Å × 86.5 Å × 113 Å). MD simulations were performed using AMBER22 (2022). The Amber ff19SB force field, the lipid21 force field and the GAFF force field were applied to the protein, POPC bilayer and ligands, respectively. TIP3P water model and Cl- were added to solvate and neutralize the membrane-protein systems, and added Joung/Cheatham ion parameters (Joung and Cheatham, 2009). The solvated system totally contained ~75,000 atoms. The other parameters were the same as we set before (Shi et al, 2023). Finally, the time step for all MD simulations was set to 2 fs, and MD was performed for 500 ns with Cα constrained (1 kcal/mol/Å), with three replications for each system.

The MMGBSA (Molecular Mechanics combined with generalized born and surface area solvation) method (Genheden and Ryde, 2015) was applied to estimate the binding free energy between protein and small molecules. To achieve this aim, 100 snapshots were collected from the last 100 ns of the MD trajectory. The binding free energy was calculated by the formula:

$$\Delta G_{MMGBSA} = G_{compiex} - G_{receptor} - G_{ligand}$$

## Quantitative real-time polymerase chain reaction (qPCR)

The EasyPure RNA Kit (#O101011, Transgene) was used to extract total RNA. Total RNA was then quantified using a spectrophotometer at 260 nm. A Thermal cycler was used to make cDNA from the total RNA at 25 °C for 5 min, 42 °C for 30 min, and 85 °C for 5 min. The Cell Lysis RT-qPCR KITs was used for qPCR as follows: 1 μL diluted cDNA was mixed with 0.4 μL forward and 0.4 μL reverse primers and 10 μL SYBR Green and 3.2 μL RNase-free $H_2O$. The cycling conditions were as follows: 5 min for initial denaturation at 95 °C, followed by 40 cycles of 30 s at 95 °C, 40 s at 58 °C, 40 s at 72 °C, and 5 min at 72 °C. All reactions were performed in triplicate and normalized to GAPDH. The calculations were carried out using the CT values. Primer sequences were: MCT1 forward primer 5′-TTTCTTTGCGGCTTCCGTTG-3′ and reverse primer 5′-CTCTGGGGTCCAACAAGGTC-3′, MCT2 forward primer 5′- GACACGTCAGGGGCCATAAAT-3′ and reverse primer 5′- GGTTCCCCAGATCACCTTGT-3′, GAPDH forward primer 5′- CCCACTCCTCCACCTTTGACG-3′ and reverse primer 5′- CACCACCCTGTTGCTGTAGCCA-3′.

## Western blot assay

HeLa (WT, sh-NC, sh-MCT1, and sh-MCT2) or Min6 cells were washed three times with PBS (#P1022, Solarbio) and then lysed with RIPA buffer supplemented with protease inhibitors. The protein quantification was performed using the BCA protein assay kit (#PC0020, Solarbio). An equal amount of protein in each group was separated using SDS-PAGE gel electrophoresis and transferred onto polyvinylidene fluoride membranes. After being blocked with 5% non-fat milk for 1 h, the membranes were incubated overnight at 4 °C with primary antibodies (MCT1 antibody-Invitrogen, P14612; MCT2 antibody-Proteintech, 202355-1-AP). The membranes were then washed three times with tris-buffered saline + 0.1% Tween 20 (TBST), and incubated for 1 h with the corresponding secondary antibodies at 37 °C and washed with TBST. Protein bands were detected using a Bio-Rad imaging system (Bio-Rad, Hercules, CA, USA)

## CCK-8 assay

Cell viability was assessed using the Cell Counting Kit-8 (Solarbio, Beijing, China), a colorimetric assay that measures the metabolic activity of cells based on the reduction of a water-soluble tetrazolium salt (WST-8) to an orange-colored formazan product.

HeLa Cells were transduced with sh-NC or sh-MCT1 or sh-MCT2 at a multiplicity of infection of 10 for 48 h and were subjected to puromycin selection (1–2 ug/μl). After selection, cells were seeded into 96-well plates. In the gene rescue experiment, we reintroduced the plasmids of MCT1 or MCT2 back into the corresponding knockdown HeLa cell lines via transfection.

For the assessment of cell viability, HeLa (WT, gene knockdown type or gene rescue type) or Min6 cells were plated in 96-well plates and exposed to different drugs for 48 h. Following this incubation period, 10 μL of CCK-8 reagent was added to each well and allowed to react for 2–4 h. The absorbance at 450 nm was then read on a Microplate Reader (Tecan, Austria), and cell viability was calculated as the ratio of the absorbance in the test group wells to that in the untreated control group wells.

## Colony formation assay

Approximately 500 to 1000 HeLa cells (WT, sh-NC, sh-MCT1, sh-MCT2) were seeded onto a six-well plate and incubated for 24 h to allow for initial attachment and growth. Subsequently, these cells were exposed to various kinds of inhibitors (Tucatinib, AR-C155858 or AZD3965) in the culture medium for a period of 10 to 14 days. During this period, the culture medium was replaced twice weekly to maintain optimal growth conditions. After the specified treatment duration, the cells were fixed using 4% formaldehyde solution (Biosharp, BL539A) and subsequently stained with 0.1% crystal violet solution (Solarbio, #C8470) to visualize cell colonies. To assess the cloning efficiency of the cells treated with inhibitors, the average number of colonies formed in the treated wells was compared to that of the untreated control wells. The number of colonies was calculated using ImageJ.

## Wound-healing assay

HeLa cells (WT, sh-NC, sh-MCT1, and sh-MCT2) were seeded onto a six-well plate and allowed to proliferate until they reached approximately 90% confluence. Subsequently, a standardized wound was created across the cell monolayer using a sterile 200 μL pipette tip. The cells were then treated with various concentrations of inhibitors (Tucatinib, AR-C155858 or AZD3965, ranging from 12.5 to 50 μM) or an equivalent volume of DMSO as a control, both dissolved in medium containing 1% fetal bovine serum (FBS) to minimize proliferation-driven wound closure. At predetermined time intervals (0, 24, 48, and 72 h), images of the wound-healing process were captured using an inverted microscope. The ImageJ software was employed to quantify the wound area at each time point. The relative wound-healing percentage was calculated by comparing the average area of the newly formed tissue in the inhibitor-treated wells to the initial wound area at the start of the experiment. This metric provides a quantitative assessment of the cells' migratory and wound-healing capabilities in response to drug treatment.

## Tumor xenografts in mice

Animal experiments were done in accordance with the authorization of animal operation and in accordance with the Chinese law for animal protection. All animal experiments were approved by the Animal Ethical and Welfare of Tianjin University (Approval No. TJUE-2024-070). Mice were housed under standard conditions at a temperature of 20–26 °C, humidity of 40–70%, with free access to food and water. HeLa cells ($5 \times 10^6$) suspended in serum-free DMEM medium, were inoculated subcutaneously to female BALB/c

nude mice (5- to 6-week-old) in the right back near the armpit. Then the mice were randomly divided into four groups (six mice in each group): (1) control group, (2) Tucatinib (50 mg/kg) group, (3) Tucatinib (100 mg/kg) group, (4) Cisplatin (6 mg/kg) group. The length and width of the tumor were measured every 3 days, and the volume (mm$^3$) was calculated according to the formula: length × width$^2$/2. Drugs were administered every 3 days to mice via intraperitoneal injection. All the injected drugs were dissolved in 2% DMSO solution, and mice body weights were measured every 3 days, while control group mice were injected with 2% DMSO solution. Mice were injected seven times and measured eight times. All the mice were sacrificed after seven times medication. Then the heart, liver, spleen, lung, kidney, and tumor tissues of mice were sliced to observe the pathological status.

## Data analysis

The graphical presentation and data analysis were conducted using GraphPad Prism 9. The data are displayed as mean ± standard deviation (SD), and the number of replicates is provided in the figure legends. Statistical significance of the differences between group means was evaluated by one-way analysis of variance (ANOVA) using the Tukey honestly significant difference test as a post hoc test; $p$ values ≤0.05 were considered statistically significant ($*p < 0.05$; $**p < 0.01$; $***p < 0.001$; $****p < 0.0001$, ns not significant).

## Declaration of blinding

Blinding was not performed in this study.

## Data availability

Cryo-EM maps and coordinates have been deposited in the EMDB and wwPDB, respectively, with accession numbers: PDB 9L0B (http://www.rcsb.org/pdb/explore/explore.do?structureId=9L0B), EMD-62694 (MCT2-emb, http://www.ebi.ac.uk/pdbe/entry/EMD-62694); PDB 9L0C (http://www.rcsb.org/pdb/explore/explore.do?structureId=9L0C), EMD-62696 (MCT2R-emb, http://www.ebi.ac.uk/pdbe/entry/EMD-62696). Further information and requests for reagents may be directed and will be fulfilled by Sheng Ye (sye@tju.edu.cn), the lead contact.

The source data of this paper are collected in the following database record: biostudies:S-SCDT-10_1038-S44319-025-00661-9.

## Peer review information

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

## Acknowledgements

This work was supported in part by the Ministry of Science and Technology (2024YFA0916800 to BX, 2020YFA0908500 to SY, 2024YFC3407300 to YW), the National Natural Science Foundation of China (82504854 to BX and 82404583 to SS), and the Tianjin Natural Science Foundation (24JCQNJC01390 to BX). Cryo-EM data were collected at the Westlake University Biomedical Research Core Facility. We thank the Haihe Laboratory of Sustainable Chemical Transformations for financial support. We thank the Medical Science Data Center of Hebei Medical University.

## Author contributions

Binghong Xu: Conceptualization; Data curation; Funding acquisition; Validation; Methodology; Writing—original draft; Writing—review and editing. Xiaoyu Zhou: Data curation; Methodology; Project administration. Yuanyue Shan: Software; Methodology. Sai Shi: Resources; Software. Jiachen Li: Methodology; Project administration. Qinqin Liang: Methodology; Project administration. Ziyu Wang: Methodology; Project administration. Mingfeng Zhang: Resources; Software. Yaxin Wang: Funding acquisition; Validation; Methodology; Project administration. Duanqing Pei: Software; Supervision. Sheng Ye: Conceptualization; Funding acquisition; Validation; Project administration.

Source data underlying figure panels in this paper may have individual authorship assigned. Where available, figure panel/source data authorship is listed in the following database record: biostudies:S-SCDT-10_1038-S44319-025-00661-9.

## Disclosure and competing interests statement

The authors declare no competing interests.

# Expanded View Figures

---

**Figure EV1.  Structural comparison between human MCT2-embigin and MCT1-embigin, MCT2, respectively, related to Fig. 1.**

(**A**) Structural comparison of outward-open human MCT2-emb (PDB ID: 9LOB) and inward-open MCT1-emb (PDB ID: 7YR5). The NTD and CTD remain relatively unchanged during the alternating-access cycle. Structures of MCT2-emb and MCT1-emb are separately aligned relative to NTD or CTD. (**B**) Rotation of TM of representative TM segments between MCT2-emb and MCT1-emb relative to the membrane norm. (**C**) Structural alignments between MCT2-emb (blue, pink, and green) and MCT1-emb (gray). The N-domain was used as a reference for structural alignments. The two structures are superimposed with an RMSD of 1.04 Å over 184 aligned Cα atoms. Black arrows indicate the oscillations of the C-domain. Top: Lumen side view; Bottom: Cytoplasm side view. (**D**) Structure of the human MCT2. The cryo-EM structure of inward-open MCT2 (PDB code: 7BP3) viewed parallel to the plasma membrane (left), extracellularly (right). (**E**) Structural comparison of outward-open human MCT2-emb and one subunit of MCT2, which are separately aligned relative to NTD or CTD. (**F**) Structures of MCT2-emb and MCT2 are aligned to NTD from the intracellular side. TM regions of MCT2 are displayed as a cylinder. (**G**) Conformational changes of TM5 between human MCT2-emb and MCT2. TM5 is a relatively straight helix in MCT2-emb, but the cytosolic half of TM5 in MCT2 swings toward the center of the transporter by about 30 degrees.

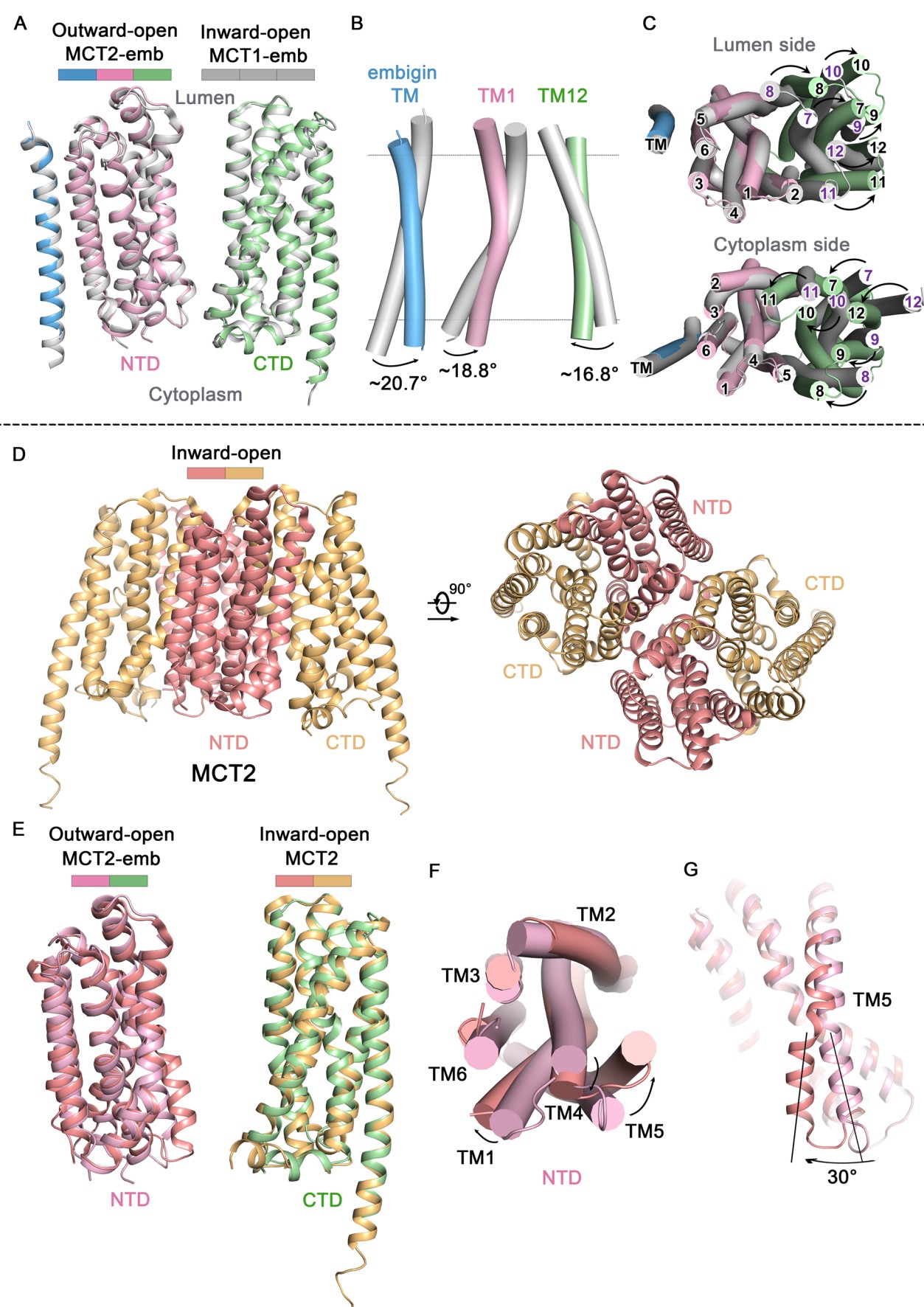

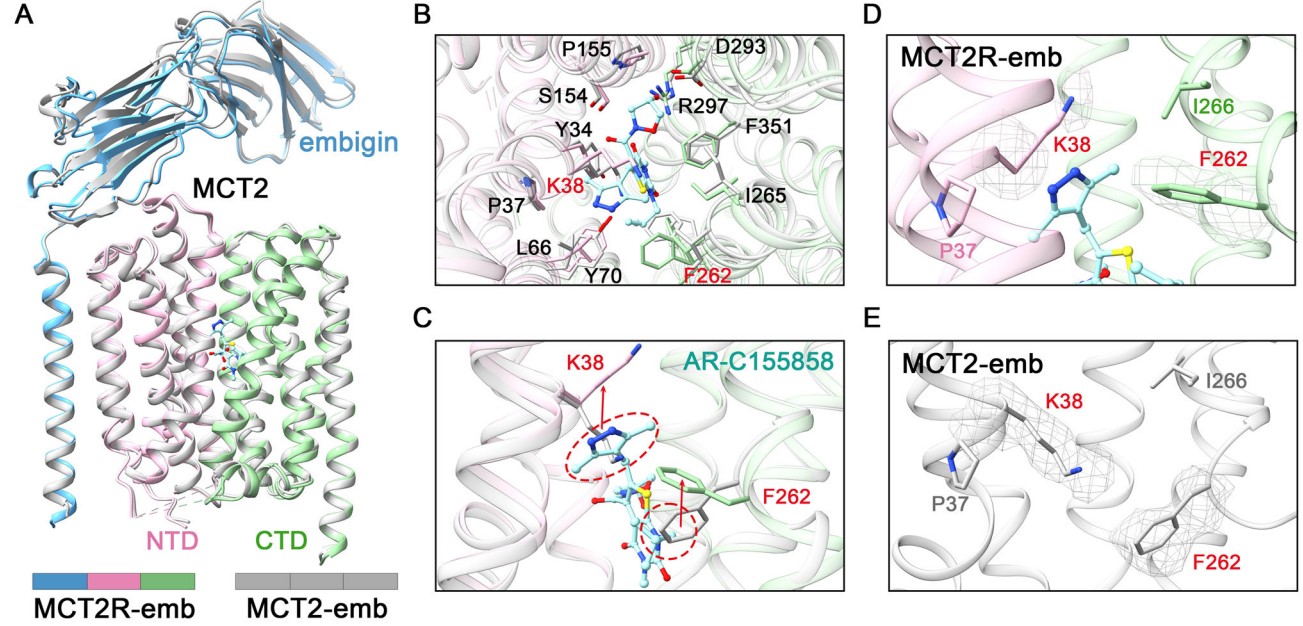

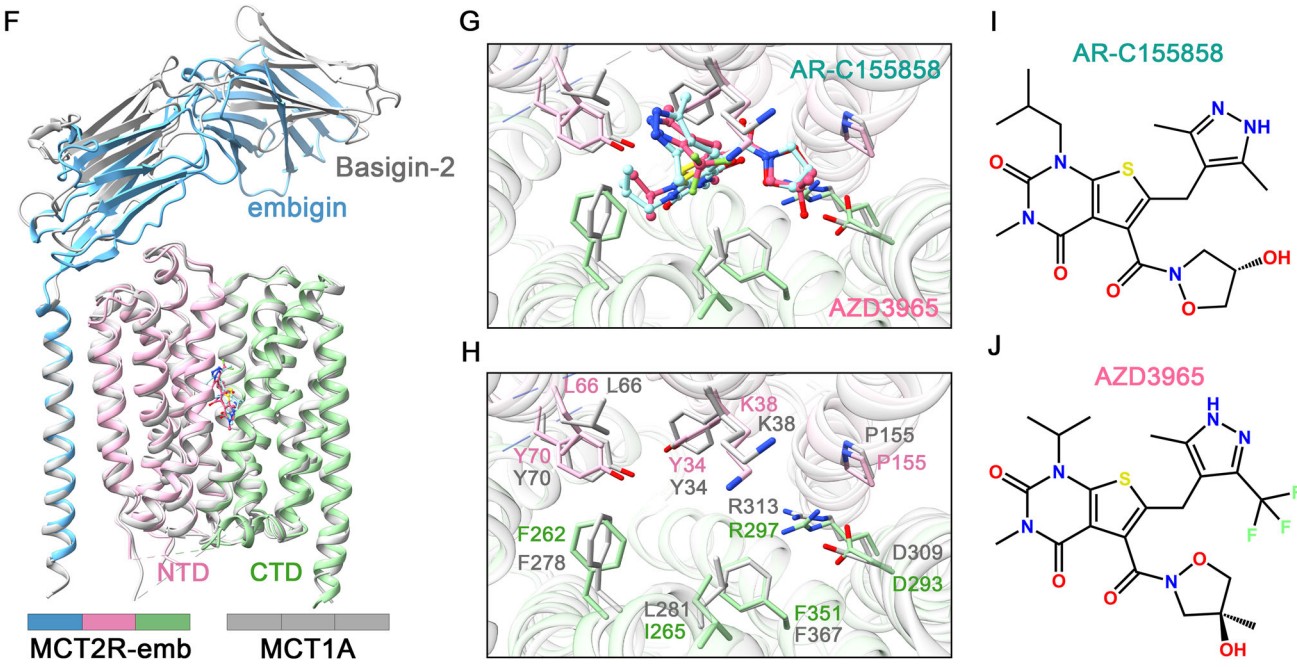

**Figure EV2. Structural comparison between human MCT2R-emb and MCT2-emb, MCT1A, respectively, related to Fig. 2.**

(A) MCT2-embigin exhibits identical conformation in the presence of AR-C155858. The two structures can be superimposed with an RMSD of 0.67 Å over 387 aligned Cα atoms. (B) Comparison of residues involved in the coordination of AR-C155858. AR-C155858 are shown as ball and sticks, and the residues are shown as sticks. (C) Conformational shift of K38 on TM1 and F262 on TM7 upon AR-C155858 binding. The red dashed circles indicate potential clashes should these residues exhibit the same conformation as in the apo state. (D, E) The cryo-EM density of K38 and F262 in MCT2R-emb and MCT2-emb, respectively. (F) Structural comparison of outward-open human MCT2R-emb (PDB ID: 9L0C) and outward-open MCT1A (PDB ID: 6LYY). The two structures can be superimposed with an RMSD of 0.85 Å over 367 aligned Cα atoms. (G, H) Comparison of residues involved in the coordination of AR-C155858 on MCT2R-emb or AZD3965 on MCT1A. AR-C155858 and AZD3965 are shown as cyan and magenta, respectively. The interacting residues are shown as sticks. The ligands are omitted for visual clarity in (H). Chemical structural formula of AR-C155858 (I), AZD3965 (J).

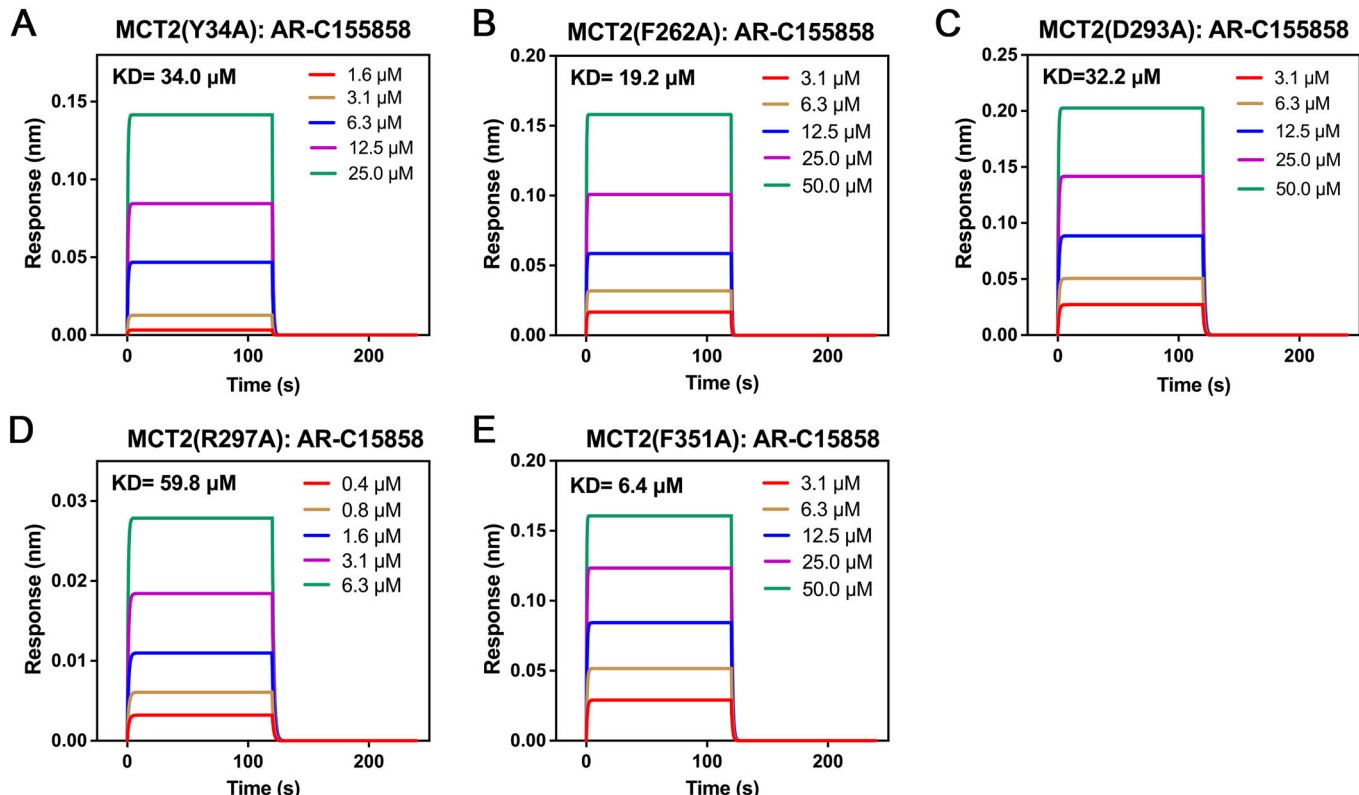

**Figure EV3. Binding affinities between MCT2 variants and AR-C155858, related to Fig. 2.**

(A–E) Bio-layer interferometry (BLI) measurement of the binding affinities between MCT2 variants and AR-C155858. The binding kinetics of AR-C155858 bound to MCT2(Y34A)-embigin (A), MCT2(F262A)-embigin (B), MCT2(D293A)-embigin (C), MCT2(R297A)-embigin (D), MCT2(F351A)-embigin (E). Source data are available online for this figure

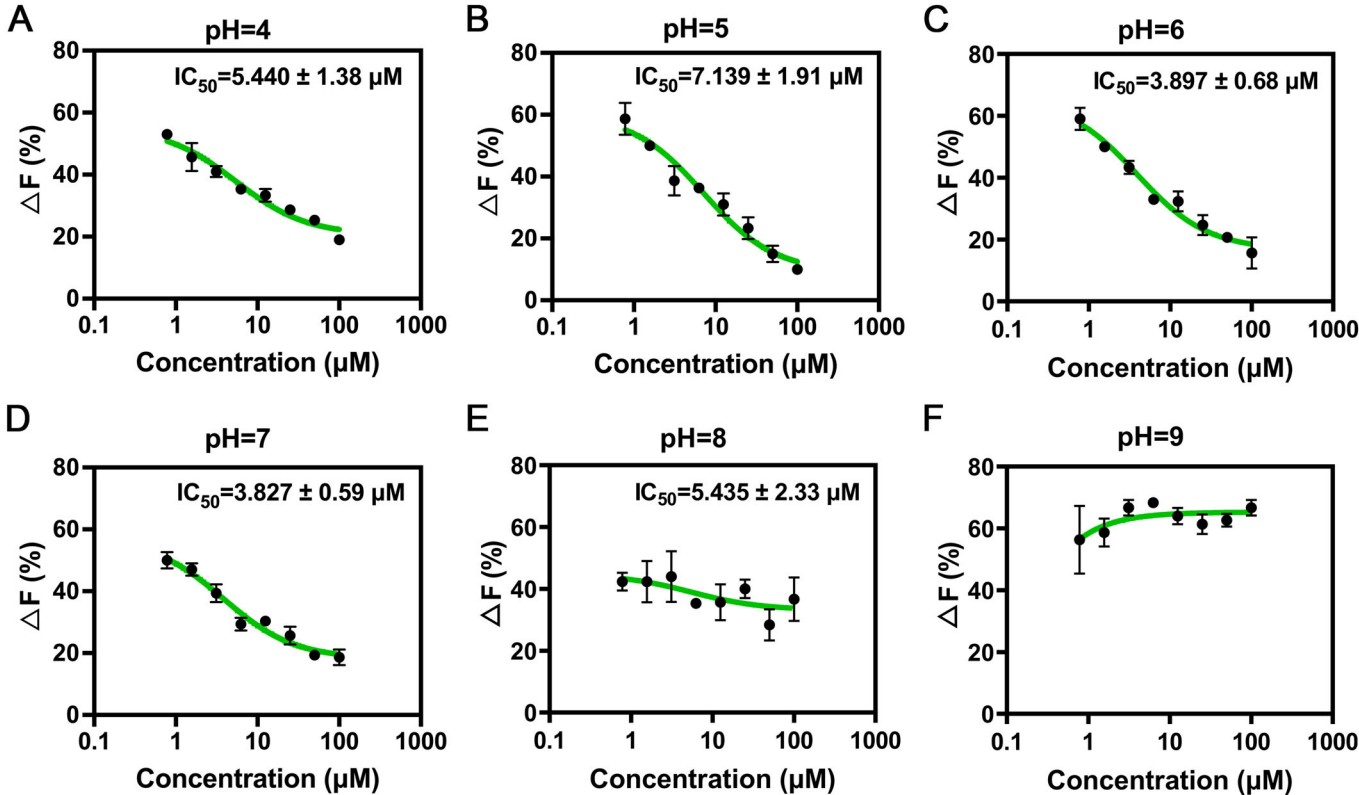

**Figure EV4. pH-dependent inhibition of MCT2 by Tucatinib, related to Fig. 3.**

The half maximal inhibitory concentration ($IC_{50}$) of Tucatinib on MCT2 across pH gradients, pH = 4 (**A**), pH = 5 (**B**), pH = 6 (**C**), pH = 7 (**D**), pH = 8 (**E**), pH = 9 (**F**). ($n = 3$). The graphical presentation and data analysis were conducted using GraphPad Prism 9. The data were displayed as mean ± standard deviation (SD). $n = 3$ biological replicates. Source data are available online for this figure

