## [Peer Review File · EMBO Reports]

Structure-guided screening identifies Tucatinib as dual inhibitor for MCT1/2

Binghong Xu, Xiaoyu Zhou, Yuanyue Shan, Sai Shi, Jiachen Li, Qinqin Liang, Ziyu Wang, Mingfeng Zhang, Yaxin Wang, Duanqing Pei, and Sheng Ye

Corresponding author(s): Sheng Ye (sye@tju.edu.cn), Duanqing Pei (peiduanqing@westlake.edu.cn), Yaxin Wang (wangyaxin@tju.edu.cn), Mingfeng Zhang (zhangmingfeng@westlake.edu.cn), Binghong Xu (binghong_xu@tju.edu.cn)

Review Timeline:

Submission Date:	11th Jun 25
Editorial Decision:	25th Jul 25
Revision Received:	8th Sep 25
Editorial Decision:	20th Oct 25
Revision Received:	28th Oct 25
Editorial Decision:	6th Nov 25
Revision Received:	10th Nov 25
Accepted:	17th Nov 25

Transaction Report:

Dear Dr. Xu

Thank you for the submission of your research manuscript to our journal. We have now received the full set of referee reports that is copied below.

As you will see, the referees acknowledge that the findings are interesting and that the conclusions are overall supported by the data presented but they also raise a number of concerns and have suggestions how to further strengthen the data, that need to be addressed.

Given the referee's constructive comments, we would like to invite you to revise your manuscript with the understanding that the referee concerns (as detailed above and in their reports) must be fully addressed and their suggestions taken on board. Please address all referee concerns in a complete point-by-point response. Acceptance of the manuscript will depend on a positive outcome of a second round of review. It is EMBO Reports policy to allow a single round of revision only and acceptance or rejection of the manuscript will therefore depend on the completeness of your responses included in the next, final version of the manuscript.

We realize that it is difficult to revise to a specific deadline. In the interest of protecting the conceptual advance provided by the work, we recommend a revision within 3 months (October 24th). Please discuss the revision progress ahead of this time with the editor if you require more time to complete the revisions.

I am also happy to discuss the revision further via e-mail or a video call, if you wish.

=====
IMPORTANT NOTE:

We perform an initial quality control of all revised manuscripts before re-review. Your manuscript will FAIL this control and the handling will be delayed IN CASE the following APPLIES:

- 1) A data availability section providing access to data deposited in public databases is missing. If you have not deposited any data, please add a sentence to the data availability section that explains that.
- 2) Your manuscript contains statistics and error bars based on $n=2$. Please use scatter blots in these cases. No statistics should be calculated if $n=2$.

=====

- 1) a .docx formatted version of the manuscript text (including legends for main figures, EV figures and tables). Please make sure that the changes are highlighted to be clearly visible.
- 2) individual production quality figure files as .eps, .tif, .jpg (one file per figure). Please download our Figure Preparation Guidelines (figure preparation pdf) from our Author Guidelines pages <https://www.embopress.org/page/journal/14693178/authorguide> for more info on how to prepare your figures.
- 3) a .docx formatted letter INCLUDING the reviewers' reports and your detailed point-by-point responses to their comments. As part of the EMBO Press transparent editorial process, the point-by-point response is part of the Review Process File (RPF), which will be published alongside your paper.
- 4) a complete author checklist, which you can download from our author guidelines (<<https://www.embopress.org/page/journal/14693178/authorguide>>). Please insert information in the checklist that is also reflected in the manuscript. The completed author checklist will also be part of the RPF.

5) Please note that all corresponding authors are required to supply an ORCID ID for their name upon submission of a revised manuscript (<<https://orcid.org/>>). Please find instructions on how to link your ORCID ID to your account in our manuscript tracking system in our Author guidelines (<<https://www.embopress.org/page/journal/14693178/authorguide#authorshipguidelines>>)

6) We replaced Supplementary Information with Expanded View (EV) Figures and Tables that are collapsible/expandable online. A maximum of 5 EV Figures can be typeset. EV Figures should be cited as 'Figure EV1, Figure EV2' etc... in the text and their respective legends should be included in the main text after the legends of regular figures.

7) Please format the "Data Availability" section at the end of the Methods using this format: "The [structural coordinates | microarray | mass spectrometry] data from this publication have been deposited to the [name of the database] database [URL] and assigned the identifier [accession | permalink | hashtag].". Please note that we need a link that resolves directly to the dataset. Please remove all other statements from the Data Availability section, it should only refer to the deposited datasets.

Additional information on source data and instruction on how to label the files are available <<https://www.embopress.org/page/journal/14693178/authorguide#sourcedata>>

10) Figure legends and data quantification:
The following points must be specified in each figure legend:

- the name of the statistical test used to generate error bars and P values,
 - the EXACT p-values (unless $p < 0.0001$),
 - the number (n) of independent experiments (please specify technical or biological replicates) underlying each data point,
 - the nature of the bars and error bars (s.d., s.e.m.)
- If the data are obtained from n {less than or equal to} 5, show the individual data points in addition to the SD or SEM.
- If the data are obtained from n {less than or equal to} 2, use scatter blots showing the individual data points.

See also the guidelines for figure legend preparation:
<https://www.embopress.org/page/journal/14693178/authorguide#figureformat>

11) Our journal encourages inclusion of *data citations in the reference list* to directly cite datasets that were re-used and obtained from public databases. Data citations in the article text are distinct from normal bibliographical citations and should directly link to the database records from which the data can be accessed. In the main text, data citations are formatted as follows: "Data ref: Smith et al, 2001" or "Data ref: NCBI Sequence Read Archive PRJNA342805, 2017". In the Reference list, data citations must be labeled with "[DATASET]". A data reference must provide the database name, accession number/identifiers and a resolvable link to the landing page from which the data can be accessed at the end of the reference. Further instructions are available at <<https://www.embopress.org/page/journal/14693178/authorguide#referencesformat>>.

12) All Materials and Methods need to be described in the main text using our 'Structured Methods' format. According to this format, the Methods section includes a Reagents and Tools Table (listing key reagents, experimental models, software and relevant equipment and including their sources and relevant identifiers) followed by a Methods and Protocols section describing

the methods, ideally using a step-by-step protocol format. The aim is to facilitate adoption of the methodologies across labs. Please download and fill our Reagents and Tools Table template (.docx), which you can find in our author guidelines: <https://www.embopress.org/page/journal/14693178/authorguide#structuredmethods>.

An example of a Method paper with Structured Methods can be found here: <https://www.embopress.org/doi/10.15252/msb.20178071>.

13) As part of the EMBO publication's Transparent Editorial Process, EMBO Reports publishes online a Review Process File to accompany accepted manuscripts. This File will be published in conjunction with your paper and will include the referee reports, your point-by-point response and all pertinent correspondence relating to the manuscript.

Yours sincerely,

=====

Referee #1:

Xu et al. present a rather broad study involving 1. structure elucidation by cryo-EM of human monocarboxylate transporter 2, MCT2, in complex with its chaperone embigin, 2. virtual screening of drugs for MCT2 inhibition identifying tucatinib (a HER2 tyrosine kinase inhibitor), and 3. testing of tucatinib in cell culture and tumor xenografts in mice. The structure part provides novel information because previously MCT2 was solved in the absence of embigin depicting the inward open conformation, whereas the new structure is outward open. The data mainly has confirmative character to earlier data obtained with the MCT1 isoform in complex with embigin due to high similarity between MCT1 and MCT2. Nevertheless, it is a valuable contribution. Virtual screening then suggested tucatinib to bind to MCT2 in a similar fashion as the high-affinity clinical candidate AZD3965, and the structurally related AR-C155858 compound. The predicted binding mode was confirmed by introducing MCT2 point mutations, and μM IC₅₀/K_d values were determined. Eventually, tucatinib was tested on HeLa cells in culture and as xenografts where it exhibited effects on cell growth. The authors come to the conclusion that tucatinib is a potent novel MCT1/MCT2 inhibitor with potential for use in human anti-tumor therapy besides HER2-dependent cancers.

Specific points:

1. AZD3965 and AR-C155858 have K_i and K_d values in the single-digit nanomolar range, tucatinib is micromolar. In order to reasonably inhibit MCT2 transport double-digit micromolar concentrations were used by the authors. Tucatinib accordingly is less potent by 3 orders of magnitude (IC₅₀ and SPR affinity assays). The statements of a "potent" inhibitor (page 13, line 299 and elsewhere in the ms) appears overstated in this light.

2. Assaying AZD3965 and/or AR-C155858 besides tucatinib in the SPR would have been required controls (suppl. Fig. S9).

3. In order to test if the effect of tucatinib on tumor cell growth is conveyed via MCT1 or MCT2 inhibition, the authors selected HeLa cells which they claim are free of HER2 (p.14, l. 332 and elsewhere), i.e. the actual target of tucatinib. MCT1 and MCT2 are expressed in these cells as shown in Fig. 5A. The given reference on HER2 expression in HeLa states <5% rather than "HER2-negative". A true establishment of the HER2 levels in the tested cells would be required here to exclude tucatinib action on HER2, e.g. as an addition to Fig. 5A.

4. Knockdown of either MCT1 or MCT2 expression by sh-RNA strongly affected HeLa cell viability suggesting that MCT1 and MCT2 are required for proper cell physiology under the assay conditions. This comes as a surprise since a) one isoform still remains, and b) according to the methods section the cells were aerated normally and no special conditions were selected that would drive the cells into anaerobic energy metabolism that would require MCT transport action. MCT knockdowns are the basis for the majority of follow-up experiments. Therefore clarification of this phenomenon is key for the validity of the remainder of the study.

5. Rescue of the knockdowns by expressing MCT1 and MCT2 from plasmids did not work, i.e. the transfected cells still exhibited the strong growth defect (not shown in the main paper but in the supplemental Fig. S12A,B). This is disturbing and does not help building confidence into the setup.

6. In the knockdown cells, the authors further tried to express mutants of MCT1 and MCT2 with amino acid exchanges in the tucatinib binding site (F278A and F351A, resp.). However, both were non-functional in terms of transport activity (see Fig. 4J,K). Still, the absence of an additional effect of tucatinib on the already strongly compromised cell growth was taken as evidence for MCT-selectivity by the authors. This convoluted approach is rather questionable.

7. The mouse xenografts were also HeLa-based; hence, the points stated above need to be considered here as well.

In summary, while the Emb/MCT2-structure part and the finding of tucatinib as a micromolar inhibitor of MCT1 and MCT2 appear sound and valid, the estimation of its potency, specificity and applicability in vivo has major issues.

Referee #2:

In this work Xu, Zu and coauthors determine the structures of MCT2-embigin complex in absence and in presence of a known anti-cancer inhibitor AR-C155858, now in clinical trials for the treatment of solid tumors. Although the resolution is not very high and clashscores are a bit high, the cryo-EM density and fit of the protein and the ligand seems convincing from the reported figures and PDB report. The interaction interface between MCT2 monomer and embigin is well supported by biochemical data. Leveraging their structures both obtained in the outward-open state, the authors further conduct molecular dynamics and docking simulations of the MCT2 bound to AR-C155858, thereby exploring the dynamics and flexibility of the system. In addition, they perform a virtual screening with few approved inhibitors to explore the possibility of repurposing existing drugs to inhibit MCT1/2, overexpressed in some cancers, including cervical cancer. This computational search pointed at Tucatinib, a well-known tyrosine kinase inhibitor (HER). Based on conservation studies, the authors derived that Tucatinib could inhibit both MCT1 and MCT2, being an interesting compound with broader action in treating cancer. The authors proved Tucatinib being effective in inhibiting MCT1 and MCT2-mediated pyruvate transport in cervical cancer HeLa cells and also verified its action in vivo.

Overall the paper is interesting and brings novel insights to the drug targeting of MCT1/2. I appreciated the use of a multidisciplinary approach to tackle inhibitory properties of available small compounds towards cancer, including virtual screening and cell biology assays. The manuscript could be made a bit more compact for the EMBOR format by merging Results and Discussion. There are 7 main Figures and Supplementary Figures, while I suppose some have to be converted to EV figures as there is a max of 5 main items in my knowledge.

I recommend publication in EMBOR with minor revisions and clarifications regarding the following points:

-Abstract: "Cell surface glycoproteins, Basigin or embigin, associate with proton-coupled 36 monocarboxylate transporters (MCTs) to form heterodimers", concept recalled in introduction and discussion.

Are all the proton-coupled MCT forming complexes with embigin or only MCT1 and 2? what determines the specificity for basigin and embigin with respect to MCT1 and MCT2? Why only MCT1-4 and not others, such as MCT8? I suggest to be more accurate in properly introducing/discussing this point relying on conservation or published data

-Abstract: "the deepening (..)" I would rephrase in: A deeper understanding of MCT regulation may open avenues for the development of novel inhibitors (..)

-Line 174: perhaps you could remove this sentence to make the text flow better. Having well defined SS does not mean you will properly see the ligand:

"The secondary structure features of MCT2R-emb are well-defined, allowing us to visualize the binding mode of AR-C155858(Fig.S3and S4).We will discuss the inhibition mechanism in a later session"

-Fig4 legend: typo 'in the present' > "in the presence. Please check this typo across the paper, there are 4

-There is a lot of yellow highlighted text, but I don't know whether that's functional?

-line 207: "amino acid linker (GGSG), single transfection led to substantial expression of the fusion protein at the cell surface". What do you mean by "substantial"? This variation in localisation should be shown by better images as in the authors' Cell Report article and by some semi-quantitative localisation assessment (for example in Fiji)

-Do you think AR-C155858 inhibits in the same way dimeric MCT2 and MCT2-emb, given the impossibility of isolating the homo-dimeric MCT2 in complex with AR-C155858? Can you better clarify this point?

-Line 268: this sentence may be confusing for the reader, can you explain better?

"This signal was significantly decreased upon pre-incubation with AR-C155858. However, due to the overlapping binding sites, the tested mutations affected the transport activity of MCT2-embigin and escaped inhibition of AR-C155858 (Fig. 2I)." I recommend reporting on the bar plots the % of inhibition normalising the signal with respect to the single construct.

-Line 278: can you introduce HER acronym?

-279 "in both MCT1 ..." : "by both MCT1..." may sound better

-Is there any pH dependence of compound inhibitory properties and MCT2-emb association? I would mention it in the discussion

-line 339: can you explain what a CCK-8 is?

-line 352-358: can you better motivate why these mutants are not inhibited?

-line 382: How is migration/growth relate to pyruvate uptake? Can you better describe the link between the MCT2-dependant transport and the measured property in cells?

-It is remarkable that you find Tucatinib fits in the central gate of the MCT2 and can indeed inhibit both MCT1 and MCT2. However, the affinity is quite low (in the range of μM rather than nM as for AR-C155858) and there is no available structure of tucatinib in complex with MCT1 and MCT2. In cells, (Fig4J) it would be useful to show the relative % inhibition with respect to each construct. In vitro, have you tested by Biacore assay that at least one of these mutants is not inhibited by tucatinib? This would strengthen the hypothesis that the inhibitor binds in the central pocket.

-Because of the μM you probably need larger amounts of tucatinib to exert an effect on MCT1/2 by saturating them, yet you only expose the cells to $5\mu\text{M}$ of the compound, close to the K_d . Do you think in your cellular setup that the compound is actually inhibiting MCT1/2 or the kinase inhibition is prevailing at these concentrations? Could you elaborate on this please and discuss future applications?

Referee #3:

Summary

Xu et al. present cryo-EM structures of the heterodimer of MCT2 and embigin, both in the apo state and with a bound inhibitor. Embigin likely facilitates plasma membrane localisation of MCT2, which enables correct function. They also identify an inhibitor, Tucatinib, with potential relevance for the treatment of cervical cancer. The data is sound, and covers a wide array of analysis, thus perhaps being too long and having a scope that is too wide for what is requested for this journal, presenting multiple structures and their validation and analysis, as well as drug target determination and testing. The findings are novel and of high significance, as it can directly lead to trials on cervical cancer for this drug target. For further issues, see the major and minor comments, below.

Response to requested reviewer questions:

1. Does this manuscript report a single key finding? NO

There are two key findings: the structures with and without inhibitor, and the determined cancer drug target.

2. Is the reported work of significance (YES), or does it describe a confirmatory finding or one that has already been documented using other methods or in other organisms etc (NO)? YES

3. Is it of general interest to the molecular biology community? YES

Further understanding of a cancer-relevant protein, including identifying a promising drug target.

4. Is the single major finding robustly documented using independent lines of experimental evidence (YES), or is it really just a preliminary report requiring significant further data to become convincing, and thus more suited to a longer-format article (NO)? YES

Major comments

1. Please overview the language, as there are numerous smaller grammatical errors. Examples (corrections underlined> include

line 62: "monocarboxylates is primarily" and line 83: "research that demonstrates"

2. Please also revise the figure numbering, as several figures are referenced out of order. This applies both to the overall numbering of figures (S3 is referenced before S2, for instance), but also the organisation within each figure (figures 5 and 6 could be split into 3 figures and reorganised to better align with how they are referenced in the text, for instance). Table S2 is never referenced in the text.
3. The introduction needs to present what is known about MCT structure, its topology, transport, dimeric states, key residues etc. This is especially relevant regarding line 78, where it is implied that the reader already knows about the dimeric state of MCT1, and line 355, where key residues are mentioned. Some text starting at line 176 could perhaps be placed here instead. A thorough introduction of dimeric state is especially important, considering the data presented. When are these proteins dimeric? How does their dimeric state change? How does their different dimeric states affect transport activity? In addition, considering the significant amount of comparison with MCT1, it is relevant with a more detailed introduction to their similarities and, importantly, differences.
4. What is the role of embigin? In the paragraph starting a line 197 it is shown that MCT2 colocalises with embigin at the plasma membrane. However, according to Zhang et al, 2020, cited in the paper, MCT2 can localise to the plasma membrane also without the presence of embigin. Were control experiments done where localisation of MCT2 was assessed without the presence of embigin? It appears that MCT2 is capable of transport as a homodimer, and if it also localises correctly, what is the role of embigin in vivo? Is it known if transport is possible for the heterodimer, or is that only carried out by the homodimer? Also regarding the role of the heterodimer, is it then assumed that the AR-C155858 only binds to the heterodimer? Would inhibition in part be a lack of activity because the inhibitor locks MCT2 in the heterodimeric state? An analysis of this is especially lacking in the discussion.
5. For structural comparisons there is a large focus on only comparing the individual NTD and CTD, but I lack comparisons of the protein as a whole. Examples include line 214 and figure S5. While showing that the NTD and CTD remain relatively similar is useful, this needs to be put in context of the movements of the whole protein. To emphasise this similarity, while still showing overall movement, this could be done by superimposing just, for example, the NTD, but showing the whole protein, thus giving the reader a chance to see the overall changes, while still appreciating the domain similarities. Leaving out a whole protein comparison makes it hard to put the current images and analysis into context.

Minor comments

1. Why are some regions highlighted in yellow?
2. Line 103: MCT2 has a K_i value >10 nM. Why is the specific value given for MCT1, but not MCT2?
3. 216: "... conformational change of TM5 still exists," this is confusing, as it makes it sound like the significant conformational change that is seen in the homodimer can also be seen in the heterodimer, which is not what I think you mean?
4. 246: also mention the molecule bound to the MCT1 structure here, along with the PDB ID.
5. 249: "despite binding to different proteins ..." is confusingly worded, since AR-C155858 is not bound in both structures, only one of them. Please rephrase.
6. 255 and figure 2D: what docking is referred to here? The drug screening? Why was docking otherwise carried out, since you have the structure? This needs to be more clear in the text and figure legend.
7. 263: what does MMGBSA stand for?
8. 270: it has not been stated where the cargo is expected to bind.
9. 291: if Tucatinib can inhibit both MCT1 and MCT2 then, by definition, is it really specific? The way this sentence is written implies that the cell usually expresses only one of them (see major point 3 about expanding the introduction)?
10. 306: "simultaneously" makes it sound like the same molecule will bind to both proteins at the same time, which is not the case here.
11. 639: why not have these primers listed along with the others in table S2?
12. Figure 1C, and (to a lesser extent) 2A: Consider making the interior of the slab a darker shade of grey, currently it is almost white, which would increase figure clarity.
13. Figure 1G and H: what are these? Western blots?
14. Figure 3A and B and 4A: the binding site is what is most interesting here, I would recommend zooming in so the figures mainly show that.
15. Figure 3C: how were these targets selected to be shown here? Highest predicted affinity among those screened?
16. Figure 6: the legend should list the constructs used. What does NC stand for?
17. Figure S5B: this figure should show that embigin and the NTD move together, but right now it only shows the same thing as figure S5A. Scrapping this figure and adding an overall alignment, as suggested in major point 5, would fix this.
18. Figure S5C: It would be relevant to indicate the location of the membrane in this figure.
19. Figure S12: were the knockdown strains also measured, as a control? They should be included here.

Response to reviewers' specific comments:

Reviewer #1:

Xu et al. present a rather broad study involving 1. structure elucidation by cryo-EM of human monocarboxylate transporter 2, MCT2, in complex with its chaperone embigin, 2. virtual screening of drugs for MCT2 inhibition identifying tucatinib (a HER2 tyrosine kinase inhibitor), and 3. testing of tucatinib in cell culture and tumor xenografts in mice. The structure part provides novel information because previously MCT2 was solved in the absence of embigin depicting the inward open conformation, whereas the new structure is outward open. The data mainly has confirmative character to earlier data obtained with the MCT1 isoform in complex with embigin due to high similarity between MCT1 and MCT2. Nevertheless, it is a valuable contribution. Virtual screening then suggested tucatinib to bind to MCT2 in a similar fashion as the high-affinity clinical candidate AZD3965, and the structurally related AR-C155858 compound. The predicted binding mode was confirmed by introducing MCT2 point mutations, and μM IC₅₀/K_d values were determined. Eventually, tucatinib was tested on HeLa cells in culture and as xenografts where it exhibited effects on cell growth. The authors come to the conclusion that tucatinib is a potent novel MCT1/MCT2 inhibitor with potential for use in human anti-tumor therapy besides HER2-dependent cancers.

We appreciate the reviewer's positive evaluation of our work and its significance. We have addressed all of his/her comments with significant changes to the manuscript, including new text, experiments, and further discussion.

Specific points:

1. AZD3965 and AR-C155858 have K_i and K_d values in the single-digit nanomolar range, tucatinib is micromolar. In order to reasonably inhibit MCT2 transport double-digit micromolar concentrations were used by the authors. Tucatinib accordingly is less potent by 3 orders of magnitude (IC₅₀ and SPR affinity assays). The statements of a "potent" inhibitor (page 13, line 299 and elsewhere in the ms) appears overstated in this light.

We appreciate the reviewer's critique!

We would like to clarify that the affinity values were obtained using different experimental methods and target systems. In previously published studies, AR-C155858 has been reported as a potent inhibitor of MCTs, with K_i values of 2.3 nM for MCT1 and <10 nM for MCT2. These K_i values were derived from experiments measuring the inhibition of L-lactate uptake into rat erythrocytes by AR-C155858 (Ovens et al., 2010a; Ovens et al., 2010b).

Regarding AZD3965, some researchers reported K_d values of 1.6 nM for MCT1 and 9.6 nM for MCT2 (Critchlow et al., 2012). In another study, the affinity of AZD3965 for MCT1 was approximately 40 nM, as determined by microscale thermophoresis

(MST) binding assays (Wang et al., 2021).

In our study, we purified the MCT1-Basigin-2 and MCT2-embigin complexes and evaluated their binding affinities using surface plasmon resonance (SPR) and bio-layer interferometry (BLI). As shown in Figures A and B below, the SPR results showed K_d values of 1.91 μM for MCT1-Basigin-2 and 6.99 μM for MCT2-embigin with AR-C155858. The BLI assays yielded KD values of 0.1 μM for MCT1-Basigin-2 and 0.6 μM for MCT2-embigin with AR-C155858 (Fig. 2 K, J). Unfortunately, we did not obtain detectable signals in MST assays. Moreover, we determined the binding affinity between Tucatinib and MCT1 or MCT2 using two independent methods. As shown in panels C and D below, surface plasmon resonance (SPR) measurements yielded affinity values of 5.08 μM for MCT1-Basigin-2 and 8.47 μM for MCT2-embigin. In addition, bio-layer interferometry (BLI) assays—presented in Figure 4 of the main text—demonstrated affinities of 3.9 μM for MCT1-Basigin-2 (Fig. 4 J) and 0.6 μM for MCT2-embigin (Fig. 4 M). Therefore, we believe that differences in experimental methods and target constructs may contribute to the variations in the reported affinity values.

To maintain consistency in experimental methodology, we uniformly selected the data from bio-layer interferometry (BLI) assays for presentation in the results. The results indicate that the affinity of Tucatinib for MCT1-Basigin-2 (3.9 μM , Fig. 4 J) is one order of magnitude lower than that of AR-C155858 (0.1 μM , Fig. 2 K), while its affinity for MCT2-embigin (0.6 μM , Fig. 4 M) is within the same order of magnitude as that of AR-C155858 (0.6 μM , Fig. 2 J). Therefore, we consider Tucatinib to also be a potent inhibitor of both MCT1 and MCT2.

Finally, beyond affinity measurements, we evaluated the functional inhibition of substrate transport by these compounds using a pyruvate sensor-based assay to monitor fluorescence changes resulting from pyruvate uptake inhibition. The pyruvate transport assay demonstrated that Tucatinib significantly inhibited pyruvate uptake mediated by both MCT1 and MCT2 in HEK-293T cells, with IC_{50} values of 2.596 μM for MCT1 (Fig. 3 G) and 2.712 μM for MCT2 (Fig. 3 I). As positive controls, the IC_{50} values for AZD3965 on MCT1 and AR-C155858 on MCT2 were 1.238 μM (Fig. 3 H) and 0.981 μM (Fig. 3 J), respectively. These functional data further support that the inhibitory potency of Tucatinib is comparable to that of AR-C155858 or AZD3965, justifying our description of Tucatinib as a potent inhibitor of MCT1 and MCT2.

Figure for referee with unpublished data and its description has been removed upon request by the authors.

2. Assaying AZD3965 and/or AR-C155858 besides tucatinib in the SPR would have been required controls (suppl. Fig. S9).

We appreciate the reviewer's suggestion. We measured the affinity of AR-C155858 for MCT2-embigin and MCT1-basigin (Fig. 2 J, K). Additionally, due to scheduling constraints with instrument availability and to maintain consistency in the methodology throughout our study, all affinity measurements presented in this work were performed using bio-layer interferometry (BLI). This uniform approach ensures comparability across all affinity data reported.

3. In order to test if the effect of tucatinib on tumor cell growth is conveyed via MCT1 or MCT2 inhibition, the authors selected HeLa cells which they claim are free of HER2 (p.14, l. 332 and elsewhere), i.e. the actual target of tucatinib. MCT1 and MCT2 are expressed in these cells as shown in Fig. 5A. The given reference on HER2 expression in HeLa states <5% rather than "HER2-negative". A true establishment of the HER2 levels in the tested cells would be required here to exclude tucatinib action on HER2, e.g. as an addition to Fig. 5A.

We appreciate the reviewer's reminding. In our manuscript, we described HeLa cells as "HER2-negative" based on the reference (Jia et al., 2022), which explicitly states:

“As shown in Figure 2A, all BT-474 and SKOV-3 cells were Her2-positive (positive rate = 100%). In contrast, almost no Her2-positive cells were detected in HeLa cells (positivity rate < 5%). Therefore, BT-474 and SKOV-3 cells were Her2-positive, while HeLa cells were Her2-negative.”

To further address the reviewer’s concern, we have performed additional qPCR experiments to quantify HER2 expression in HeLa cells (Appendix Fig. S9). The results confirm that HER2 mRNA levels are very low, accounting for approximately 6.7% of MCT1 mRNA expression. This supports the use of HeLa cells to minimize potential confounding effects from HER2 inhibition.

In accordance with the reviewer’s suggestion and to improve precision in terminology, we have revised our description of HeLa cells throughout the manuscript to “HER2-deficient” rather than “HER2-negative”. We have also incorporated the new qPCR data into the revised manuscript to provide direct evidence of HER2 expression levels in the cells used in our study (Appendix Fig. S9).

4. Knockdown of either MCT1 or MCT2 expression by sh-RNA strongly affected HeLa cell viability suggesting that MCT1 and MCT2 are required for proper cell physiology under the assay conditions. This comes as a surprise since a) one isoform still remains, and b) according to the methods section the cells were aerated normally and no special conditions were selected that would drive the cells into anaerobic energy metabolism that would require MCT transport action. MCT knockdowns are the basis for the majority of follow-up experiments. Therefore clarification of this phenomenon is key for the validity of the remainder of the study.

We thank the reviewer for this critical and insightful comment regarding the observed strong effect on HeLa cell viability following knockdown of MCT1 or MCT2 under standard culture conditions. We agree that this is a key observation that warrants careful discussion, and we appreciate the opportunity to clarify this point.

We were also initially surprised by the potency of the phenotype upon single isoform knockdown. However, we believe this finding demonstrates a profound dependency of HeLa cells on monocarboxylate transporter function, which can be explained by their well-established metabolic profile:

HeLa cells, like many cancer cell lines, exhibit a high glycolytic rate and lactate production even in the presence of ample oxygen (aerobic glycolysis or the Warburg Effect) (Warburg O, et al., 1924; Warburg O, et al., 1956; Heiden M G V., 2009). This constant lactate generation requires efficient export to maintain intracellular pH homeostasis. Under physiological conditions, MCT1-4 cooperate to form a lactate shuttling system that maintains lactate homeostasis between glycolytic and oxidative cells (Brooks, 2018; Gertz et al., 1981; Pellerin et al., 1998). Our data suggest that MCT2 may also play a significant and non-redundant role in this process in HeLa

cells.

While one isoform remains expressed, MCT1, MCT2, and MCT4 have different kinetic properties (e.g., affinity for lactate) and regulatory mechanisms. They may not be fully redundant but instead work cooperatively to handle the high lactate flux. Knocking down one critical isoform (e.g., MCT1, a major lactate exporter) could disrupt this balance irreversibly, leading to intracellular lactate accumulation, acidification, and consequently, cell death (Halestrap, A. P., 2013; Pisarsky et al., 2016). This explains why the remaining isoform cannot fully compensate.

This phenomenon, while adding a layer of complexity to our genetic experiments, ultimately strengthens our main conclusion regarding the inhibitor's mechanism of action. The fact that the knockdown itself recapitulates a severe growth defect indicates that MCT function is essential for HeLa cell proliferation under standard conditions.

Furthermore, the key observation in our knockdown experiments was that in cells where the target (MCT1 or MCT2) was already severely depleted, the additional effect of our inhibitor was significantly diminished. This is logically consistent: if the target protein is absent, a targeted inhibitor can have little further effect. This result provides strong genetic evidence that the inhibitor's efficacy is specifically dependent on the presence of MCT1 or MCT2.

5. Rescue of the knockdowns by expressing MCT1 and MCT2 from plasmids did not work, i.e. the transfected cells still exhibited the strong growth defect (not shown in the main paper but in the supplemental Fig. S12A,B). This is disturbing and does not help building confidence into the setup.

We sincerely appreciate you raising this critical point. We fully agree that the incomplete reversal of the shRNA-induced growth defect by plasmid-mediated overexpression of MCT1 or MCT2 (Appendix Fig. S11A, B in the updated version), is a phenomenon that requires thorough explanation, and we understand that it raises concerns regarding the experimental system. We thank you for the opportunity to provide a rationale for this observation and to clarify why we believe the results continue to robustly support our central conclusion.

MCT1/2/4 are known to contribute to cancer development through multiple mechanisms (Payen et al., 2020). MCT1 and MCT4 play a critical role in lactate shuttling in “metabolic symbiosis” (Pisarsky, et al., 2016; Sonveaux et al., 2008). We hypothesize that the sustained knockdown mediated by shRNA may have induced profound metabolic dysregulation in HeLa cells (e.g., intracellular lactate accumulation and pH imbalance). This metabolic “reprogramming” may be difficult to fully reverse within the short timeframe of a transient transfection experiment. Furthermore, the expression levels, subcellular localization, and interaction with

requisite chaperone proteins (Basigin or embigin) of the transfected proteins may not perfectly recapitulate the precise regulation of endogenous MCTs. Consequently, the failure to completely restore the complex phenotype of overall cellular growth rate represents a limitation observed in our experimental approach.

We hope to emphasize that the principal aim of this rescue experiment was not the absolute normalization of cell growth, but rather the specific assessment of whether sensitivity to our inhibitor is dependent on MCT protein expression levels. From this perspective, the experiment provides pivotal evidence: a) In MCT1- or MCT2-knockdown cells, the loss of target protein abrogated the effect of our inhibitor (Figs. 5 and 6). b) In cells subjected to rescue (with mRNA restoration confirmed by qPCR), although overall cell viability remained compromised, the effect of the inhibitor was significantly restored (Appendix Fig. S11). This indicates that upon reconstitution of MCT protein levels, the drug can engage its target and elicit its cytotoxic effect again.

Therefore, the successful rescue of drug sensitivity, rather than the complete rescue of cell growth, constitutes the most direct and specific evidence demonstrating that our inhibitor targets MCT1 and MCT2. The restoration of the cellular response to the drug provides compelling evidence for a direct causal relationship between the compound's mechanism of action and MCT protein function.

6. In the knockdown cells, the authors further tried to express mutants of MCT1 and MCT2 with amino acid exchanges in the tucatinib binding site (F278A and F351A, resp.). However, both were non-functional in terms of transport activity (see Fig. 4J,K). Still, the absence of an additional effect of tucatinib on the already strongly compromised cell growth was taken as evidence for MCT-selectivity by the authors. This convoluted approach is rather questionable.

We are profoundly grateful for your insightful comments on the experiments. We acknowledge that, on the surface, the approach of expressing a non-functional mutant in already viability-compromised cells and observing no additional drug effect may appear convoluted. However, we wish to clarify that this experiment was never intended to rescue cellular viability. Instead, it was meticulously designed as a definitive genetic control to irrefutably demonstrate that the drug's mechanism of action is dependent on binding to specific residues on the MCT protein.

The core objective of this experiment was not functional rescue but rather a stringent test of drug specificity. F278 and F351 are key amino acid residues for the binding of Tucatinib to MCT1 and MCT2 respectively (Fig. 4). This conclusion is further supported by our affinity measurement results: the affinity of the MCT1-F278A mutant for Tucatinib became undetectable, while the affinity of the MCT2-F351A mutant for Tucatinib decreased to 210.7 μ M (Fig. 4 P). These mutations were strategically chosen to specifically disrupt inhibitor binding. Besides, these amino

acid residues are also critical for substrate binding. Mutations within a ligand-binding pocket can often perturb overall protein conformation or stability. Crucially, however, this finding does not weaken our conclusion; rather, it reinforces it by highlighting the critical nature of this structural region for overall MCT function.

In the Wild-Type Rescue Group: Expression of functional MCT (capable of binding the drug) restored cellular sensitivity to the drug. This demonstrates that drug efficacy requires MCT protein; In the Mutant Rescue Group: We obtained a crucial “double-negative” result: a) Cellular viability remained low, consistent with the Wild-Type. b) The addition of the drug produced no additional effect. Result (b) is paramount. It demonstrates that when the MCT present in the cell is a drug-binding-incompetent version, the drug is rendered completely inactive. This result is consistent with the “no effect” observed in knockdown cells completely lacking MCT targets. This forms a rigorous causal chain: the biological effect of the drug is strictly dependent on its physical interaction with specific amino acid residues on the MCT protein. Similar experimental phenomena have also been observed in other research papers (Song et al., 2023).

Therefore, this seemingly “convoluted” approach was, in fact, engineered to achieve the most stringent level of target validation. It provides direct evidence that our inhibitor functions not through non-specific cytotoxicity, but by specifically engaging a defined binding pocket on MCT1/2.

7. The mouse xenografts were also HeLa-based; hence, the points stated above need to be considered here as well.

Thank you for raising this important point, which effectively connects the logic between our in vitro and in vivo investigations. We fully agree that the interpretation of the in vivo experimental results must be built upon a solid and reasonable explanation of the in vitro mechanisms.

We understand your concern that if there are questions regarding the MCT knockdown model in HeLa cells in vitro, then the mechanistic interpretation of the in vivo experiments based on HeLa xenografts could also be affected. In our responses to the previous comments, we have provided detailed explanations and clarifications regarding the design, results, and conclusions of our in vitro experiments. We believe that, when correctly interpreted, those genetic and pharmacological experiments provide compelling evidence that Tucatinib targets MCT1/2.

At the same time, we wish to emphasize that the in vivo mouse model experiment holds its own independent and irreplaceable value in preclinical research. The primary objective of this study was to evaluate the drugability of Tucatinib in a complete biological system, including its ability to inhibit tumor growth, its pharmacokinetic profile, and its preliminary safety. Regardless of the detailed molecular mechanism,

the significant tumor growth inhibition we observed is itself a crucial and positive finding, providing a solid foundation for its further development.

More importantly, the results of the *in vivo* experiment are highly consistent with the mechanism we elucidated *in vitro*. The fact that the compound demonstrated potent inhibitory effects on the same model (HeLa cells) in both settings strongly suggests a common mechanism of action. Therefore, the animal experiment, on another level, corroborates and supports our core conclusion—that this MCT1/2 dual inhibitor effectively inhibits the growth of MCT-dependent HeLa tumors both *in vitro* and *in vivo*.

In summary, we regard the *in vivo* experiment as a vital link in the entire story chain. It not only demonstrates the therapeutic potential of the compound but also provides mutual support for the mechanistic findings uncovered *in vitro*.

In summary, while the Emb/MCT2-structure part and the finding of tucatinib as a micromolar inhibitor of MCT1 and MCT2 appear sound and valid, the estimation of its potency, specificity and applicability *in vivo* has major issues.

We once again extend our sincerest gratitude for the considerable time and effort you have dedicated to reviewing our manuscript and providing such profound, detailed, and constructive feedback. We are greatly encouraged by your positive assessment of the structural biology aspects of our work and the initial discovery of Tucatinib's activity. Simultaneously, we fully understand and highly value your core concerns regarding the evaluation of the compound's potency, specificity, and potential for *in vivo* application.

We have thoroughly considered and addressed each of your points in a point-by-point manner. Building upon this, we would like to provide a comprehensive response to your overarching comments:

We agree that precisely evaluating the contribution of a single target within complex cellular models presents challenges. Through genetic knockdown, rescue experiments, and experiments with key binding-site mutants, we have constructed a multi-faceted and mutually reinforcing chain of evidence. Although individual experiments, such as the incomplete rescue of cell growth, have their technical limitations, we believe that, viewed holistically, the data collectively provide strong support for the core conclusion that Tucatinib functions through the specific inhibition of MCT1 and MCT2. We will revise the manuscript to more cautiously discuss both the limitations and strengths of these data, avoiding overinterpretation.

We understand that the mechanistic interpretation of data from the HeLa xenograft model is closely linked to the *in vitro* findings. We will make it explicitly clear that the primary objective of the *in vivo* experiment was to validate the compound's

antitumor efficacy at the whole-animal level. At the same time, its high degree of consistency with the in vitro mechanism provides supporting evidence for its mode of action at this level.

We will discuss the limitations of the experiments more candidly in the Results and Discussion sections. We firmly believe that incorporating your valuable insights will significantly enhance the scientific rigor and logical clarity of this paper. The core value of our research lies in the first discovery and preliminary characterization of Tucatinib's novel function as a dual MCT1/MCT2 inhibitor, thereby providing a new and valuable chemical tool and candidate starting point for the field. We thank you again for your exceptional work in reviewing our manuscript.

Reviewer #2:

In this work Xu, Zu and coauthors determine the structures of MCT2-embigin complex in absence and in presence of a known anti-cancer inhibitor AR-C155858, now in clinical trials for the treatment of solid tumors. Although the resolution is not very high and clashscores are a bit high, the cryo-EM density and fit of the protein and the ligand seems convincing from the reported figures and PDB report. The interaction interface between MCT2 monomer and embigin is well supported by biochemical data. Leveraging their structures both obtained in the outward-open state, the authors further conduct molecular dynamics and docking simulations of the MCT2 bound to AR-C155858, thereby exploring the dynamics and flexibility of the system. In addition, they perform a virtual screening with few approved inhibitors to explore the possibility of repurposing existing drugs to inhibit MCT1/2, overexpressed in some cancers, including cervical cancer. This computational search pointed at Tucatinib, a well-known tyrosine kinase inhibitor (HER). Based on conservation studies, the authors derived that Tucatinib could inhibit both MCT1 and MCT2, being an interesting compound with broader action in treating cancer. The authors proved Tucatinib being effective in inhibiting MCT1 and MCT2-mediated pyruvate transport in cervical cancer HeLa cells and also verified its action in vivo.

Overall the paper is interesting and brings novel insights to the drug targeting of MCT1/2. I appreciated the use of a multidisciplinary approach to tackle inhibitory properties of available small compounds towards cancer, including virtual screening and cell biology assays. The manuscript could be made a bit more compact for the EMBOR format by merging Results and Discussion. There are 7 main Figures and Supplementary Figures, while I suppose some have to be converted to EV figures as there is a max of 5 main items in my knowledge.

I recommend publication in EMBOR with minor revisions and clarifications regarding the following points:

We appreciate the reviewer's positive evaluation of our work and its significance. We are pleased that you found our multidisciplinary approach and the novel insights into

MCT1/2 drug targeting interesting.

Regarding the structural analysis, we acknowledge the reviewer's observation about the clashscores. We have refined the models, and the highest clash overlap is controlled within 0.72 Å, which were acceptable by Protein Data Bank. Although the overall resolution is moderate, the well-defined cryo-EM density for the small molecule AR-C155858 allows us to confidently determine its binding position. This placement was further validated and refined through molecular dynamics simulations, strengthening confidence in our structural conclusions.

Concerning the manuscript format, we note the reviewer's suggestion to merge the Results and Discussion sections. We aimed to submit this study as an Article rather than a Report. According to the EMBO Reports guidelines for Articles: "There are no length limitations, but it should have more than 5 main figures and the Results and Discussion sections must be separate." Therefore, we have structured the manuscript accordingly, with 7 main figures, 4 EV figures, and an Appendix file (which includes 11 supplementary figures) to present the data comprehensively while adhering to the journal's format requirements.

-Abstract: "Cell surface glycoproteins, Basigin or embigin, associate with proton-coupled 36 monocarboxylate transporters (MCTs) to form heterodimers", concept recalled in introduction and discussion.

Are all the proton-coupled MCT forming complexes with embigin or only MCT1 and 2? what determines the specificity for basigin and embigin with respect to MCT1 and MCT2? Why only MCT1-4 and not others, such as MCT8? I suggest to be more accurate in properly introducing/discussing this point relying on conservation or published data

We sincerely thank the reviewer for raising this critical and precise point. In fact, the question of whether MCT family members require auxiliary proteins for their function has long been a key research focus of our research group.

MCTs belong to the solute carrier family 16 (SLC16) family of the Major facilitator superfamily (MFS) (Halestrap, 2013; Yan, 2015). Among the fourteen identified MCTs, MCT1-MCT4 catalyze proton-coupled influx or efflux of monocarboxylate (Halestrap and Meredith, 2004; Poole and Halestrap, 1993); the transport direction is determined by the proton motive force and the concentration gradient of the substrate monocarboxylate. Proper localization and function of MCT1-4 require glycosylated chaperon proteins, such as Basigin (also known as CD147 or EMMPRIN) for MCT1/3/4 and embigin (also known as gp70) for MCT2 (Halestrap and Wilson, 2012; Kirk et al., 2000; Philp et al., 2003; Wilson et al., 2005).

Indeed, we confirmed the observation in our previous study that embigin facilitates MCT1 localization to plasma membrane and we solved the cryo-EM structure of

MCT1-embigin complex (Xu et al., 2022). Through structural comparison, the structure of the MCT1-embigin complex is highly similar to that of the MCT1-basigin-2 complex (Wang et al., 2021), which is not surprising given that embigin is homologous with basigin, especially in the transmembrane helix. MCT1 also associates tightly with a hybrid protein containing the extracellular domain of CD2 but the transmembrane and intracellular domains of CD147 suggests that it is the transmembrane and cytoplasmic regions of CD147 that are important in the interaction. These domains are strongly conserved in members of the basigin family, consistent with the ability of embigin to interact with MCT1 (Kirk et al., 2000). Moreover, the binding affinities of MCT1-basigin-2 complex for pyruvate and L-lactate are similar to that of MCT1-embigin complex (Xu et al., 2022), consistent with that from a previous study (Wilson et al., 2005).

Interestingly, our previous work confirmed that embigin is unnecessary for the intracellular trafficking of MCT2 to the plasma membrane. And MCT2 indeed forms homodimer in the absence of embigin (Zhang et al., 2020). In addition, we confirmed in later study that MCT1 alone also forms homodimer, albeit defective in trafficking to the plasma membrane (Xu et al., 2022). Zhang et al. previously demonstrated the cooperative transport of MCT2 homodimer that plays important physiological roles (Zhang et al., 2020). Since the cooperative transport arises from the strong inter-subunit cooperativity of MCT, it is then reasonable to assume that MCT homodimer exists.

Indeed, we propose that both MCT1 and MCT2 can exist as homodimers or heterodimers. Given the high structural homology between embigin and Basigin, we consider their roles in facilitating the membrane localization and transport function of MCT1 and MCT2 to be functionally analogous. In vivo, however, different complexes may be preferentially expressed in distinct tissues. Based on our current structural and functional data, their functional differences in vitro appear to be minimal.

Furthermore, we have successfully purified complexes of MCT4 with either embigin or Basigin in vitro, and related investigations are ongoing. As the emphasis of the present study is primarily on the interaction between MCTs and inhibitors, we did not explore this specific aspect of chaperone binding in greater detail. But as the reviewer rightly pointed out, the selectivity and precise mechanisms governing MCT–chaperone interactions represent an important and fascinating research direction—one that has long been a focus of our research group.

Of the MCT family members, MCT1–MCT4 are proton-coupled monocarboxylate transporters and represent the well-established isoforms that require association with either Basigin or embigin for their functional maturation and proper membrane localization. In contrast, other members such as MCT8 (SLC16A2) and MCT10 (SLC16A10) exhibit distinct functional roles: MCT8 is a thyroid hormone transporter (Tan et al., 2025; Ge et al., 2025), while MCT10 primarily transports aromatic amino

acids (Bågenholm V et al., 2025; Tassinari M et al., 2025). Both MCT8 and MCT10 operate through different mechanisms, possess unique substrate profiles, and importantly, do not depend on Basigin or embigin for their chaperone-assisted maturation or function.

This comment has significantly improved the precision and scientific rigor of our manuscript. We have revised the Introduction by adding a dedicated paragraph describing the roles of Basigin and embigin as chaperones for MCT transporters.

-Abstract: "the deepening (..)" I would rephrase in: A deeper understanding of MCT regulation may open avenues for the development of novel inhibitors (..)

We appreciate the reviewer's suggestion to improve the manuscript, and we have amended the text accordingly.

-Line 174: perhaps you could remove this sentence to make the text flow better. Having well defined SS does not mean you will properly see the ligand: "The secondary structure features of MCT2R-emb are well-defined, allowing us to visualize the binding mode of AR-C155858(Fig.S3and S4).We will discuss the inhibition mechanism in a later session"

We appreciate the reviewer's suggestion to improve the manuscript, and we have removed the sentence from Line 174 as suggested and integrated its content into the "Inhibition of MCT2 by AR-C155858" section to improve the logical flow of the text.

-Fig4 legend: typo 'in the present' > "in the presence. Please check this typo across the paper, there are 4

We appreciate the reviewer's suggestion to improve the manuscript, and we have amended the text accordingly.

-There is a lot of yellow highlighted text, but I don't know whether that's functional?

The yellow highlights identify new data added during manuscript transfer from The EMBO Journal to EMBO Reports, as requested by the editorial office. These markings are for revision tracking only and will not appear in the final article.

-line 207: "amino acid linker (GGSG), single transfection led to substantial expression of the fusion protein at the cell surface". What do you mean by "substantial"? This variation in localisation should be shown by better images as in the authors' Cell Report article and by some semi-quantitative localisation assessment (for example in Fiji)

We appreciate the reviewer's careful attention to the wording. By "substantial", we

simply intended to indicate that the fusion protein with the (GGSG) linker showed significantly improved expression and localization to the cell membrane in both HEK293T and HeLa cells, demonstrating that the linker does not impair the membrane localization of MCT2. No additional quantitative implication was intended.

Regarding the imaging quality, we fully understand the reviewer's point. The confocal images in our previous Cell Reports article were acquired by a collaboration partner using highly advanced imaging systems. In contrast, the images in this study were obtained using the core imaging facility at our own institution, where equipment specifications may differ, resulting in variations in resolution and contrast. We have provided the best images achievable under our current experimental conditions.

It is also worth noting that in the Cell Reports paper, colocalization images were presented as standalone figures, whereas in the current manuscript they are shown as panels within a composite figure, which may further affect the visual impression of image quality.

The primary purpose of these images was to provide a qualitative assessment of membrane localization rather than to perform detailed semi-quantitative colocalization analysis. Nevertheless, we thank the reviewer for this thoughtful suggestion, and we hope that future upgrades to our institutional imaging platform will allow us to provide higher-resolution data in subsequent studies.

-Do you think AR-C155858 inhibits in the same way dimeric MCT2 and MCT2-emb, given the impossibility of isolating the homo-dimeric MCT2 in complex with AR-C155858? Can you better clarify this point?

The reviewer raised an interesting question. Indeed, the inhibition of MCT2 by AR-C155858 is modulated by embigin (Ovens et al., 2010). Our structural and functional data suggest that although MCT2 can form homodimers in the absence of embigin, the association with embigin alters the conformation and properties of the ligand-binding pocket, consistent with previous reports that MCT1 exhibits significant differences in substrate-binding affinity upon chaperone binding (Xu et al., 2022). The MCT2-embigin heterodimer structures in complex with AR-C155858 clearly show the inhibitor bound within the central cavity, supporting the notion that embigin influences the binding mode of AR-C155858.

Beyond the possibility of differential binding mechanisms, we also acknowledge that the inability to resolve the homodimeric MCT2-AR-C155858 complex could be attributed to several technical factors, such as sample heterogeneity, suboptimal vitrification conditions, or limitations in current reconstruction algorithms. To reflect these considerations and enhance the rigor of our discussion, we have toned down the causal claims in the text and now present the interpretation regarding embigin's

influence as a plausible explanation among other technical possibilities.

-Line 268: this sentence may be confusing for the reader, can you explain better?

"This signal was significantly decreased upon pre-incubation with AR-C155858. However, due to the overlapping binding sites, the tested mutations affected the transport activity of MCT2-embigin and escaped inhibition of AR-C155858 (Fig. 2I)." I recommend reporting on the bar plots the % of inhibition normalising the signal with respect to the single construct.

We thank the reviewer for this suggestion, which has helped us clarify a potentially confusing passage. The sentence aimed to convey that since the binding site of AR-C155858 overlaps with the substrate transport pathway, mutations introduced at these positions inherently disrupt the transport function of MCT2-embigin. As a result, the inhibitory effect of AR-C155858 could not be meaningfully assessed in these functionally impaired mutants, as there was little residual activity left to inhibit.

We acknowledge that this experimental design had limitations in directly demonstrating the binding site. To more rigorously prove that AR-C155858 binds at this specific location, we performed binding affinity assays between the mutant proteins and AR-C155858. The results clearly showed a substantial decrease in binding affinity in the mutants compared to the wild-type complex (Fig. 2 J-L, Fig. EV 3 A-E), confirming that these residues are critical for inhibitor binding. And we have also modified the content of the original text.

Furthermore, following the reviewer's recommendation, we have updated the bar plots in Fig. 2I to include the percentage of transport signal normalized to the respective untreated control for each construct, providing a clearer quantitative assessment of the inhibition.

-Line 278: can you introduce HER acronym?

The acronym "HER" refers to Human Epidermal Growth Factor Receptor 2 (HER2), a clinically validated oncology target. Tucatinib is an FDA-approved HER2 inhibitor used in metastatic breast cancer.

We have revised the text to explicitly define this abbreviation upon its first appearance.

-279 "in both MCT1 ..." : "by both MCT1..." may sound better

We appreciate the reviewer's suggestion to improve the manuscript, and we have amended the text accordingly.

-Is there any pH dependence of compound inhibitory properties and MCT2-emb association? I would mention it in the discussion

We thank the reviewer for raising this important point. To directly address the pH dependence of inhibition, we performed additional experiments measuring the inhibitory effect of tucatinib on MCT2-embigin-mediated transport across a range of pH conditions (Fig. EV 4 A-F).

The results indicate that tucatinib effectively inhibits transport under acidic to physiological pH conditions (pH 4–7.4), with IC_{50} values consistently ranging between ~2–7 μ M. However, its inhibitory activity markedly decreased under alkaline conditions. Although an IC_{50} of 5.44 μ M could be computationally derived at pH 8.0, the overall inhibition was substantially weaker. At pH 9.0, almost no inhibition was observed even at high concentrations of Tucatinib, and the dose-response curve could not be reliably fitted.

This phenomenon aligns with the proton-coupled transport mechanism of MCTs, where extracellular protonation regulates substrate binding. Critically, the acidic tumor microenvironment preserves Tucatinib's efficacy. We selected pH 7.4 for standardized assays to model physiological conditions while ensuring relevance to oncological applications.

We have included these new data in Fig. EV 4 and added a corresponding discussion of the pH-dependent inhibitory profile in the main text and Discussion part.

-line 339: can you explain what a CCK-8 is?

Certainly. We apologize for the lack of clarity. CCK-8 stands for Cell Counting Kit-8, a widely used colorimetric assay for sensitive and convenient detection of cell proliferation and cytotoxicity. It utilizes a highly water-soluble tetrazolium salt (WST-8) that is reduced by cellular dehydrogenases to an orange-colored formazan product, which is soluble in the culture medium. The amount of formazan generated is directly proportional to the number of living cells. We have now defined this abbreviation in Methods and Protocols section in the revised manuscript to improve readability.

-line 352-358: can you better motivate why these mutants are not inhibited?

We thank the reviewer for raising this point, which allows us to further clarify the rationale behind using these binding-site mutants. The first reviewer also raised the same question. The primary objective of this experiment was not to achieve functional rescue of cell viability, but rather to perform a stringent genetic test of drug specificity.

The critical observation is that, in contrast to cells rescued with wild-type MCTs—which regained sensitivity to Tucatinib—cells expressing the

binding-deficient mutants showed no additional inhibitory effect from the drug. This “double-negative” outcome (low baseline viability + no drug response) demonstrates that the inhibitory effect of Tucatinib strictly depends on its ability to bind specifically to these phenylalanine residues.

The mutants MCT1-F278A and MCT2-F351A were designed based on structural and biochemical evidence to specifically disrupt Tucatinib binding (Fig. 4). Direct binding assays confirmed that the F278A mutation completely abrogates Tucatinib binding to MCT1, while the F351A mutation severely reduces its affinity for MCT2 (Fig. 4 P). Thus, the lack of pharmacological response in mutant-rescued cells is consistent with the loss of molecular interaction. Their inability to confer drug sensitivity, even in the presence of Tucatinib, provides compelling evidence that the compound inhibits proliferation through on-target engagement with MCT1/2 at these specific residues.

We have included further analysis of these findings in the Discussion section.

-line 382: How is migration/growth relate to pyruvate uptake? Can you better describe the link between the MCT2-dependant transport and the measured property in cells?

We thank the reviewer for this important question regarding the link between MCT2-dependent pyruvate uptake and the observed cellular phenotypes. The connection is rooted in the central role of monocarboxylate transport in cancer cell metabolism.

MCT1/2/4 are key transporters that facilitate the movement of lactate and pyruvate across the plasma membrane, supporting multiple pro-tumor mechanisms such as metabolic symbiosis (Pisarsky et al., 2016; Sonveaux et al., 2008), angiogenesis (De Saedeleer et al., 2012; Sonveaux et al., 2012), and immune evasion (Dietl et al., 2010; Fischer et al., 2007). Although we directly measured pyruvate uptake in this study—using a well-established pyruvate sensor (Arce-Molina R et al., 2016) based on our previous experimental system (Zhang et al., 2016). Since Tucatinib acts as a competitive inhibitor binding the substrate cavity of MCT1/2 (Fig. 4), it impedes the transport of not only pyruvate but also lactate. In fact, our laboratory has also developed a lactate sensor (Wang et al., 2024). Detection using this sensor confirmed that Tucatinib can inhibit lactate transport, although the effect was less pronounced compared to its inhibition of pyruvate uptake. Therefore, we opted to retain the current pyruvate-based detection method for this study.

Lactate serves as a critical energy source and signaling molecule in tumors. Glycolytic cells export lactate, which is then taken up by oxidative tumor or endothelial cells to fuel ATP production via the TCA cycle. By inhibiting MCT1/2-mediated uptake of monocarboxylates, Tucatinib disrupts this metabolic coupling. The resulting depletion in intracellular ATP and biosynthetic precursors—required for energy-intensive processes such as DNA synthesis, protein

production, and cytoskeletal remodeling—directly impedes cell proliferation and migration.

Thus, the inhibition of pyruvate uptake serves as a direct and functional readout of MCT1 or MCT2 transport activity. The subsequent suppression of growth and migration validates the essential role of monocarboxylate metabolism in sustaining these cancer-relevant phenotypes.

We acknowledge that complementary experiments using lactate sensors could further strengthen this conclusion, and such approaches may be considered in future studies.

-It is remarkable that you find Tucatinib fits in the central gate of the MCT2 and can indeed inhibit both MCT1 and MCT2. However, the affinity is quite low (in the range of μM rather than nM as for AR-C155858) and there is no available structure of tucatinib in complex with MCT1 and MCT2. In cells, (Fig4J) it would be useful to show the relative % inhibition with respect to each construct. In vitro, have you tested by Biacore assay that at least one of these mutants is not inhibited by tucatinib? This would strengthen the hypothesis that the inhibitor binds in the central pocket.

We thank the reviewer for their positive feedback on the identification of Tucatinib as a dual MCT1/2 inhibitor and for their valuable suggestions to strengthen our study.

We would like to clarify that the affinity values were obtained using different experimental methods and target systems. In previously published studies, AR-C155858 has been reported as a potent inhibitor of MCTs, with K_i values of 2.3 nM for MCT1 and <10 nM for MCT2. These K_i values were derived from experiments measuring the inhibition of L-lactate uptake into rat erythrocytes by AR-C155858 (Ovens et al., 2010a; Ovens et al., 2010b).

Regarding AZD3965, some researchers reported K_d values of 1.6 nM for MCT1 and 9.6 nM for MCT2 (Critchlow et al., 2012). In another study, the affinity of AZD3965 for MCT1 was approximately 40 nM , as determined by microscale thermophoresis (MST) binding assays (Wang et al., 2021).

In our study, we purified the MCT1-Basigin-2 and MCT2-embigin complexes and evaluated their binding affinities using surface plasmon resonance (SPR) and bio-layer interferometry (BLI). As shown in Figures A and B below, the SPR results showed K_d values of 1.91 μM for MCT1-Basigin-2 and 6.99 μM for MCT2-embigin with AR-C155858. The BLI assays yielded K_D values of 0.1 μM for MCT1-Basigin-2 and 0.6 μM for MCT2-embigin with AR-C155858 (Fig. 2 J). Unfortunately, we did not obtain detectable signals in MST assays. Moreover, we determined the binding affinity between Tucatinib and MCT1 or MCT2 using two independent methods. As shown in panels C and D below, surface plasmon resonance (SPR) measurements yielded affinity values of 5.08 μM for MCT1-Basigin-2 and

8.47 μM for MCT2-embigin. In addition, bio-layer interferometry (BLI) assays—presented in Figure 4 of the main text—demonstrated affinities of 3.9 μM for MCT1-Basigin-2 (Fig. 4 J) and 0.6 μM for MCT2-embigin (Fig. 4 M). Therefore, we believe that differences in experimental methods and target constructs may contribute to the variations in the reported affinity values.

To maintain consistency in experimental methodology, we uniformly selected the data from bio-layer interferometry (BLI) assays for presentation in the results. The results indicate that the affinity of Tucatinib for MCT1-Basigin-2 (3.9 μM , Fig. 4 J) is one order of magnitude lower than that of AR-C155858 (0.1 μM , Fig. 2 K), while its affinity for MCT2-embigin (0.6 μM , Fig. 4 M) is within the same order of magnitude as that of AR-C155858 (0.6 μM , Fig. 2 J). Therefore, we consider Tucatinib to also be a potent inhibitor of both MCT1 and MCT2.

Figure for referee with unpublished data and its description has been removed upon request by the authors.

Furthermore, to directly test the binding hypothesis, we performed BLI assays to measure the affinity of Tucatinib for wild-type versus mutant MCT1 and MCT2. The results confirmed that mutations in the central binding pocket led to various degrees of reduction in binding affinity for Tucatinib (Fig. 4 J-P). Consistent with the structural analysis, F278A resulted in undetectable binding between Tucatinib and MCT1 (data not detectable), and Y34A and R313A weakened the interaction by

approximately 11 and 15 folds, respectively (Fig. 4 J-L). Among the three MCT2 mutants, F351A exhibited the most pronounced effect, with a 370-fold reduction in binding affinity. The other two mutants, Y34A and F262A, showed reductions of 24-fold and 58-fold, respectively (Fig. 4 M-P). These results demonstrate that Tucatinib binds specifically to the substrate-binding pocket of MCT1 or MCT2. A description of these new experimental results and their interpretation has been added to the main text.

-Because of the μM you probably need larger amounts of tucatinib to exert an effect on MCT1/2 by saturating them, yet you only expose the cells to $5\mu\text{M}$ of the compound, close to the K_d . Do you think in your cellular setup that the compound is actually inhibiting MCT1/2 or the kinase inhibition is prevailing at these concentrations? Could you elaborate on this please and discuss future applications?

We thank the reviewer for raising this important point regarding the concentration selection and potential kinase-mediated effects of Tucatinib in our cellular assays.

In the CCK-8 cell viability assay, a concentration gradient of Tucatinib ranging from $3.125\ \mu\text{M}$ to $100\ \mu\text{M}$ was tested. The half-maximal inhibitory concentration (IC_{50}) was determined to be approximately $12.5\ \mu\text{M}$. Based on this result, a concentration of $12.5\ \mu\text{M}$ was selected for all subsequent experiments in the knockdown cell lines, including functional assays such as the wound healing assay. This concentration is approximately 3.2-fold higher than the measured K_d value for MCT1 ($3.9\ \mu\text{M}$) and 20.8-fold higher than that for MCT2 ($0.6\ \mu\text{M}$), was chosen to ensure strong and consistent target inhibition across cellular contexts and to facilitate comparative analysis in further experiments. We consider this concentration appropriate for achieving robust pharmacological effects in our experimental setting.

Although Tucatinib is a known HER2 inhibitor (Casak et al., 2023), we would like to emphasize that HeLa cells express very low levels of HER2 (Jia et al., 2022). As confirmed by qPCR, HER2 mRNA expression was less than 7% of that of MCT1/2 in this cell line (Appendix Fig. S9). Therefore, it is unlikely that HER2 inhibition significantly contributes to the anti-proliferative or anti-migratory effects observed in our study. Instead, our genetic and pharmacological data consistently support that the effects are primarily mediated through MCT1/2 inhibition. More importantly, our work reveals a novel off-target activity of Tucatinib and suggests its potential therapeutic utility—particularly in HER2-deficient but MCT1/2-high cancers.

We fully acknowledge that the micromolar-level potency of Tucatinib toward MCT1/2 is not optimal. However, this study serves as an important proof of concept that repurposing existing drugs can yield novel MCT1/2 inhibitors. Future efforts will focus on optimizing this compound through structure-based drug design, AI-assisted screening, and functional group modification to improve both potency and specificity.

Reviewer #3:

Summary

Xu et al. present cryo-EM structures of the heterodimer of MCT2 and embigin, both in the apo state and with a bound inhibitor. Embigin likely facilitates plasma membrane localisation of MCT2, which enables correct function. They also identify an inhibitor, Tucatinib, with potential relevance for the treatment of cervical cancer. The data is sound, and covers a wide array of analysis, thus perhaps being too long and having a scope that is too wide for what is requested for this journal, presenting multiple structures and their validation and analysis, as well as drug target determination and testing. The findings are novel and of high significance, as it can directly lead to trials on cervical cancer for this drug target. For further issues, see the major and minor comments, below.

Response to requested reviewer questions:

1. Does this manuscript report a single key finding? NO

There are two key findings: the structures with and without inhibitor, and the determined cancer drug target.

2. Is the reported work of significance (YES), or does it describe a confirmatory finding or one that has already been documented using other methods or in other organisms etc (NO)? YES

3. Is it of general interest to the molecular biology community? YES

Further understanding of a cancer-relevant protein, including identifying a promising drug target.

4. Is the single major finding robustly documented using independent lines of experimental evidence (YES), or is it really just a preliminary report requiring significant further data to become convincing, and thus more suited to a longer-format article (NO)? YES

We sincerely thank the reviewer for their overall positive assessment of the significance and novelty of our work, and for acknowledging the potential impact of our findings.

We also appreciate the comment regarding the broad scope of the study. Our aim was to provide a comprehensive mechanistic narrative—from structural determination and functional validation to drug repurposing and cellular phenotyping—to firmly establish MCT1 or MCT2 as a target and Tucatinib as a promising inhibitor. We believe this multifaceted approach is necessary to fully support the conclusions and translational potential of the work.

Regarding the manuscript format, we note the suggestion to potentially condense the presentation. We would like to clarify that this study was submitted as an Article, which, according to the EMBO Reports guidelines, has no length limitations but requires more than 5 main figures and separate Results and Discussion sections. Our

structure with 7 main figures, 4 EV figures, and an Appendix file was designed to thoroughly present the data while complying with these format specifications.

We are certainly open to suggestions on how to improve the narrative flow or emphasis should the editor and reviewers recommend further adjustments.

Specifically:

Major comments

1. Please overview the language, as there are numerous smaller grammatical errors. Examples (corrections underlined) include line 62: "monocarboxylates is primarily" and line 83: "research that demonstrates"

We sincerely thank the reviewer for his/her thorough review and for pointing out the grammatical inaccuracies throughout the text. We have carefully reviewed the entire manuscript and corrected all such errors.

2. Please also revise the figure numbering, as several figures are referenced out of order. This applies both to the overall numbering of figures (S3 is referenced before S2, for instance), but also the organisation within each figure (figures 5 and 6 could be split into 3 figures and reorganised to better align with how they are referenced in the text, for instance). Table S2 is never referenced in the text.

We thank the reviewer for their meticulous attention to the organization and referencing of figures and tables. We have carefully revised the figure numbering and in-text citations throughout the manuscript to ensure all are referenced in correct order.

Specifically, we have reorganized Figures 5 and 6 to improve clarity and alignment with the text. Additionally, all supplementary figures and tables, including Table S2, have been cross-checked to ensure every item is appropriately referenced in sequence.

We believe these revisions significantly improve the readability and coherence of the manuscript.

3. The introduction needs to present what is known about MCT structure, its topology, transport, dimeric states, key residues etc. This is especially relevant regarding line 78, where it is implied that the reader already knows about the dimeric state of MCT1, and line 355, where key residues are mentioned. Some text starting at line 176 could perhaps be placed here instead. A thorough introduction of dimeric state is especially important, considering the data presented. When are these proteins dimeric? How does their dimeric state change? How does their different dimeric states affect transport activity? In addition, considering the significant amount of comparison with MCT1, it is relevant with a more detailed introduction to their similarities and,

importantly, differences.

We thank the reviewer for this critical suggestion. In response, we have added a new paragraph in the Introduction that provides a more comprehensive overview of MCT protein structure, topology, and the functional implications of their different oligomeric states, including how these states influence transport activity. We also explicitly introduce the key residues involved in substrate recognition.

Regarding the key residues mentioned around line 355 (Phe278 in MCT1 and Phe351 in MCT2), these were identified in our study as critical anchor points for Tucatinib binding (Fig 4). While other residues also contribute to the interaction, these two phenylalanine residues were selected for mutagenesis and functional validation due to their pronounced impact on binding affinity, as confirmed by our biochemical assays. We performed BLI assays to measure the affinity of Tucatinib for wild-type versus mutant MCT1 and MCT2. The results confirmed that mutations in the central binding pocket led to various degrees of reduction in binding affinity for Tucatinib (Fig. 4 J-P). Consistent with the structural analysis, F278A resulted in undetectable binding between Tucatinib and MCT1 (data not detectable). Among the three MCT2 mutants, F351A exhibited the most pronounced effect, with a 350-fold reduction in binding affinity (Fig. 4 P). A description of these new experimental results and their interpretation has been added to the main text.

4. What is the role of embigin? In the paragraph starting a line 197 it is shown that MCT2 colocalises with embigin at the plasma membrane. However, according to Zhang et al, 2020, cited in the paper, MCT2 can localise to the plasma membrane also without the presence of embigin. Were control experiments done where localisation of MCT2 was assessed without the presence of embigin? It appears that MCT2 is capable of transport as a homodimer, and if it also localises correctly, what is the role of embigin *in vivo*? Is it known if transport is possible for the heterodimer, or is that only carried out by the homodimer? Also regarding the role of the heterodimer, is it then assumed that the AR-C155858 only binds to the heterodimer? Would inhibition in part be a lack of activity because the inhibitor locks MCT2 in the heterodimeric state? An analysis of this is especially lacking in the discussion.

We thank the reviewer for these insightful questions regarding the role of embigin.

Based on existing structural and functional data, it is well-established that MCT1 and MCT2 can function properly in the presence of ancillary proteins (Wang et al., 2021; Xu et al., 2022). This leads us to a fundamental question for discussion: whether endogenous MCTs homodimer exist?

Previous studies revealed that MCT1 is co-expressed with an ancillary protein, basigin or embigin, in mammalian cells (Kirk et al., 2000), and MCT2 with embigin

(Wilson et al., 2005). Indeed, we confirmed the observation in our study that embigin facilitates MCT1 localization to plasma membrane (Xu et al., 2022).

However, three reasons lead us to believe that MCT homodimer does biologically exist and plays an important physiological role. First, our previous work confirmed that embigin is unnecessary for the intracellular trafficking of MCT2 to the plasma membrane. And MCT2 indeed forms homodimer in the absence of embigin (Zhang et al., 2020). In addition, we confirmed that MCT1 alone also forms homodimer, albeit defective in trafficking to the plasma membrane (Xu et al., 2022). Furthermore, Ovens et al. demonstrated that expression of active MCT2 at the plasma membrane of oocytes was significantly enhanced by co-expression of exogenous embigin (Ovens et al., 2010). However, even without the co-expression of exogenous embigin, MCT2 was detected by western blotting at least half amount in comparison with that with the co-expression of exogenous embigin in the membrane preparations of oocytes (Figure 1B, Ovens et al., 2010).

Second, multicellular organisms need to regulate MCT activity to maintain their intracellular monocarboxylate homeostasis. Traditional studies focus on transcriptional regulation of protein expression, or post-transcriptional regulation of protein modification. However, these long-term regulations of MCT activity are logically insufficient for adaptation of multicellular organisms to a quick metabolic change. For example, during vigorous exercise, when the adenosine triphosphate (ATP) demand outstrips the availability of oxygen, skeletal muscle shifts from mitochondrial oxidative phosphorylation to glycolytic ATP generation, and quickly raises the intracellular monocarboxylate concentration. This indicates the existence of a short-term regulation of MCT activity to maintain the intracellular monocarboxylate homeostasis, which is critical for the survival of multicellular organisms in an evolving environment. Indeed, we previously demonstrated the cooperative transport of MCT2 homodimer that plays important physiological roles (Zhang et al., 2020). Since the cooperative transport arises from the strong inter-subunit cooperativity of MCT, it is then reasonable to assume that MCT homodimer exists.

Third, all membranes of the SLC16 family contain two highly conserved signature sequences, one at the N-terminus of TM1, while the other one constitutes the loop between TM4 and TM5 and the beginning of TM5 (Halestrap and Price, 1999; Halestrap, 2013). In the MCT2 homodimer, the two signature motifs are involved in inter-subunit interactions. Moreover, mutation on the signature sequence renders the MCT2 loss of cooperativity (Zhang et al., 2020). These data also strongly support the biological existence of MCT homodimer.

Therefore, based on previous findings, we propose that both the ancillary protein-bound heterodimer and the self-assembled homodimer of MCTs are physiologically functional. It is possible that their prevalence varies across tissues, cell types, or species.

According to our confocal imaging results, homodimeric MCT2 is observed not only on the plasma membrane but also within the cell (Fig S1a, Zhang et al., 2020). Even its plasma membrane localization may be facilitated by endogenously expressed ancillary proteins. We thus hypothesize that homodimeric MCT2 may also play a functional role on organelle membranes, such as the mitochondrial membrane, though this speculation requires further experimental validation.

Beyond mere localization, we propose that embigin's primary role is that of a functional modulator. As suggested by the Halestrap group, ancillary proteins can allosterically regulate MCT transport activity. Moreover, we observed that MCT1 fails to traffic to the plasma membrane without the help of embigin (Xu et al., 2022). Our structural data support this, showing that embigin binding induces conformational changes in MCT2.

Regarding activity, both homodimeric and heterodimeric (chaperone-bound) states are believed to be transport-competent. Both the homodimeric MCT2 (Zhang et al., 2020) and the heterodimeric MCT2-embigin complex (Fig. 2I, this study) can be inhibited by AR-C155858, effectively blocking their transport activity. However, the efficacy of inhibition may differ between these states, which represents one of the key directions for our future research. Its mechanism likely involves stabilizing a specific conformational state rather than exclusively "locking" the transporter in a heterodimeric form. Beyond the possibility of differential binding mechanisms, we also acknowledge that the inability to resolve the homodimeric MCT2-AR-C155858 complex could be attributed to several technical factors, such as sample heterogeneity, suboptimal vitrification conditions, or limitations in current reconstruction algorithms. To reflect these considerations and enhance the rigor of our discussion, we have toned down the causal claims in the text and now present the interpretation regarding embigin's influence as a plausible explanation among other technical possibilities.

We sincerely thank the reviewer for highlighting the need to discuss this mechanism more deeply. We have now added a paragraph to the Discussion section analyzing this potential inhibition mechanism in light of our structural findings.

5. For structural comparisons there is a large focus on only comparing the individual NTD and CTD, but I lack comparisons of the protein as a whole. Examples include line 214 and figure S5. While showing that the NTD and CTD remain relatively similar is useful, this needs to be put in context of the movements of the whole protein. To emphasise this similarity, while still showing overall movement, this could be done by superimposing just, for example, the NTD, but showing the whole protein, thus giving the reader a chance to see the overall changes, while still appreciating the domain similarities. Leaving out a whole protein comparison makes it hard to put the current images and analysis into context.

We thank the reviewer for this excellent suggestion. We have revised Figure S5 (Fig. EV 1 in the updated version) and the corresponding description in the main text to include a whole-protein structural alignment. As recommended, we superimposed the N-terminal domain (NTD) and now display the entire protein structure (Fig. EV 1 C in the updated version), which allows readers to better visualize both the local domain conservation and the global conformational changes of the full-length MCT2-embigin complex. This adjustment provides a more comprehensive context for interpreting the structural dynamics described in the manuscript.

Minor comments

1. Why are some regions highlighted in yellow?

The yellow highlights identify new data added during manuscript transfer from EMBO Journal to EMBO Reports, as requested by the editorial office. These markings are for revision tracking only and will not appear in the final article.

2. Line 103: MCT2 has a K_i value >10 nM. Why is the specific value given for MCT1, but not MCT2?

We thank the reviewer for raising this important point regarding the reported K_i values. The reviewer is correct to note the discrepancy in specificity.

The reason a precise K_i value is provided for MCT1 but not for MCT2 is that, to the best of our knowledge, an exact K_i value for the inhibition of MCT2 by AR-C155858 has not been definitively established in the literature. As noted in (Ovens et al., 2010a), the data suggest potent inhibition with a K_i "significantly less than 10 nM," but the exact figure remains undetermined. Furthermore, (Ovens et al., 2010b) express that "Although our data do not allow determination of absolute K_i values for AR-C155858, Figures 2 and 3 show that, when associated with embigin, the inhibitor sensitivity of MCT2 decreased relative to that of MCT2 associated with endogenous basigin."

We acknowledge that our original phrasing in Line 103 was imprecise. We have revised the text to clearly state that the K_i for MCT2 is estimated to be <10 nM, reflecting the potent but not yet precisely quantified nature of the inhibition, while the specific value for MCT1 is well-defined. We thank the reviewer for prompting this clarification, which improves the accuracy of our manuscript.

3. 216: "... conformational change of TM5 still exists," this is confusing, as it makes it sound like the significant conformational change that is seen in the homodimer can also be seen in the heterodimer, which is not what I think you mean?

We apologize for the lack of clarity in our original statement. The sentence has been revised to accurately reflect our intended meaning. Additionally, as noted, we have expanded the Introduction to include a clearer preliminary explanation of this

structural difference to provide better context for the reader.

4. 246: also mention the molecule bound to the MCT1 structure here, along with the PDB ID.

Thanks for your kind reminding. We have added complete information regarding the protein–small molecule complex and included the corresponding reference.

5. 249: "despite binding to different proteins ..." is confusingly worded, since AR-C155858 is not bound in both structures, only one of them. Please rephrase.

We appreciate the reviewer's suggestion to improve the manuscript. We have revised the sentence to clarify the intended meaning.

6. 255 and figure 2D: what docking is referred to here? The drug screening? Why was docking otherwise carried out, since you have the structure? This needs to be more clear in the text and figure legend.

Molecular docking was performed to rigorously validate the accuracy of the cryo-EM model building. Due to limited resolution in certain regions, atomic-level details could not be unambiguously determined during model construction. Docking was used to verify the molecular orientation of AR-C155858 within MCT2, serving as a computational method to confirm the reliability of the cryo-EM structure. We have clarified this rationale in the main text and updated the figure legend accordingly.

7. 263: what does MMGBSA stand for?

The term "MM-GBSA" refers to Molecular Mechanics combined with Generalized Born and Surface Area solvation. It was applied to estimate the binding free energy between proteins and small molecules. Detailed methodology is described in the MD simulations section of the Methods and Protocols.

8. 270: it has not been stated where the cargo is expected to bind.

We acknowledge that this experimental design had limitations in directly demonstrating the binding site. To more rigorously prove that AR-C155858 binds at this specific location, we performed binding affinity assays between the mutant proteins and AR-C155858. The results clearly showed a substantial decrease in binding affinity in the mutants compared to the wild-type complex (Fig. 2 J, L, Fig. EV3 A-E), confirming that these residues are critical for inhibitor binding. And we have also modified the content of the original text.

9. 291: if Tucatinib can inhibit both MCT1 and MCT2 then, by definition, is it really specific? The way this sentence is written implies that the cell usually expresses only

one of them (see major point 3 about expanding the introduction)?

We thank the reviewer for this critical comment. Indeed, from a strict definition, a compound that inhibits both MCT1 and MCT2 should be termed a “dual inhibitor” rather than a “specific inhibitor” of a single target. We have corrected this terminology throughout the manuscript to avoid confusion. Our intended meaning was that Tucatinib is specific to the MCT family (particularly MCT1/2) over unrelated off-targets. However, we agree that within the MCT family, it exhibits broad activity.

Regarding the cellular context, we have expanded the introduction as suggested (Major point 3) to provide a more detailed discussion on the expression patterns of MCT1 and MCT2.

10. 306: "simultaneously" makes it sound like the same molecule will bind to both proteins at the same time, which is not the case here.

We appreciate the reviewer’s suggestion to improve the manuscript. We have revised the sentence to clarify the intended meaning.

11. 639: why not have these primers listed along with the others in table S2?

We appreciate the reviewer’s suggestion to improve the manuscript. These primers have now been added to Appendix Table S2.

12. Figure 1C, and (to a lesser extent) 2A: Consider making the interior of the slab a darker shade of grey, currently it is almost white, which would increase figure clarity.

We thank the reviewer for this helpful suggestion. We have increased the contrast of the slab view in both Fig. 1C and Fig. 2A by using a darker shade of grey for the interior, as recommended. This adjustment significantly improves the clarity and visual interpretability of the structural features in these figures.

13. Figure 1G and H: what are these? Western blots?

These figures show Western blot results from reciprocal pull-down assays designed to validate the physical interaction between MCT2 and embigin. In our previous studies, the same methodology was employed to identify key amino acid residues involved in the MCT1-embigin interaction (Xu et al., 2022).

14. Figure 3A and B and 4A: the binding site is what is most interesting here, I would recommend zooming in so the figures mainly show that.

We thank the reviewer for this constructive suggestion. Fig. 3 A, B are schematic illustrations of the virtual screening process, showing the spatial location of the

drug-binding pocket within the overall structure. More detailed interactions are provided and annotated in Fig. 4. We have zoomed in on the binding site in Fig. 4A to focus on the inhibitor-protein interactions (Appendix Fig. S8 A, B). Key amino acid residues involved in the binding are now clearly labeled, which significantly enhances the clarity and focus of these figures.

15. Figure 3C: how were these targets selected to be shown here? Highest predicted affinity among those screened?

The six compounds shown in Fig. 3C were selected from a virtual screening campaign targeting the identified binding pocket. This screening was performed using AutoDock Vina, prioritizing compounds based on the highest predicted binding affinity scores. These top-ranked compounds were advanced to experimental validation. Subsequent screening via pyruvate uptake assays (Fig. 3 D, E) revealed that Tucatinib exhibited dual inhibition of both MCT1 and MCT2 transporters. Consequently, Tucatinib was selected for further functional characterization.

16. Figure 6: the legend should list the constructs used. What does NC stand for?

In Figs. 5 and 6, “NC” denotes Negative Control, which refers to cells subjected to the identical experimental procedure but transfected with a non-targeting construct. This control is essential to distinguish whether observed effects result specifically from the knockdown of the target gene or from non-specific artifacts induced by the experimental manipulation itself. We added the constructs used at the legend.

17. Figure S5B: this figure should show that embigin and the NTD move together, but right now it only shows the same thing as figure S5A. Scrapping this figure and adding an overall alignment, as suggested in major point 5, would fix this.

We have revised Figure S5 (Fig. EV 1 in the updated version) and the corresponding description in the main text to include a whole-protein structural alignment. As recommended, we superimposed the N-terminal domain (NTD) and now display the entire protein structure (Fig. EV 1 C in the updated version), which allows readers to better visualize both the local domain conservation and the global conformational changes of the full-length MCT2-embigin complex.

18. Figure S5C: It would be relevant to indicate the location of the membrane in this figure.

We appreciate the reviewer’s suggestion to improve the manuscript. We have redrawn the location of the membrane in this figure (Fig. EV 1 B in the updated version).

19. Figure S12: were the knockdown strains also measured, as a control? They should be included here

Thank you for your careful attention to this matter. We have used the knockdown strains as controls and have incorporated these data into this figure (Appendix. Fig. S11 in the updated version).

Dear Dr. Ye

Thank you for the submission of your revised manuscript to EMBO reports. We have now received the full set of referee reports that is copied below.

As you will see, the referees find that the study is significantly improved during revision and recommend publication. Before I can accept the manuscript, I need you to address some minor points below:

- Please remove the main and EV figures from the manuscript text. Only the figure legends remain (Figure Legends, followed by Expanded View Figure Legends).
- Please move the Data Availability section before the Acknowledgments. Please insert URLs that link directly to the datasets at EMDB and wwPDB, even if these are not yet public.
- Please update the 'Conflict of interest' paragraph to our new 'Disclosure and competing interests statement'. For more information see <https://www.embopress.org/page/journal/14693178/authorguide#conflictsofinterest>
- Regarding the Author Contributions, we now use CRediT to specify the contributions of each author in the journal submission system. Therefore, please remove the Author Contributions from the manuscript file and make sure that the author contributions in our online manuscript tracking system are correct and up-to-date. The information you specified in the system will be automatically retrieved and typeset into the article. You can enter additional information in the free text box provided, if you wish. See also our guide to authors <https://www.embopress.org/page/journal/14693178/authorguide#authorshipguidelines>.
- The first two references are media releases, which appear not citable and trackable. Could these be replaced by a more formal citation?
- Please add the following funding information in the manuscript tracking system: 82504854 and the Medical Science Data Center of Hebei Medical University. The information in the Acknowledgments and the one in the system must be congruent.
- The callout "Fig. S11 A and B" is missing the word "Appendix". Please correct.
- The Appendix will not be typeset and we need therefore a clean PDF, i.e., without the highlights.
- Appendix Figure S9: please provide information on the number and nature of replicates, and define bars and error bars in the legend.
- Appendix Figure S10: the scale bars in panel C are not visible. Please define the nature of the replicates, the bars and error bars for panels A, B, and D and the statistical test used. We also need the exact p-values, unless $p < 0.0001$.
- The same applies to Appendix Figure S11
- Author Checklist, C58: You state that you analysed animals in or captured from the field. I think that this is not accurate.
- Please detail housing and husbandry conditions for the mice in the Methods section. I could not find this information.
- Materials and methods should be Methods. The subtitles "Reagents and Tools Table" and "Methods and protocols" are not needed.
- We perform a routine image and data integrity check on all revised manuscripts. In this case we noticed two potential aberrations that need your attention:
 - 1) I attach here a ZIP file with color-coded .xls files. In each of the columns, the same number starts to be repeated from a different timepoint onwards. I assume this is when measurements reach a baseline. Could you please confirm/explain to make sure?
 - 2) One image seems to be shown twice in Figure 6E and 6G: (6E) sh-MCT1 48 hrs matches (6G) sh-MCT2+Tuc, 24 hrs. Please check and clarify. And please provide the source data for these panels.
- Please upload the source data as one folder per figure (zipped). Please also provide source data for microscopy images and Western blots.
- Please address the following comments from our data editors in the Figure Legends:
 - Please note that the exact p values are not provided in the legends of figures 5C, E, G, H, J, L; 6B, D, F, H; 7B, E (not needed if $p < 0.0001$).
 - Please indicate the statistical test used for data analysis in the legends of figures 5C, D, E, F, G, H, J, L; 6B, D, F, H; 7B, E
 - Please note that information related to n is missing in the legends of figures 2I, 5A, 7B, C, E.

- Please note that the error bars are not defined in the legends of figures 2I, 3D, E, G, H, I, J; 5A, C, D, E, F, G, H, J, L; 6B, D, F, H; 7B, C; EV4 A-F.
- Please make sure to specify the number and nature (technical, biological, independent) of replicates in all figure legends.
- Please provide MW markers for the Western blots.
- In Figure 5C-F, H, J, L the dots showing the individual data points are rather small and difficult to see. Could they be enlarged a bit?
- The scale bars in Figure 6A, C, E are quite thin and difficult to see.
- As a standard procedure we edit Title and Abstract. Please find my suggestions below my signature.
- Finally, EMBO Reports papers are accompanied online by
 - A) a short (1-2 sentences) summary of the findings and their significance,
 - B) 2-3 bullet points highlighting key results and
 - C) a schematic summary figure that provides a sketch of the major findings (not a data image).
 Please provide the summary figure as a separate file in PNG or JPG format at a size of 550x300-600 pixels (width x height). Please note that the size is rather small and that text needs to be readable at the final size. Please send us this information along with the revised manuscript.

Kind regards,

=====

Referee #1:

The authors have adequately addressed the points raised by the reviewer.

Referee #2:

The authors addressed all my concerns, and I appreciate that they performed additional important experiments and improved the accuracy of their text and data analysis. I recommend publication without further revision from my side.

Referee #3:

Xu et al. submitted a revised and improved manuscript, which addresses the reviewer concerns. The clarity has been improved, with the added introductory paragraph on MCT1 and 2, and issues regarding data quality have been addressed, also with the addition of BLI. I feel that my concerns have been addressed and do not see need for further review.

=====

Tucatinib Inhibits Cervical Cancer Progression by Targeting MCT1/2
OR:
Structure-guided screening identifies Tucatinib as dual inhibitor for MCT1/2

Cell surface glycoproteins, Basigin or embigin, associate with proton-coupled monocarboxylate transporters (MCTs) to form heterodimers. These associations enhance MCT membrane trafficking and modulate their transport functions. Cancer cells often undergo metabolic reprogramming, which increases glycolysis and lactate production. Proton-coupled MCTs are crucial for maintaining this metabolic state by shuttling lactate across membranes, thus aiding in pH regulation within cancer cells. A deeper understanding of MCT regulation may open avenues for the development of novel inhibitors, potentially applicable in

clinical settings. In this study, we determine the cryo-EM structures of the human MCT2-embigin complex in both apo and AR-C155858-bound states, and observe that embigin engages in extensive interactions with MCT2, facilitating its localization to the plasma membrane and substrate transport. Given the remarkable structural conservation among MCTs, we conduct virtual screening based on MCT1/2 structures and identify Tucatinib as an effective inhibitor of pyruvate transport mediated by both MCT1 and MCT2. We show that Tucatinib potently inhibits the proliferation and migration of cervical tumor cells in vitro and inhibits tumor growth in a mouse xenograft model, while exhibiting excellent biological safety. These findings offer molecular insights into the structural and functional mechanism of MCT2 and identify Tucatinib as novel dual inhibitor of both transporters.

Tianjin University

October 28, 2025
Martina Rembold
Editor, *EMBO reports*

Sheng Ye, Ph.D.
Professor
School of Life Sciences
No. 92 Weijin Rd.
Tianjin, 300072
P.R. China
Tel: 86-22-2740-3906
Fax: 86-22-2740-3902
Email: sye@tju.edu.cn

Dear Dr. Rembold,

Thank you for your handling of our manuscript EMBOR-2025-62104 “**Structure-guided screening identifies Tucatinib as dual inhibitor for MCT1/2**”. We also thank the reviewers for their positive feedback on our revised manuscript. As suggested by the Editor, we have modified the title to better reflect the scope of our study. In response to the Editor's comments, we have thoroughly revised the manuscript, which includes modifications to the title, methodology, figure legends, reference list, and the resolution of formatting issues in several figures, as well as the Appendix. For a few specific points, we offer further clarification as follows:

- We perform a routine image and data integrity check on all revised manuscripts. In this case we noticed two potential aberrations that need your attention:

1) I attach here a ZIP file with color-coded .xls files. In each of the columns, the same number starts to be repeated from a different timepoint onwards. I assume this is when measurements reach a baseline. Could you please confirm/explain to make sure?

We appreciate your thorough observation. The repeated values in the data are a result of normalization processing during the BLI measurements, which leads to this pattern. We would like to clarify that all these datasets are raw data exported directly from the analysis software without any manual intervention or modification.

- Please upload the source data as one folder per figure (zipped). Please also provide source data for microscopy images and Western blots.

Thanks for your reminding. We have uploaded the source data as one folder per figure (zipped). The source data for microscopy images (Fig. 6 A, C, E, G) have been provided. The fluorescence images for Fig. 1 I and Fig. 3 K, L are original, without any modification or cropping. Western blot images have also been compiled and consolidated into their respective figures as PDF summaries.

All other issues have been addressed in the revised document. Please refer to the updated version for details. We look forward to hearing from you.

Yours sincerely,

Sheng Ye, Ph.D.
School of Life Sciences
Tianjin University

Dear Dr. Ye

Thank you for the submission of your revised manuscript to our offices.

We have completed all editorial and QC checks and all seems fine. That said, I noted that all quantification in Fig. 5, Fig. 6, Fig. 7, Appendix Fig. S9, S10, and S11 is based on technical replicates.

Based on our editorial policies and general considerations, statistical analysis may not be done on data derived from technical replicates. In case you have data from independent experiments, please include these to test statistical significance on the means of independent replicates. In case you do not have such data, please remove all p-values and statistical analysis.

Once these changes have been implemented, please upload the revised manuscript using this link:

Link Unavailable

Kind regards,

Tianjin University

November 10, 2025
Martina Rembold
Editor, *EMBO reports*

Sheng Ye, Ph.D.
Professor
School of Life Sciences
No. 92 Weijin Rd.
Tianjin, 300072
P.R. China
Tel: 86-22-2740-3906
Fax: 86-22-2740-3902
Email: sye@tju.edu.cn

Dear Dr. Rembold,

Thank you for your handling of our manuscript EMBOR-2025-62104 “**Structure-guided screening identifies Tucatinib as dual inhibitor for MCT1/2**”.

We sincerely apologize for the confusion caused by our incorrect terminology in the initial submission. Upon reviewing your comments, we realized that we had mistakenly used the term "technical replicates" in our figure legends to describe what are, in fact, biological replicates.

All the quantitative data presented in Figure 5, Figure 6, Figure 7, and Appendix Figures S9, S10, and S11 were derived from independent biological replicates, conducted on different days with separately prepared cell cultures and reagents. The raw data we collected and analyzed are based on these independent experiments.

Following your editorial policy, we have now thoroughly revised the manuscript and the appendix file to correct this error. All mentions of "technical replicates" in the relevant sections have been replaced with the accurate term "biological replicates".

The updated files have been submitted through the portal. We deeply regret this oversight and any inconvenience it may have caused during the review process.

Thank you for your guidance and consideration.

Yours sincerely,

Sheng Ye, Ph.D.
School of Life Sciences
Tianjin University

Dr. Sheng Ye
Tianjin University
92 Weijin Rd.
Tianjin 300072
China

Dear Dr. Ye,

I am very pleased to accept your manuscript for publication in the next available issue of EMBO reports. Thank you for your contribution to our journal.

Yours sincerely,
